# SO-Lazy-BiO: Accelerating Bilevel Optimization with Reduced Second-Order Information Computation

## Abstract

Bilevel optimization has attracted significant attention recently due to its applicability in various large-scale machine learning tasks (e.g., the large language model (LLM) pretraining-finetuning pipeline). In the literature, one popular approach for solving bilevel optimization problems is to use hypergradient-based methods. However, computing the hypergradients requires evaluating second-order information (Hessians/Jacobians) of the lower-level objective function, which is computationally expensive. To address this challenge, we propose SO-Lazy-BiO (Second-Order Lazy Bilevel Optimization), an algorithmic framework that significantly accelerates the state-of-the-art (SOTA) bilevel optimization methods by allowing *infrequent* evaluation of second-order information. We theoretically establish the performance of SO-Lazy-BiO and show that, despite the additional errors incurred by the infrequent evaluations of second-order information, SO-Lazy-BiO *surprisingly* matches the computation complexity of existing non-lazy bilevel algorithms, while requiring *fewer* second-order information evaluations. This leads to substantial savings in both computational cost and wall-clock running time. We further conduct extensive experiments to demonstrate that SO-Lazy-BiO enjoys significant gains in numerical performance compared to SOTA, especially for large-scale tasks. To our knowledge, this is the first work to employ infrequent second-order computations while still guaranteeing the convergence of stochastic bilevel algorithms.

## 1 Introduction

**1) Background and Motivation:** Bilevel optimization refers to the class of problems with two levels of hierarchy, where the solution of the upper-level (UL) problem depends on the minimizer of the lower-level (LL). Formally, we have

$$\min_{\mathbf{x}\in\mathbb{R}^u}\left\{\ell(\mathbf{x})\triangleq f\left(\mathbf{x},\mathbf{y}^*(\mathbf{x})\right)\triangleq\mathbb{E}_{\xi\sim\pi_f}\left[f\left(\mathbf{x},\mathbf{y}^*(\mathbf{x});\xi\right)\right]\right\}$$

$$\text{s.t. } \mathbf{y}^*(\mathbf{x})=\arg\min_{\mathbf{y}\in\mathbb{R}^l}\left\{g(\mathbf{x},\mathbf{y})\triangleq\mathbb{E}_{\zeta\sim\pi_g}[g(\mathbf{x},\mathbf{y};\zeta)]\right\}, \tag{1}$$

where $f(\mathbf{x},\mathbf{y}):\mathbb{R}^u\times\mathbb{R}^l\to\mathbb{R}$ and $g(\mathbf{x},\mathbf{y}):\mathbb{R}^u\times\mathbb{R}^l\to\mathbb{R}$ are UL and LL objectives, respectively. Stochastic bilevel optimization in Problem (1) has gained prominence due to its modeling versatility in machine learning (ML) applications. Classical examples include hyperparameter optimization Shaban et al. (2019); Bao et al. (2021), meta-learning Rajeswaran et al. (2019); Ji et al. (2020), adversarial training Tian et al. (2021); Zhang et al. (2022), reinforcement learning Hong et al. (2020), neural architecture search Lian et al. (2019); Hu et al. (2020), data hyper-cleaning Franceschi et al. (2018); Shaban et al. (2019), and dictionary learning Lecouat et al. (2020a;b). Recently, bilevel optimization has also found its applications in large language models (LLMs) (e.g., the pretraining-finetuning pipeline Li et al. (2024); Wu et al. (2024); Ding et al. (2024), data weighting Shen et al. (2024); Pan et al. (2024), and adversarial attacks on LLMs Jiao et al. (2025)). As a consequence, in the ML research community, a major research effort has been focused on developing efficient algorithms for solving stochastic bilevel optimization problems.

**2) Technical Challenges:** Among all existing methods (see Section 2 for detailed discussion), the approximate implicit differentiation (AID) approach, where the approximate implicit gradient of the UL objective $\ell(\cdot)$ is directly computed using the implicit function theorem Ghadimi & Wang

(2018), is widely adopted due to its ease of implementation. In typical AID algorithms, while the LL variable is updated via standard stochastic gradient descent (SGD), the UL variable is updated using: $\mathbf{x}^+ = \mathbf{x} - \alpha h^f$, where the descent direction $h^f$ (also often referred to as *hypergradient*) is an approximation of the implicit gradient $\nabla\ell(\mathbf{x})$, which can be computed as:

$$h^f \approx \nabla\ell(\mathbf{x}) = \nabla_{\mathbf{x}}f(\mathbf{x}, \mathbf{y}^*(\mathbf{x})) - \underbrace{\nabla^2_{\mathbf{xy}}g(\mathbf{x}, \mathbf{y}^*(\mathbf{x}))\big[\nabla^2_{\mathbf{yy}}g(\mathbf{x}, \mathbf{y}^*(\mathbf{x}))\big]^{-1}\nabla_{\mathbf{y}}f(\mathbf{x}, \mathbf{y}^*(\mathbf{x}))}_{Second-order\ computation\ involving\ HVP\ and\ JVP}. \quad (2)$$

However, due to this hypergradient computation, AID-based bilevel algorithms face two major challenges. First, the hypergradient in Eq. (2) is a function of the optimal solution $\mathbf{y}^*(\mathbf{x})$ of the LL problem, which often requires an iterative method to solve. Thus, obtaining the exact value of $\mathbf{y}^*(\mathbf{x})$ can become computationally expensive, often rendering the algorithms infeasible in practice. This challenge has been intensively studied in literature and addressed to some extent (e.g., the hypergradient is approximated using $\mathbf{y}^+ \approx \mathbf{y}^*(\mathbf{x})$ Ghadimi & Wang (2018); Hong et al. (2020); Chen et al. (2021)). The second challenge, which is the **main focus** of this paper, is that computing the hypergradient requires second-order information. Since Hessian inversion and Jacobian computation in Eq. (2) have computation complexities of $\mathcal{O}(l^3)$ and $\mathcal{O}(ul)$, respectively, where $l$ and $u$ are the dimensions of $\mathbf{x}$- and $\mathbf{y}$-variables in Problem (1), evaluating them makes bilevel algorithms computationally expensive even for moderately sized problems.

To mitigate the second challenge, Hessian-vector products (HVPs) and Jacobian-vector products (JVPs) are commonly used to approximate the Hessian inverse and the Jacobian, respectively. Some modern automatic differentiation tools (e.g., Pearlmutter's trick Pearlmutter (1994)) enable both HVP and JVP computations with $\mathcal{O}(l)$ complexity. Despite using HVPs and JVPs, computing hypergradients is still computationally expensive in many practical scenarios, especially in resource and computation-constrained settings (e.g., edge-based devices with no access to GPUs). It is known that each HVP computation is still at least *two to six times more expensive* than gradient computation using optimized library Jax Bradbury et al. (2018) when performed on CPUs. Moreover, when the model size scales, as particularly shown in LLMs, the computation cost of HVPs and JVPs significantly increases, even on high-performance GPUs. These computation costs can dominate the runtime of bilevel algorithms and severely limit their scalability. What exacerbates the problem is the fact that a single Hessian inverse estimation requires **multiple** HVP computations Ghadimi & Wang (2018); Hong et al. (2020), which can easily multiply the total cost for the desired approximation accuracy. This compounds the overall cost and poses a serious challenge in reducing the computation cost of bilevel optimization algorithms in practical settings. To tackle this challenge, first-order methods (i.e., Hessian/Jacobian-free) have been proposed for bilevel optimization; however, they often exhibit inferior convergence guarantees and degraded practical performance due to the absence of second-order information (see Section 2 for a detailed discussion). This underscores the critical role of second-order information in bilevel optimization and leads to a foundational open problem:

> **(Q)**: Can we design novel bilevel optimization algorithms that require *fewer* second-order information evaluations, while being able to guarantee theoretical convergence performance?

In this paper, we answer the above question by developing a new algorithmic framework called SO-Lazy-BiO (Second-Order Lazy Bilevel Optimization), which allows infrequent second-order information (HVP/JVP) evaluations to alleviate the computational bottleneck when solving stochastic bilevel optimization problems. In our framework, stale second-order information is used for multiple iterations, and only new gradients are computed at each step for computational savings. The intuition behind SO-Lazy-BiO is that, for iterations that are not far from each other, the second-order information remains highly correlated and do not vary significantly. Therefore, stale second-order information may be used to approximate the new value.

However, it is unclear whether SO-Lazy-BiO still converges due to the following factors: 1) the use of stale Hessians (HVPs), 2) the use of stale Jacobians (JVPs), 3) the "multiplicative" structure of JVP, which is coupled with HVP and may amplify the error from lazy evaluations, 4) approximations of the Hessian-inverse, and 5) the coupled hierarchical structure of bilevel problems. Somewhat surprisingly, we prove that, despite the potential errors accumulated by the aforementioned factors, SO-Lazy-BiO not only converges but also attains a convergence rate comparable to that of the SOTA non-lazy bilevel algorithms. To our knowledge, this is the first work that uses *infrequent second-order information computations* for computational savings and still achieves a convergence guarantee in solving stochastic bilevel problems. We summarize our major contributions as follows:

- We develop a new algorithmic framework SO-Lazy-BiO that allows infrequent second-order information computations in stochastic bilevel optimization. Specifically, SO-Lazy-BiO achieves a dual reduction in computational cost: 1) by using single-step SGD to estimate each Hessian-inverse vector product, avoiding the need for multiple HVP computations per approximation; and 2) by incorporating a lazy update strategy that updates second-order information (HVPs/JVPs) only at selected iterations while reusing stale information in the rest of the iterations. These innovations collectively lead to substantial computational savings over existing methods.

- We theoretically establish the performance of SO-Lazy-BiO. Specifically, we show that the proposed lazy approach, which is supposed to perform worse due to stale second-order information, can actually *match* the convergence performance of the SOTA bilevel algorithms. We show that, to achieve an $\epsilon$-stationary point, SO-Lazy-BiO requires $\mathcal{O}(\epsilon^{-2})$ second-order information evaluations, which is fewer than non-lazy bilevel algorithms that incur multiple HVP computations per iteration. Moreover, thanks to the less frequent second-order information evaluations, the *wall-clock time* (i.e., running time) of SO-Lazy-BiO is significantly reduced compared to the SOTA approaches.

- We extensively evaluate the performance of our proposed SO-Lazy-BiO algorithm via numerical experiments, including three highly non-trivial tasks: 1) data weighting for reinforcement learning from human feedback (RLHF) reward model training, 2) data weighting for LLM alignment, and 3) deep hyper-representation. Our results verify that the infrequent evaluations of second-order information lead to considerable computational savings, particularly for large-scale models, even when using high-performance GPUs.

## 2 RELATED WORK

In this section, we provide an overview on three closely related areas: ① AID-based bilevel optimization, ② Hessian/Jacobian-free bilevel optimization, and ③ Other uses of infrequent evaluations. Due to space limitations, we give a summary of other related bilevel optimization methods in Appendix B.

①**AID-Based Bilevel Optimization:** AID-based bilevel optimization has gained popularity due to its ease of implementation. BSA Ghadimi & Wang (2018) provided the first finite-time convergence guarantees for bilevel optimization. The stochastic bilevel algorithms that either use vanilla-SGD updates (e.g., stocBiO in Ji et al. (2021), ALSET in Chen et al. (2021), AmIGO in Arbel & Mairal (2022), and SOBA in Dagréou et al. (2022)) or use momentum-based SGD for updating the UL parameters (e.g., MA-SOBA in Chen et al. (2024)) require $\mathcal{O}\left(\epsilon^{-2}\right)$ for both partial gradient evaluations and second-order information (HVP/JVP) evaluations to reach an $\epsilon$-stationary point. Although these works guarantee finite-time convergence, their practical performance is often limited due to high per-iteration computation costs: they require one or even multiple Hessian (or HVP) evaluations of the LL objective in each iteration to approximate the Hessian inverse, as well as one Jacobian (or JVP) evaluation per iteration to approximate the hypergradient of the UL problem. In this work, we show that both Hessian and Jacobian computations can be *skipped* and *stale* Hessian and Jacobian information computed from previous iterations can be reused without hurting the convergence performance. This significantly reduces computational cost and enables much faster execution.

② **Hessian/Jacobian-Free Bilevel Optimization:** To avoid the expensive Hessian/Jacobian (or HVP/JVP) evaluations, several Hessian/Jacobian-free methods have been proposed. For example, FO-MAML Finn et al. (2017); Nichol et al. (2018) directly ignores the second-order information computation but does not offer any performance guarantee Antoniou et al. (2018); Fallah et al. (2020). Several approaches have also been proposed to replace the LL problem with optimality-based constraints Chen et al. (2023b); Liu et al. (2022a); Shen & Chen (2023). However, these methods mostly focus on deterministic settings rather than stochastic ones. Several zeroth-order methods have been proposed to approximate the hypergradient (e.g., ES-MAML Song et al. (2019), HOZOG Gu et al. (2021), and PZOBO Sow et al. (2022)). However, ES-MAML and HOZOG do not provide any theoretical convergence guarantee, while PZOBO achieves $\mathcal{O}\left(u^2\epsilon^{-2}\right)$ to reach an $\epsilon$-stationary point, where $u$ is the UL problem dimension. Recently, $F^2SA$ and $F^3SA$ (momentum-based version of $F^2SA$) Kwon et al. (2023) have been proposed, which are two first-order methods based on the value-function-based lower-level problem reformulation. To reach an $\epsilon$-stationary point, $F^2SA$ and $F^3SA$ require $\mathcal{O}\left(\epsilon^{-3.5}\right)$ and $\mathcal{O}\left(\epsilon^{-2.5}\right)$ iterations, respectively. The work in Chen et al. (2023a) improves the convergence rate for $F^2SA$, resulting in a rate of $\mathcal{O}\left(\epsilon^{-2}\log(1/\epsilon)\right)$. Unfortunately, achieving this rate requires computation of very large batch gradients (depending on solution accuracy). Compared

to Kwon et al. (2023), our proposed SO-Lazy-BiO algorithm strikes a **good balance** in terms of the use of second-order information: On one hand, we leverage second-order information to maintain good convergence performance; on the other hand, we infrequently use second-order information to significantly reduce the wall-clock time.

③ **Other Uses of Infrequent Evaluations:** Infrequent Hessian evaluations have also been used for speeding up second-order methods for single-level optimization Shamanskii (1967); Adler et al. (2020); Lampariello & Sciandrone (2001); Wang et al. (2006); Fan (2013); Doikov et al. (2023). However, in bilevel optimization, the Hessian information *necessarily* emerges due to the hypergradient computation, rather than as a "second-order" option in single-level optimization. Importantly, the multiplicative structure of the JVP coupled with the HVP in bilevel optimization further increases the complexity of the analysis. Moreover, to the best of our knowledge, we are the first to incorporate infrequent Hessian/Jacobian evaluations into algorithm design to reduce computation cost in bilevel optimization.

## 3 Preliminaries

In this section, we provide some preliminaries for solving Problem (1) and highlight the challenges that arise from using second-order information.

**1) Hessian-Inverse Approximation:** As mentioned earlier, using the implicit function theorem Rudin et al. (1976), the hypergradient of the UL objective $\ell(\cdot)$ can be computed as: $\nabla \ell(\mathbf{x}) = \nabla_{\mathbf{x}} f(\mathbf{x}, \mathbf{y}^*(\mathbf{x})) - \nabla_{\mathbf{xy}}^2 g(\mathbf{x}, \mathbf{y}^*(\mathbf{x}))[\nabla_{\mathbf{yy}}^2 g(\mathbf{x}, \mathbf{y}^*(\mathbf{x}))]^{-1} \nabla_{\mathbf{y}} f(\mathbf{x}, \mathbf{y}^*(\mathbf{x}))$. Instead of computing the Hessian inverse explicitly, there exist different ways to approximate the Hessian inverse or HVPs in bilevel optimization, such as conjugate gradient (CG) Pedregosa (2016), Neumann series Ghadimi & Wang (2018), and SGD methods. In this paper, we use SGD to efficiently estimate the Hessian-inverse vector products (HIVP) ($\left[\nabla_{\mathbf{yy}}^2 g(\mathbf{x}, \mathbf{y}^*(\mathbf{x}))\right]^{-1} \nabla_{\mathbf{y}} f(\mathbf{x}, \mathbf{y}^*(\mathbf{x}))$), which finds the minimizer of a quadratic function by solving a linear system as:

$$\min_{\mathbf{z} \in \mathbb{R}^l} q(\mathbf{x}, \mathbf{y}^*(\mathbf{x}), \mathbf{z}) \triangleq \frac{1}{2} \mathbf{z}^\top \nabla_{\mathbf{yy}}^2 g(\mathbf{x}, \mathbf{y}^*(\mathbf{x})) \mathbf{z} + \mathbf{z}^\top \nabla_{\mathbf{y}} f(\mathbf{x}, \mathbf{y}^*(\mathbf{x})). \tag{3}$$

The admitted unique minimizer $\mathbf{z}^*(\mathbf{x}, \mathbf{y}^*(\mathbf{x}))$ of Eq. (3) can then be utilized to compute the hypergradient estimate as $\nabla \ell(\mathbf{x}) = \nabla_{\mathbf{x}} f(\mathbf{x}, \mathbf{y}^*(\mathbf{x})) + \nabla_{\mathbf{xy}}^2 g(\mathbf{x}, \mathbf{y}^*(\mathbf{x})) \mathbf{z}^*(\mathbf{x}, \mathbf{y}^*(\mathbf{x}))$. Since it is challenging to obtain $\mathbf{y}^*(\mathbf{x})$ and $\mathbf{z}^*(\mathbf{x}, \mathbf{y}^*(\mathbf{x}))$ in closed form, it is natural to consider their approximations. Specifically, let $\bar{\mathbf{y}}$ and $\bar{\mathbf{z}}$ be some approximations of $\mathbf{y}^*(\mathbf{x})$ and $\mathbf{z}^*(\mathbf{x}, \mathbf{y}^*(\mathbf{x}))$, respectively. Then, we have the approximation for $\nabla \ell(\mathbf{x})$ defined as follows:

$$\nabla f(\mathbf{x}, \bar{\mathbf{y}}, \bar{\mathbf{z}}) = \nabla_{\mathbf{x}} f(\mathbf{x}, \bar{\mathbf{y}}) + \nabla_{\mathbf{xy}}^2 g(\mathbf{x}, \bar{\mathbf{y}}) \bar{\mathbf{z}}. \tag{4}$$

Since Problem (1) can potentially be a large-scale stochastic optimization problem, computing a full gradient approximation in Eq. (4) can be computationally expensive. To address this challenge, a common approach for evaluating Eq. (4) is to build a stochastic gradient estimator. Define stochastic approximations as $f\left(\mathbf{x}, \mathbf{y}; \mathcal{D}^f\right) \triangleq \frac{1}{|\mathcal{D}^f|} \sum_{\xi \in \mathcal{D}^f} f(\mathbf{x}, \mathbf{y}; \xi)$ and $g\left(\mathbf{x}, \mathbf{y}; \mathcal{D}^g\right) \triangleq \frac{1}{|\mathcal{D}^g|} \sum_{\zeta \in \mathcal{D}^g} g(\mathbf{x}, \mathbf{y}; \zeta)$, where $\mathcal{D}^f$ and $\mathcal{D}^g$ are the batches of independent and identically distributed samples with sizes $\left|\mathcal{D}^f\right| \geq 1$ and $|\mathcal{D}^g| \geq 1$, respectively. Then, a stochastic estimator of Eq. (4) can be computed as:

$$\nabla f(\mathbf{x}, \mathbf{y}, \mathbf{z}; \bar{\mathcal{D}}^f) = \nabla_{\mathbf{x}} f\left(\mathbf{x}, \mathbf{y}; \mathcal{D}^{f_x}\right) + \nabla_{\mathbf{xy}}^2 g\left(\mathbf{x}, \mathbf{y}; \mathcal{D}^{g_{xy}}\right) \mathbf{z},$$

where $\bar{\mathcal{D}}^f \triangleq \left\{\mathcal{D}^{f_x}, \mathcal{D}^{g_{xy}}\right\}$. Here, for simplicity, we slightly abuse the notations $\bar{\mathbf{y}}$ and $\bar{\mathbf{z}}$ as $\mathbf{y}$ and $\mathbf{z}$ in the above equation and the rest of the paper, as long as there is no confusion from the context.

**2) Challenges due to Second-Order Information:** Although HIVP can be relatively more efficiently approximated by solving a quadratic optimization problem and the Jacobian can be evaluated via JVP, several challenges remain: *i)* The approximation error in $\mathbf{y}^*(\mathbf{x})$ propagates and exacerbates the error in approximating $\mathbf{z}^*(\mathbf{x}, \mathbf{y}^*(\mathbf{x}))$ due to the dependency of the latter on the former. *ii)* While HVPs and JVPs have been introduced to reduce complexity, their practical implementation still demands considerable computational resources, particularly in resource-constrained environments or when deploying large-scale models such as LLMs. *iii)* Achieving an accurate approximation of HIVP requires multiple iterations to solve Problem (3), which further increases computational cost, especially due to repeated HVP evaluations.

## 4    THE SO-Lazy-BiO ALGORITHM

In this section, we propose SO-Lazy-BiO to solve the bilevel optimization problem in Eq. (1). Our goal is to reduce the computation of second-order information (HVPs/JVPs), and the key idea is to update the second-order information periodically on a subset of the entire training iterations while using stale second-order information in the remaining iterations.

We illustrate SO-Lazy-BiO in Algorithm 1. Notably, SO-Lazy-BiO uses a *single-loop* structure and constructs the iterates of $\mathbf{x}_t$, $\mathbf{y}_t$ and $\mathbf{z}_t$, where the iteration counter $t$ runs from 0 to $T-1$. Note that $\mathbf{y}_t$ and $\mathbf{z}_t$ keep track of the quantities $\mathbf{y}^*(\mathbf{x}_t)$ and $\mathbf{z}^*(\mathbf{x}_t, \mathbf{y}^*(\mathbf{x}_t))$. The algorithm updates $\mathbf{x}_t$ and $\mathbf{y}_t$ using the stochastic gradient estimators $h_t^f$ and $h_t^g$ defined as:

$$h_t^f = \nabla_{\mathbf{x}} f\left(\mathbf{x}_t, \mathbf{y}_t; \mathcal{D}_t^{f_x}\right) + \mathbf{v}_t, \qquad (5)$$

$$h_t^g = \nabla_{\mathbf{y}} g\left(\mathbf{x}_t, \mathbf{y}_t; \mathcal{D}_t^g\right), \qquad (6)$$

where $\mathbf{v}_t$ denotes the JVP and it is updated *lazily* every $N$ iterations (**Option I** in SO-Lazy-BiO):

$$\mathbf{v}_t = \nabla_{\mathbf{x}\mathbf{y}}^2 g\left(\mathbf{x}_t, \mathbf{y}_t; \mathcal{D}_t^{g_{xy}}\right) \mathbf{z}_t. \qquad (7)$$

Every $N$ iterations, variable $\mathbf{z}_t$ in (7) is updated *lazily* using a stochastic gradient estimator $h_t^q$:

$$h_t^q = \nabla_{\mathbf{y}\mathbf{y}}^2 g(\mathbf{x}_t, \mathbf{y}_t; \mathcal{D}_t^{g_{yy}})\mathbf{z}_t + \nabla_{\mathbf{y}} f(\mathbf{x}_t, \mathbf{y}_t; \mathcal{D}_t^{f_y}). (8)$$

Note that, compared to $h_t^f$ and $h_t^g$, only $h_t^q$ and $\mathbf{v}_t$ contain the HVP and JVP, respectively, and are computed *infrequently* in a lazy fashion after every $N$ iterations. Since $\mathbf{z}_t$ is the HVP estimator, reducing the frequency of JVP computations also inherently reduces the frequency of HVP computations. Therefore, the reductions in computational cost for JVPs and HVPs are intrinsically coupled. In addition, $N$ needs to be appropriately chosen with a tolerable approximation error. If $N$ is too large, the error of the second-order information

---

**Algorithm 1** The SO-Lazy-BiO Algorithm.

**Input:** Initial parameters $\mathbf{x}_0$, $\mathbf{y}_0$, $\mathbf{z}_0$, stepsizes $\{\alpha_t, \beta_t, \gamma_t\}_{t=0}^{T-1}$, momentum coefficient $\{\mu_t\}_{t=0}^{T-1}$, and flag Lazy_JVP $\in$ {**True, False**}
**for** $t = 0$ to $T - 1$ **do**
    **if** $t \bmod N = 0$ **then**
        Sample data batches $\mathcal{D}_t^{g_{yy}}$, and $\mathcal{D}_t^{f_y}$
        Compute the gradient estimate $h_t^q$ using (8)
        Update $\mathbf{z}_{t+1} = \mathbf{z}_t - \gamma_t h_t^q$
    **else**
        $\mathbf{z}_{t+1} = \mathbf{z}_t$       ▷ **Reuse stale HVP**
    **end if**
    **if** Lazy_JVP == **True then**
        <- - - - - - **Option I: Lazy JVP** - - - - - ->
        **if** $t \bmod N = 0$ **then**
            Sample data batches $\mathcal{D}_t^{g_{xy}}$
            Compute the JVP using (7)
        **else**
            $\mathbf{v}_t = \mathbf{v}_{t-1}$    ▷ **Reuse stale JVP**
        **end if**
    **else**
        <- - - - **Option II: Regular JVP** - - - ->
        Sample data batches $\mathcal{D}_t^{g_{xy}}$
        Compute the JVP using (7)
    **end if**
    Sample data batches $\mathcal{D}_t^g$ and $\mathcal{D}_t^{f_x}$
    Compute the gradient estimate $h_t^g$ using (6)
    Update $\mathbf{y}_{t+1} = \mathbf{y}_t - \beta_t h_t^g$
    Compute the gradient estimate $h_t^f$ using (5)
    Compute the momentum-based $\bar{h}_t^f$ using (9)
    Update $\mathbf{x}_{t+1} = \mathbf{x}_t - \alpha_t \bar{h}_t^f$
**end for**

---

approximation could increase too dramatically, thus decaying the performance of SO-Lazy-BiO.

Before updating the UL parameter $\mathbf{x}$, we integrate a standard momentum approach into the update step (see Section 5.3 for a discussion of its necessity), defined as follows:

$$\bar{h}_{t+1}^f = \mu_t h_t^f + (1 - \mu_t)\bar{h}_t^f, \qquad (9)$$

where $\mu_t \in [0, 1]$ is the momentum coefficient. Setting $\mu_t = 1$ recovers the standard SGD update.

To balance the *trade-off* between reducing the overall computational cost in bilevel optimization and controlling the error introduced by stale HVP and JVP, we also consider a special case of the SO-Lazy-BiO framework, which is shown as **Option II** in SO-Lazy-BiO and referred to as SO-Lazy-BiO-II. In SO-Lazy-BiO-II, only the computation of $h_t^q$, which involves the HVP, is performed *infrequently* once every $N$ iterations, while the JVP is computed at every iteration. Although SO-Lazy-BiO-II contains additional computation from the non-lazy JVP evaluations, this reduced laziness may actually improve overall implementation wall-clock time compared to **Option I** in SO-Lazy-BiO, due to a trade-off between per-iteration cost and overall convergence speed.

It is worth noting that while most existing bilevel algorithms compute only one single JVP per iteration, they typically require multiple HVP computations in each iteration Arbel & Mairal (2022); Ji et al. (2021), even in some single-loop bilevel algorithms (e.g., SUSTAIN Khanduri et al. (2021b), TTSA Hong et al. (2020), BSA Ghadimi & Wang (2018), and ALSET Chen et al. (2021)). In contrast, our proposed SO-Lazy-BiO achieves a **two-fold reduction** in computational costs: **(1)**

A single-step SGD to estimate each Hessian-inverse vector product, thereby eliminating the need for multiple HVP computations per approximation; and **(2)** A lazy update strategy that evaluates second-order information (HVPs/JVPs) infrequently. Combined together, these two new algorithmic techniques lead to significant overall computational savings and reduced implementation *wall-clock time* compared to existing non-lazy methods.

## 5 THEORETICAL PERFORMANCE ANALYSIS

In this section, we first focus on conduct the theoretical convergence analysis for the **most-lazy scenario** within the SO-Lazy-BiO framework, specifically **Option I** (referred to as SO-Lazy-BiO-I), for solving the bilevel optimization problem in (1). We relegate the theoretical convergence analysis of **Option II** in the SO-Lazy-BiO framework to Appendix G, since the proofs for the two options are similar and **Option II** can be viewed as a special case of **Option I**, where no errors are introduced by JVP updates. We note that both **Option I** and **Option II** share the same convergence guarantees. Note that, although SO-Lazy-BiO executes faster per iteration, we have a *noisier* hypergradient due to the use of stale second-order information, particularly in SO-Lazy-BiO-I where both stale Hessian and stale Jacobian evaluations are used. As a result, it remains unclear whether SO-Lazy-BiO-I can converge and, if yes, what theoretical convergence rate (i.e., iteration complexity) it will achieve. Intuitively, due to the lazy second-order information updates, one can expect that the convergence rate of SO-Lazy-BiO-I cannot outperform its non-lazy counterpart. Surprisingly, in this paper, we show that SO-Lazy-BiO-I achieves a convergence rate comparable to that of its non-lazy counterpart. This, together with significantly fewer HVP/JVP computations and much lower per-iteration wall-clock time, implies that SO-Lazy-BiO-I will enjoy a much faster speed in terms of wall-clock time. This will also be verified by our experiments in Section 6.

We note that the convergence analysis for SO-Lazy-BiO-I is highly non-trivial due to the following **technical challenges**: **1)** The use of lazy Hessian and Jacobian evaluations increases the error of the stochastic gradient estimator $h_t^f$ for the upper-level function; **2)** The *"multiplicative" structure* of $\mathbf{v}_t$ in SO-Lazy-BiO-I, which couples the JVP with the HVP, significantly complicates the error analysis introduced by the lazy computations; **3)** Due to the hierarchical and coupled structure of bilevel optimization problems, the error resulting from the stochastic gradient estimator $h_t^f$ with stale Hessian and Jacobian information further propagates to and increases the approximation error of $\mathbf{y}^*(\mathbf{x})$ and the approximation error of $\mathbf{z}^*(\mathbf{x}, \mathbf{y}^*(\mathbf{x}))$. What is even worse is that the approximation error in $\mathbf{y}^*(\mathbf{x})$ further exacerbates the error in $\mathbf{z}^*(\mathbf{x}, \mathbf{y}^*(\mathbf{x}))$, since $\mathbf{z}^*(\mathbf{x}, \mathbf{y}^*(\mathbf{x}))$ is also associated with $\mathbf{y}^*(\mathbf{x})$. All the complex couplings of *laziness-induced errors* above and the complications associated with these approximation errors are **unseen** in bilevel optimization algorithm analysis, which significantly increases the difficulty of analyzing the convergence of SO-Lazy-BiO.

### 5.1 ASSUMPTIONS

We first state a set of assumptions that are needed to establish the convergence of SO-Lazy-BiO-I:

**Assumption 5.1** (UL Objective). $f(\mathbf{x}, \mathbf{y})$ satisfies: *1)* The map $\mathbf{y} \mapsto \nabla_{\mathbf{x}} f(\mathbf{x}, \mathbf{y})$ is Lipschitz $\forall \mathbf{x} \in \mathbb{R}^u$ with $L_{f_x} \geq 0$, and the map $(\mathbf{x}, \mathbf{y}) \mapsto \nabla_{\mathbf{y}} f(\mathbf{x}, \mathbf{y})$ is Lipschitz with $L_{f_y} \geq 0$. *2)* For all $\mathbf{x} \in \mathbb{R}^u$, we have $\|\nabla_{\mathbf{y}} f(\mathbf{x}, \mathbf{y}^*(\mathbf{x}))\| \leq B_{f_y}$ for some $B_{f_y} \geq 0$.

**Assumption 5.2** (LL Objective). $g(\mathbf{x}, \mathbf{y})$ satisfies: *1)* For any $\mathbf{x} \in \mathbb{R}^u$, $\mathbf{y} \mapsto g(\mathbf{x}, \mathbf{y})$ is $\mu_g$-strongly convex for some $\mu_g > 0$. *2)* The map $\mathbf{y} \mapsto \nabla_{\mathbf{y}} g(\mathbf{x}, \mathbf{y})$ is Lipschitz $\forall \mathbf{x} \in \mathbb{R}^u$ with $L_g \geq 0$, and the maps $(\mathbf{x}, \mathbf{y}) \mapsto \nabla_{\mathbf{xy}}^2 g(\mathbf{x}, \mathbf{y})$ and $(\mathbf{x}, \mathbf{y}) \mapsto \nabla_{\mathbf{yy}}^2 g(\mathbf{x}, \mathbf{y})$ are Lipschitz with $L_{g_{xy}} \geq 0$ and $L_{g_{yy}} \geq 0$, resp. *3)* For all $(\mathbf{x}, \mathbf{y}) \in \mathbb{R}^u \times \mathbb{R}^l$, we have $\|\nabla_{\mathbf{xy}}^2 g(\mathbf{x}, \mathbf{y})\| \leq B_{g_{xy}}$ for some $B_{g_{xy}} > 0$.

Note that, aside from the boundedness assumption on $\nabla_{\mathbf{y}} f(\mathbf{x}, \mathbf{y}^*(\mathbf{x}))$, all other assumptions are standard in the analysis of bilevel optimization problems (e.g., Ghadimi & Wang (2018); Hong et al. (2020); Khanduri et al. (2021b); Liu et al. (2022b); Qiu et al. (2022)). Our analysis assumes the boundedness of $\nabla_{\mathbf{y}} f(\mathbf{x}, \mathbf{y}^*(\mathbf{x}))$, which differs from the more commonly used assumption on $\nabla_{\mathbf{y}} f(\mathbf{x}, \mathbf{y})$ in previous works and is comparatively more relaxed.

Next, for the stochastic gradient estimators $\nabla f(\mathbf{x}, \mathbf{y}, \mathbf{z}; \mathcal{D}^{f_x}, \mathcal{D}^{g_{xy}})$ and $\nabla_{\mathbf{y}} g(\mathbf{x}, \mathbf{y}; \mathcal{D}^{g_y})$, we make the following typical assumption in stochastic optimization analysis:

**Assumption 5.3** (Stochastic Gradients). For any $(\mathbf{x}, \mathbf{y}) \in \mathbb{R}^u \times \mathbb{R}^l$ and data batch $\mathcal{D}^{f_x}$, $\mathcal{D}^{f_y}$, $\mathcal{D}^{g_y}$, $\mathcal{D}^{g_{xy}}$ and $\mathcal{D}^{g_{yy}}$, the gradient estimates $\nabla_{\mathbf{x}} f(\mathbf{x}, \mathbf{y}; \mathcal{D}^{f_x})$, $\nabla_{\mathbf{y}} f(\mathbf{x}, \mathbf{y}; \mathcal{D}^{f_y})$, $\nabla_{\mathbf{y}} g(\mathbf{x}, \mathbf{y}; \mathcal{D}^{g_y})$,

$\nabla^2_{\mathbf{xy}} g(\mathbf{x}, \mathbf{y}; \mathcal{D}^{g_{xy}})$ and $\nabla^2_{\mathbf{yy}} g(\mathbf{x}, \mathbf{y}; \mathcal{D}^{g_{yy}})$ are unbiased and have bounded variances:

$$\mathbb{E}[\|\nabla_{\mathbf{x}} f(\mathbf{x}, \mathbf{y}; \mathcal{D}^{f_x}) - \nabla_{\mathbf{y}} f(\mathbf{x}, \mathbf{y})\|^2] \leq \sigma^2_{f_x}, \quad \mathbb{E}[\|\nabla_{\mathbf{y}} f(\mathbf{x}, \mathbf{y}; \mathcal{D}^{f_y}) - \nabla_{\mathbf{y}} f(\mathbf{x}, \mathbf{y})\|^2] \leq \sigma^2_{f_y},$$

$$\mathbb{E}[\|\nabla_{\mathbf{y}} g(\mathbf{x}, \mathbf{y}; \mathcal{D}^{g_y}) - \nabla_{\mathbf{y}} g(\mathbf{x}, \mathbf{y})\|^2] \leq \sigma^2_{g_y}, \quad \mathbb{E}[\|\nabla^2_{\mathbf{xy}} g(\mathbf{x}, \mathbf{y}; \mathcal{D}^{g_{xy}}) - \nabla^2_{\mathbf{xy}} g(\mathbf{x}, \mathbf{y})\|^2] \leq \sigma^2_{g_{xy}},$$

$$\mathbb{E}[\|\nabla^2_{\mathbf{yy}} g(\mathbf{x}, \mathbf{y}; \mathcal{D}^{g_{yy}}) - \nabla^2_{\mathbf{yy}} g(\mathbf{x}, \mathbf{y})\|^2] \leq \sigma^2_{g_{yy}}.$$

Lastly, we define $\epsilon$-stationarity as a performance measure for an algorithm for solving Problem (1):

**Definition 5.4** ($\epsilon$-Stationarity). $\mathbf{x}$ is an $\epsilon$-stationary solution if $\mathbb{E}[\|\nabla\ell(\mathbf{x})\|^2] \leq \epsilon$, where $\mathbf{x}$ is the output of a stochastic algorithm, and the expectation is taken over all randomness of the algorithm.

## 5.2 MAIN CONVERGENCE RESULTS

We now state the main convergence result of the "most-lazy" scenario of the proposed SO-Lazy-BiO framework, i.e., SO-Lazy-BiO-I, for non-convex $\ell(\mathbf{x})$ in Theorem 5.5:

**Theorem 5.5** (Convergence Rate of SO-Lazy-BiO-I). *Under Assumptions 5.1–5.3, choose step-sizes* $\alpha_t = \alpha = \mathcal{O}((\sqrt{NT})^{-1})$, $\beta_t = \beta = \mathcal{O}((\sqrt{NT})^{-1})$, $\gamma_t = \gamma = \mathcal{O}(\sqrt{N}(\sqrt{T})^{-1})$, *and the momentum coefficient as* $\mu_t = \mu = \mathcal{O}((\sqrt{NT})^{-1})$ *for all* $t = 0, \ldots, T-1$. *Then, the iterates generated by* SO-Lazy-BiO-I *satisfy:*

$$\frac{1}{T} \sum_{t=0}^{T-1} \mathbb{E}\left[\|\nabla\ell(\mathbf{x}_t)\|^2\right] = \mathcal{O}\left(\frac{\sqrt{N}\Delta_0}{\sqrt{T}} + \frac{\sigma^2_{g_y}}{\sqrt{NT}} + \frac{\sqrt{N}}{\sqrt{T}}\sigma^2_{g_{yy}} + \frac{\sqrt{N}}{\sqrt{T}}\sigma^2_{f_y} + \frac{\sigma^2_{g_{xy}}}{\sqrt{NT}} + \frac{\sigma^2_{f_x}}{\sqrt{NT}}\right),$$

*where* $\Delta_0 = (\ell(\mathbf{x}_0) - \ell^*) + \|\mathbf{y}_0 - \mathbf{y}^*(\mathbf{x}_0)\|^2 + \|\mathbf{z}_0 - \mathbf{z}^*(\mathbf{x}_0, \mathbf{y}^*(\mathbf{x}_0))\|^2$.

The proof of Theorem 5.5 is included in Appendix E. Theorem 5.5 establishes the convergence of SO-Lazy-BiO-I under the most general and most lazy settings, where the function $\ell(\cdot)$ is non-convex and both HVP and JVP are stale. The result characterizes the effect of different parameters on the convergence of SO-Lazy-BiO-I. Specifically, as $N$ increases, the performance in terms of iteration complexity of SO-Lazy-BiO-I degrades. This is unsurprising since more stale second-order information is expected to slow the convergence. Interestingly, under an appropriate $N$-value, the $N$-dependent slowdown effect in SO-Lazy-BiO-I can be offset by skipping second-order information computations, allowing SO-Lazy-BiO-I to run even faster than non-lazy approaches in terms of wall-clock time. The computation complexity of SO-Lazy-BiO-I follows immediately from Theorem 5.5:

**Corollary 5.6** (Computation Complexity of SO-Lazy-BiO-I). *Under the setting of Theorem 5.5, choose the batch size as* $\mathcal{O}(1)$. *Then,* SO-Lazy-BiO-I *requires* $\mathcal{O}(N\epsilon^{-2})$ *partial gradient evaluations and* $\mathcal{O}(\epsilon^{-2})$ *second-order information evaluations to reach an* $\epsilon$-stationary solution.

We note that the computation complexity of second-order information evaluations in Corollary 5.6 is lower than that of standard non-lazy bilevel algorithms, which require multiple HVP computations in each iteration, such as AmIGO Arbel & Mairal (2022), stocBiO Ji et al. (2021), ALSET Chen et al. (2021), and BSA Ghadimi & Wang (2018). Specifically, these algorithms incur a total of $\mathcal{O}(K\epsilon^{-2})$ HVP computations, where $K$ denotes the number of HVP evaluations per iteration, whereas our proposed SO-Lazy-BiO-I algorithm requires only $\mathcal{O}(\epsilon^{-2})$ HVP computations. In addition, our proposed SO-Lazy-BiO-I converges significantly faster than standard non-lazy bilevel algorithms in terms of wall-clock time, further demonstrating the effectiveness of SO-Lazy-BiO-I.

## 5.3 PERFORMANCE WITHOUT THE MOMENTUM

To show the benefit of incorporating the momentum information in the updates of the upper-level parameter $\mathbf{x}$, we conduct a theoretical analysis of the vanilla SGD-based SO-Lazy-BiO-I, where the momentum coefficient is set to $\mu_t = 1$. We refer to this variant of SO-Lazy-BiO as SO-Lazy-BiO-SGD.

For a fair comparison, the convergence analysis is conducted under the same assumptions as SO-Lazy-BiO-I, and the main convergence result for SO-Lazy-BiO-SGD is presented in Theorem 5.7.

**Theorem 5.7** (Convergence Rate of SO-Lazy-BiO-SGD). *Under Assumptions 5.1–5.3, choose constant step-sizes* $\alpha_t = \alpha = \mathcal{O}(1)$, $\beta_t = \beta = \mathcal{O}(1)$, *and* $\gamma_t = \gamma = \mathcal{O}(N)$ *for all* $t = 0, 1, \ldots, T-1$. *Then, the iterates generated by* SO-Lazy-BiO-SGD *satisfy:*

$$\frac{1}{T} \sum_{t=0}^{T-1} \mathbb{E}\left[\|\nabla\ell(\mathbf{x}_t)\|^2\right] = \mathcal{O}\left(\frac{\Delta_0}{T} + N\sigma^2_{g_y} + N\sigma^2_{g_{yy}} + N\sigma^2_{f_y} + \sigma^2_{g_{xy}} + \sigma^2_{f_x}\right),$$

*where* $\Delta_0 = (\ell(\mathbf{x}_0) - \ell^*) + \|\mathbf{y}_0 - \mathbf{y}^*(\mathbf{x}_0)\|^2 + \|\mathbf{z}_0 - \mathbf{z}^*(\mathbf{x}_0, \mathbf{y}^*(\mathbf{x}_0))\|^2$.

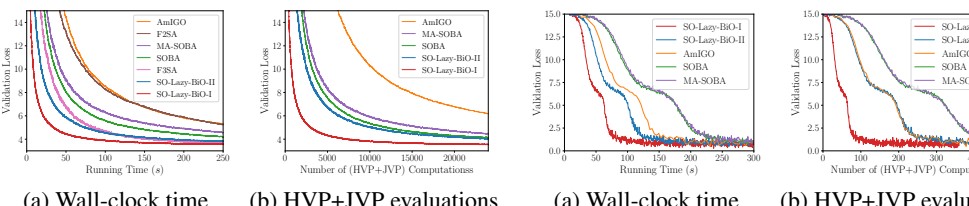

(a) Wall-clock time    (b) HVP+JVP evaluations      (a) Wall-clock time    (b) HVP+JVP evaluations

Figure 1: Validation loss comparison for data weighting in RLHF reward model training.

Figure 2: Validation loss comparison for data weighting in LLM alignment.

The proof of Theorem 5.7 is provided in Appendix F. The computation complexity of SO-Lazy-BiO-SGD immediately follows from Theorem 5.7:

**Corollary 5.8** (Computation Complexity of SO-Lazy-BiO-SGD). *Under the setting of Theorem 5.7, choose* $\left|\mathcal{D}^{f_x}\right|, \left|\mathcal{D}^{g_{xy}}\right| = \Theta\left(\epsilon^{-1}\right)$, *and* $\left|\mathcal{D}^{g_y}\right|, \left|\mathcal{D}^{f_y}\right|, \left|\mathcal{D}^{g_{yy}}\right| = \Theta\left(N\epsilon^{-1}\right)$. *Then, SO-Lazy-BiO requires* $\mathcal{O}(N\epsilon^{-2})$ *partial gradient evaluations and* $\mathcal{O}(\epsilon^{-2})$ *second-order information evaluations to reach an* $\epsilon$-*stationary point.*

Both SO-Lazy-BiO-I and SO-Lazy-BiO-SGD exhibit the same computation complexity in terms of partial gradient evaluations and second-order information evaluations, with the latter being lower than that of non-lazy bilevel algorithms that perform multiple HVP evaluations per iteration. This confirms the effectiveness of our proposed framework, as both SO-Lazy-BiO-I and SO-Lazy-BiO-SGD leverage lazy second-order information evaluations. Moreover, our proposed framework SO-Lazy-BiO achieves substantially faster convergence in terms of wall-clock time compared to standard non-lazy bilevel algorithms, further validating the efficiency of SO-Lazy-BiO. However, unlike SO-Lazy-BiO-I, which requires a batch size of $\mathcal{O}(1)$, SO-Lazy-BiO-SGD requires significantly larger batch sizes. This highlights the *benefits* of incorporating momentum into the updates of the upper-level parameter $\mathbf{x}$.

# 6 NUMERICAL EXPERIMENTS

In this section, we verify the performance of SO-Lazy-BiO with three complex bilevel optimization tasks: 1) data weighting for RLHF Ouyang et al. (2022) reward model training; 2) data weighting for LLM alignment; and 3) deep hyper-representation with ResNet network. Due to space limitations, some experimental details and additional results are relegated to Appendix C.

We compare our proposed SO-Lazy-BiO with standard second-order stochastic bilevel algorithms: AmIGO Arbel & Mairal (2022), SOBA Dagréou et al. (2022), and MA-SOBA Chen et al. (2024). Especially for Tasks 1 and 3, we also compare SO-Lazy-BiO with two fully first-order (Hessian/Jacobian-free) stochastic bilevel algorithms F$^2$SA Kwon et al. (2023) and F$^3$SA Kwon et al. (2023) to assess the importance of second-order information during training.

**Task 1) Data weighting for RLHF reward model training:** The goal of data weighting is to determine optimal sampling weights on training data that maximize validation performance. We train the reward model on the HelpSteer dataset Wang et al. (2023), where each prompt-response pair is labeled according to different score criteria.

As shown in Fig. 1a, despite having more errors due to infrequent HVP and/or JVP computations, SO-Lazy-BiO-I converges the *fastest* in terms of wall-clock time among all algorithms, including two fully first-order algorithms F$^2$SA and F$^3$SA, and achieves the *lowest* validation loss, which corresponds to our UL objective, within the same runtime. This is attributed to infrequent second-order computations of SO-Lazy-BiO-I, which allows the shortest per-iteration time and consequently the ability to perform more updates for a given runtime. In addition, leveraging second-order information introduces fewer errors compared to fully first-order algorithms.

Also, Fig. 1b shows that the convergence speed with respect to the cumulative number of HVP and JVP evaluations for SO-Lazy-BiO-I is much *faster* compared to all other algorithms. Table 1 also demonstrates that, to reach the same validation loss, both SO-Lazy-BiO-I and SO-Lazy-BiO-II require 600 HVP computations at most, which is at least $3.72\times$ fewer than those required by other non-lazy methods. In addition, compared to SO-Lazy-BiO-II, the infrequent JVP design in SO-Lazy-BiO-I reduces JVP computations by a factor of $5$, resulting in further reduced running time.

**Task 2) Data weighting for LLM alignment:** In this task, we aim to determine weights on dataset used during LLM alignment. We use Llama-3.2-1B-Instruct Meta (2024) as the base model and align it on HH-RLHF dataset Bai et al. (2022), where each sample is labeled as either *chosen* or *rejected*.

In Figs. 2a and 2b, we observe the same performance trend as in Task 1, with SO-Lazy-BiO-I converging the *fastest*. The performance gaps across the algorithms, however, become more noticeable than in Task 1. This is because, as the LLM model size increases, the computational savings from infrequent second-order evaluations become more significant. These results verify that our algorithm provides greater computational advantages for large-scale problems.

**Task 3) Deep hyper-representation with ResNet network:** We conduct experiments on a deep hyper-representation task Sow et al. (2022) with the ResNet-20 model He et al. (2016) on CIFAR-10 dataset Krizhevsky et al. (2009), which aims to classify CIFAR-10 images.

As shown in Fig. 3a, the validation loss for both SO-Lazy-BiO-I and SO-Lazy-BiO-II is *comparable* to those of second-order baseline algorithms, and is notably *lower* than those of the fully first-order baseline methods. The superior performance of SO-Lazy-BiO-I, SO-Lazy-BiO-II, and other second-order methods compared to the "Hessian/Jacobian-free" F$^2$SA and F$^3$SA highlights the *benefits* of Hessian/Jacobian information in bilevel optimization. Without them, both the convergence speed and validation loss would degrade. Moreover, SO-Lazy-BiO-II converges *fastest* in terms of wall-clock time among all baselines. Fig. 3b demonstrates that SO-Lazy-BiO-II achieves the *fastest* convergence among all baselines in terms of the cumulative number of HVP and JVP computations. Furthermore, as shown in Table 1, to reach the same

Table 1: Number of HVP/JVP computations and runtime required by various algorithms to achieve the same validation loss (averaged over 5 repetitions).

|  | ALGORITHM | # OF HVP | # OF JVP | RUNTIME (S) |
|---|---|---|---|---|
| TASK 1 | AMIGO | 12,195 | 12,195 | 110.28 |
|  | SOBA | 2,231 | 2,650 | 49.92 |
|  | MA-SOBA | 2,402 | 2,402 | 61.07 |
|  | **SO-Lazy-BiO-I** | **526** | **526** | **26.64** |
|  | **SO-Lazy-BiO-II** | **600** | **3,000** | **11.99** |
| TASK 2 | AMIGO | 170 | 34 | 131.71 |
|  | SOBA | 176 | 176 | 192.49 |
|  | MA-SOBA | 176 | 176 | 194.42 |
|  | **SO-Lazy-BiO-I** | **35** | **35** | **66.89** |
|  | **SO-Lazy-BiO-II** | **35** | **173** | **106.85** |
| TASK 3 | AMIGO | 518 | 259 | 129.04 |
|  | SOBA | 501 | 501 | 163.86 |
|  | MA-SOBA | 471 | 471 | 153.54 |
|  | **SO-Lazy-BiO-I** | **353** | **353** | **188.73** |
|  | **SO-Lazy-BiO-II** | **116** | **232** | **63.93** |

validation loss, SO-Lazy-BiO-II requires the *fewest* HVP computations and JVP computations. This significantly reduces computational costs and wall-clock running time.

It is not surprising that SO-Lazy-BiO-I exhibits longer wall-clock time, as infrequent JVP evaluations introduce more error compared to SO-Lazy-BiO-II, potentially requiring more iterations to reach convergence. As a result, the cumulative number of HVP and JVP computations increases, as shown in Fig. 3b. Nevertheless, as demonstrated in Table 1, despite requiring more iterations, SO-Lazy-BiO-I still requires *fewer* HVP evaluations to reach the same validation loss compared to the non-lazy algorithms.

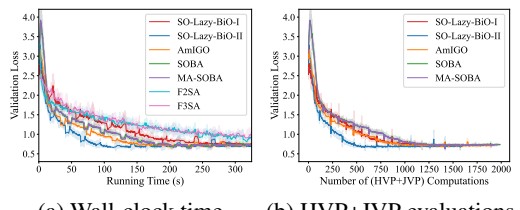

(a) Wall-clock time     (b) HVP+JVP evaluations

Figure 3: Validation loss for deep hyper-representation.

# 7 CONCLUSION

In this paper, we proposed the SO-Lazy-BiO algorithmic framework for solving bilevel optimization problems. Compared to existing works, SO-Lazy-BiO reduces the evaluations of second-order information (Hessian/Jacobian-vector products) by updating them periodically and less frequently. Although SO-Lazy-BiO uses stale second-order information that introduce additional errors, our theoretical analysis demonstrated that SO-Lazy-BiO not only surprisingly enjoys convergence rate guarantees comparable to those of state-of-the-art (SOTA) non-lazy bilevel methods, but also achieves a much faster wall-clock time performance. Specifically, to reach an $\epsilon$-stationary point, SO-Lazy-BiO requires $\mathcal{O}(\epsilon^{-2})$ second-order information evaluations, which is fewer than those required by non-lazy bilevel algorithms that perform multiple HVP evaluations per iteration. We validated the effectiveness and efficiency of our proposed SO-Lazy-BiO through experiments on multiple bilevel optimization tasks.

## ETHICS STATEMENT

We confirm that the ICLR Code of Ethics has been reviewed and that this work fully adheres to it. It involves no human subjects, sensitive data, or foreseeable risks. There are no ethical, legal, or conflict-of-interest concerns.

## REPRODUCIBILITY STATEMENT

We confirm the reproducibility of this work. Specifically, for the theoretical results, we state all assumptions in Section 5 and provide detailed proofs in Appendix E–G. For the experimental results, we include the source code in the supplementary material and describe implementation details in Appendix C.

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

## A    THE USE OF LARGE LANGUAGE MODELS (LLMs)

LLMs were used to assist with grammar correction and language polishing during the writing process. They did not contribute to research ideation.

## B    ADDITIONAL RELATED WORK

**Bilevel Optimization:** The history of bilevel optimization dates back to 1973 Bracken & McGill (1973). Some early attempts for solving bilevel problems include: value function Liu et al. (2021); Sinha et al. (2018); Zemkoho & Zhou (2021), Karush–Kuhn–Tucker conditions based reformulations Allende & Still (2013); Sinha et al. (2019); Zemkoho & Zhou (2021), penalty function White & Anandalingam (1993); Anandalingam & White (1990); Wan et al. (2014), approximate descent Falk & Liu (1995); Vicente et al. (1994), and trust region methods Dempe & Bard (2001); El-Sobky & Abo-Elnaga (2018). Among these approaches, approximate descent methods have gained prominence recently because of their ease of implementation as well as strong theoretical and empirical performance in many machine learning applications. Two standard descent-based approaches to tackle problems of form (1) are iterative differentiation (ITD) Domke (2012); Maclaurin et al. (2015); Franceschi et al. (2017; 2018); Shaban et al. (2019); Grazzi et al. (2020); MacKay et al. (2019) and approximate implicit differentiation (AID) Domke (2012); Pedregosa (2016); Liao et al. (2018); Ghadimi & Wang (2018); Grazzi et al. (2020); Lorraine et al. (2020); Gould et al. (2016); Ji & Liang (2021); MacKay et al. (2019); Khanduri et al. (2021a); Hong et al. (2020). The basic idea of ITD is to obtain an approximate hypergradient of the loss function $\ell(\mathbf{x})$ in Eq. (1) by differentiating the unrolled iterates of the LL problem. Consequently, ITD-based approaches need to store all the LL iterates in the memory Shaban et al. (2019). On the other hand, AID relies on the implicit function theorem to compute the implicit gradient of $\ell(\mathbf{x})$ without the need to maintain the sequence of LL iterates. Instead of differentiating the iterates of the LL problem, AID computes the implicit gradient by approximately solving a linear system of equations using HVPs. In this work, we focus on AID-based approaches for solving stochastic bilevel problems.

## C    EXPERIMENTAL DETAILS AND ADDITIONAL RESULTS

In this section, we present additional experimental results, which are not included in the main text, and provide a detailed description of the experimental setup.

### C.1    ADDITIONAL EXPERIMENTAL RESULTS

**Task 1) Data weighting for RLHF reward model training**

We first evaluate the effect of $N$ on the performance of SO-Lazy-BiO-I  algorithm for **Task 1**. Fig. 4 captures the effect of different values of $N$ on the performance of SO-Lazy-BiO-I . Note that when $N = 1$, SO-Lazy-BiO-I becomes a non-lazy algorithm, which is equivalent to SOBA. We observe that as we increase the value of $N$, the execution of the algorithm becomes faster. The fact that the validation loss remains stable as $N$ increases suggests that the HVP and JVP information evolves gradually during training. This indicates that using stale HVP and JVP can still yield accurate approximations of the hypergradient in bilevel optimization.

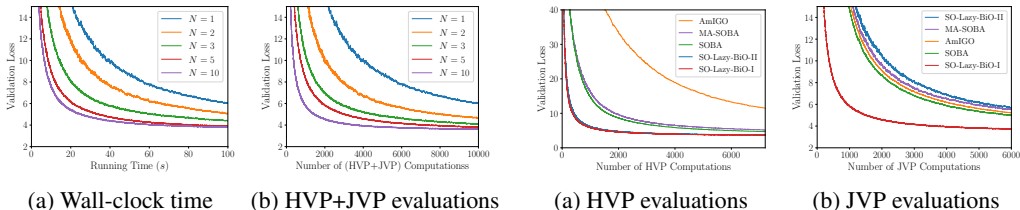

| (a) Wall-clock time | (b) HVP+JVP evaluations | (a) HVP evaluations | (b) JVP evaluations |

Figure 4: Validation loss comparison with different values of $N$ for data weighting in RLHF reward model training (Task 1).

Figure 5: Validation loss comparison with different bilevel algorithms for data weighting in RLHF reward model training (Task 1).

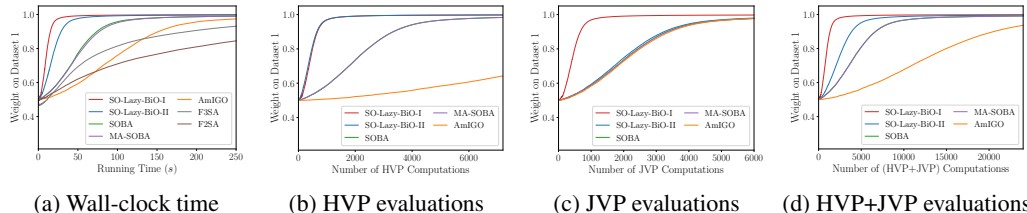

| (a) Wall-clock time | (b) HVP evaluations | (c) JVP evaluations | (d) HVP+JVP evaluations |

Figure 6: Comparison of weights assigned on the dataset corresponding to the validation set for data weighting in RLHF reward model training (Task 1).

In Fig. 5, we compare the convergence speed of different bilevel optimization algorithms with respect to the number of HVP and JVP computations. In Fig. 5a, both SO-Lazy-BiO-I and SO-Lazy-BiO-II achieve significantly faster convergence due to their infrequent HVP updates. Similarly, Fig. 5b shows that SO-Lazy-BiO-I converges faster than the other algorithms, which is attributed to its infrequent JVP computations. Although SO-Lazy-BiO-I and AmIGO demonstrate similar convergence performance, SO-Lazy-BiO-I requires substantially fewer HVP evaluations compared to AmIGO. These results verify that both HVP and JVP computations significantly impact the overall computational cost in bilevel optimization, and thus using stale second-order information can efficiently accelerate the convergence.

Fig. 6 shows the data weighting result for different bilevel optimization algorithms. All algorithms successfully assign higher weights to dataset 1, which is labeled using the same score criterion as the validation set. This validates the effectiveness of bilevel optimization framework when addressing the data weighting problem for RLHF reward model training. We observe that, while the weight value from every algorithm converges to 1, our SO-Lazy-BiO-I and SO-Lazy-BiO-II algorithms show faster convergence within the same runtime. This confirms the computational efficiency of our proposed SO-Lazy-BiO framework in bilevel optimization. In addition, by leveraging second-order information, our SO-Lazy-BiO-I and SO-Lazy-BiO-II algorithms assign higher weights compared to the fully first-order methods $F^2SA$ and $F^3SA$, thereby validating the effectiveness of our proposed algorithms.

**Task 2) Data weighting for LLM alignment**

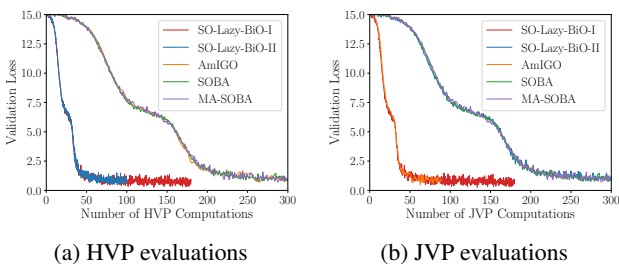

| (a) HVP evaluations | (b) JVP evaluations |

Figure 7: Validation loss comparison for data weighting in LLM alignment (Task 2).

Fig. 7 compares the convergence speed of various bilevel optimization algorithms with respect to the number of HVP and JVP computations. As anticipated, we observe a similar trend to that in Fig. 5: SO-Lazy-BiO-I and SO-Lazy-BiO-II converge faster in terms of HVP computations (Fig. 7a), while SO-Lazy-BiO-I shows faster convergence with JVP computations (Fig. 7b). However, for the case of LLM alignment, the performance gap becomes significantly larger. This is because the optimization variables for this problem are high-dimensional LLM parameters, making the overall bilevel optimization computationally intensive. Our results indicate that the computational advantage of our SO-Lazy-BiO algorithm becomes more significant when the scale of the bilevel problem becomes large.

**Task 3) Deep hyper-representation with ResNet network**

Fig. 8 illustrates the impact of HVP and JVP evaluations during training. Fig. 8a shows that SO-Lazy-BiO-I and SO-Lazy-BiO-II achieve faster convergence in terms of HVP evaluations compared to the other algorithms. Since SO-Lazy-BiO-II introduces less error than SO-Lazy-BiO-I, it

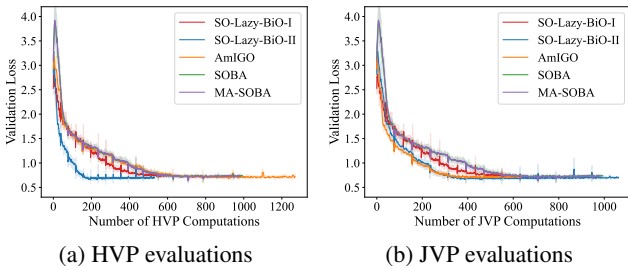

(a) HVP evaluations    (b) JVP evaluations

Figure 8: Validation loss comparison for deep hyper-representation (Task 3).

requires fewer iterations to converge and thus significantly fewer HVP evaluations. In Fig. 8b, we observe that SO-Lazy-BiO-II also requires fewer JVP computations to converge. Although SO-Lazy-BiO-II and AmIGO have comparable JVP computation costs, SO-Lazy-BiO-II achieves this with substantially fewer HVP evaluations than AmIGO. These findings validate the effectiveness of our lazy update design in reducing second-order information computation while maintaining strong convergence performance.

## C.2 Specifications of the baseline algorithms in Section 6

In this subsection, we describe the baseline algorithms used in our experiments, which are as follows:

- AmIGO Arbel & Mairal (2022): A double-loop stochastic AID-based bilevel algorithm that employs SGD to estimate the Hessian inverse.
- SOBA Dagréou et al. (2022): A single-loop stochastic AID-based bilevel algorithm that also uses SGD to approximate the Hessian inverse.
- MA-SOBA Chen et al. (2024): A single-loop stochastic AID-based bilevel algorithm that maintains an additional sequence of averaged hypergradients and uses SGD to estimate the Hessian inverse.
- F$^2$SA Kwon et al. (2023): A fully first-order (Hessian/Jacobian-free) stochastic bilevel algorithm with a double-loop structure.
- F$^3$SA Kwon et al. (2023): A fully first-order stochastic bilevel method that employs momentum-based SGD to accelerate convergence and operates on a single timescale.

## C.3 Experimental details of data weighting in RLHF reward model training

In this subsection, we provide the experimental details for the data weighting task in RLHF reward model training. In RLHF Ouyang et al. (2022), the reward model evaluates LLM prompt–response pairs using scores based on human-valued criteria like helpfulness, correctness, and verbosity. It is thus important to train the reward model using a carefully selected dataset. As considered in Shen et al. (2024); Pan et al. (2024), we determine dataset preferences through numerical weights and apply bilevel optimization to solve the problem.

Let $N_T$ be the number of datasets available for training. Each dataset $\mathcal{T}_n$, where $n = 1, 2, \ldots, N_T$, contains $|\mathcal{T}_n|$ samples, and each data sample $i = 1, 2, \ldots, |\mathcal{T}_n|$ consists of a prompt-response pair $\{p_{n,i}, r_{n,i}\}$ and its associated labeled score $s_{n,i}$. The goal of data weighting is to assign a weight on each dataset such that validation loss on a dataset $\mathcal{V}$ is minimized. We introduce $\mathbf{x} = [x_1, x_2, \ldots, x_{N_T}]^\top$ to be a vector of raw dataset weights, to which we apply a softmax function to derive the normalized weights. We also define $\mathbf{y} \in \mathbb{R}^l$ as the parameter vector of the reward model to be trained.

The bilevel optimization problem for our data weighting task in RLHF reward model training is then formulated as:

$$\min_{\mathbf{x} \in \mathbb{R}^{N_T}} \sum_{i=1}^{|\mathcal{V}|} \mathcal{L}(\tilde{s}_{0,i}, s_{0,i}; \mathbf{y}^*(\mathbf{x}))$$

$$\text{s.t.} \quad \mathbf{y}^*(\mathbf{x}) = \arg\min_{\mathbf{y} \in \mathbb{R}^l} \sum_{n=1}^{N_T} \frac{e^{x_n}}{\sum_{n'=1}^{N_T} e^{x_{n'}}} \sum_{i=1}^{|\mathcal{T}_n|} \mathcal{L}(\tilde{s}_{n,i}, s_{n,i}; \mathbf{y}),$$

where $\mathcal{L}(\tilde{s}_{n,i}, s_{n,i}; \mathbf{y})$ is the loss between the true score label $s_{n,i}$ and the predicted score $\tilde{s}_{n,i}$ generated by the reward model with parameters $\mathbf{y}$. In the problem, $\mathbf{y}^*(\mathbf{x})$ represents the optimal model parameters trained under data weights $\mathbf{x}$.

We configure our experimental setting as follows. We use the HelpSteer dataset Wang et al. (2023) (CC-by-4.0 License), where each prompt-response pair is labeled according to six different score criteria. We first filter the dataset to only include samples that have fewer than $1,000$ characters in total. Then, we select the two most uncorrelated criteria: *coherence* and *complexity*, and construct a mixed training dataset (i.e., $N_T = 2$). For the validation set, we exclusively label all samples with *coherence* scores, from which we expect the data weighting algorithm to assign greater weights on data labeled with *coherence*.

We use the DeBERTaV3 tokenizer He et al. (2021) (MIT License) to embed the text inputs. For the reward model, we implement a multi-layer perceptron (MLP) with width $500$ and depth $5$. The input dimension is set to $500$ to ensure that the tokenized texts are fully covered without truncation. We use mean squared error (MSE) as our loss function for both the UL and LL problem objectives. For our proposed SO-Lazy-BiO algorithms, we set $N = 5$, $\alpha = 1 \times 10^{-6}$, $\beta = 5 \times 10^{-7}$, and $\gamma = 5 \times 10^{-7}$. For AmIGO Arbel & Mairal (2022), we use $5$ update steps for both $\mathbf{y}$ and $\mathbf{z}$, with $\alpha = 1 \times 10^{-6}$, $\beta = 1 \times 10^{-7}$, and $\gamma = 1 \times 10^{-7}$. For both SOBA Dagréou et al. (2022) and MA-SOBA Chen et al. (2024), we set $\alpha = 1 \times 10^{-6}$, $\beta = 5 \times 10^{-7}$, $\gamma = 5 \times 10^{-7}$, and $\mu = 0.8$. For first-order methods, we set $\alpha = 5 \times 10^{-5}$, $\beta = 2 \times 10^{-8}$, and $\gamma = 2 \times 10^{-8}$ for F2SA and $\alpha = 5 \times 10^{-5}$, $\beta = 1 \times 10^{-7}$, $\gamma = 1 \times 10^{-7}$, and $\mu = 0.8$ for F3SA. All algorithms are trained with a batch size of $256$ and normalized gradient clipping with norm $1000$. We run the experiment on NVIDIA H100 NVL GPU.

## C.4 EXPERIMENTAL DETAILS OF DATA WEIGHTING IN LLM ALIGNMENT

In this subsection, we describe the experimental setup for the data weighting task in LLM alignment. Similar to the data weighting task in Section C.3, the goal is to find training sample weights that minimize the validation loss. However, instead of training a reward model on scalar reward labels, we fine-tune an LLM directly on prompt-response pairs that reflect human preferences.

We assume that each prompt-response sample for training has been categorized into one of $N_T$ distinct groups. Taking the notation from Section C.3, the bilevel optimization problem for our data weighting task in LLM alignment is formulated as:

$$\min_{\mathbf{x} \in \mathbb{R}^{N_T}} \sum_{i=1}^{|\mathcal{V}|} \mathcal{L}(\tilde{r}_{0,i}, r_{0,i}; p_{0,i}, \mathbf{y}^*(\mathbf{x}))$$

$$\text{s.t.} \quad \mathbf{y}^*(\mathbf{x}) = \arg\min_{\mathbf{y} \in \mathbb{R}^l} \sum_{n=1}^{N_T} \frac{e^{x_n}}{\sum_{n'=1}^{N_T} e^{x_{n'}}} \sum_{i=1}^{|\mathcal{T}_n|} \mathcal{L}(\tilde{r}_{n,i}, r_{n,i}; p_{n,i}, \mathbf{y}),$$

where $\mathcal{L}(\tilde{r}_{n,i}, r_{n,i}; p_{n,i}, \mathbf{y})$ denotes the loss between the true response $r_{n,i}$ and the response $\tilde{r}_{n,i}$ generated by the LLM of parameters $\mathbf{y}$ with given prompt $p_{n,i}$.

We use Llama-3.2-1B-Instruct Meta (2024) (Llama3.2 License) as the base LLM and apply the low-rank adaptation (LoRA) technique of rank $8$. We train the LLM on Anthropic HH-RLHF dataset Bai et al. (2022) (MIT License), where each text sample is labeled as either *chosen* or *rejected* (i.e., $N_T = 2$). For the validation set, we only include samples that have been *chosen*. In this setting, we anticipate that the validation loss can be further minimized when higher weights are assigned on training samples that have been *chosen*. We use cross-entropy as our loss function for both the UL and LL problem objectives. For our proposed SO-Lazy-BiO algorithms, we set $N = 5$. For AmIGO Arbel & Mairal (2022), we set both the number of $\mathbf{y}$ and $\mathbf{z}$ update steps as $5$. For MA-SOBA Chen et al. (2024), we set $\mu = 0.8$. All algorithms use the same update parameter values $\alpha = 5 \times 10^{-3}$, $\beta = 2 \times 10^{-4}$, $\gamma = 3 \times 10^{-4}$, and a batch size of $32$. We run the experiment on NVIDIA H100 NVL GPU.

### C.5 EXPERIMENTAL DETAILS FOR DEEP HYPER-REPRESENTATION WITH RESNET NETWORK

In this subsection, we show the details of the experiments on deep hyper-representation, which aims to classify the images. The objective function is given by:

$$\min_\lambda \mathcal{L}_{\mathcal{D}_{val}}(\lambda, w^*) = \frac{1}{|\mathcal{D}_{val}|} \sum_{(\mathbf{x}_i, \mathbf{y}_i) \in \mathcal{D}_{val}} \mathcal{L}\left(w^* f\left(\lambda; \mathbf{x}_i\right), \mathbf{y}_i\right)$$

$$\text{s.t.} \quad w^* = \arg\min_w \frac{1}{|\mathcal{D}_{tr}|} \sum_{(\mathbf{x}_i, \mathbf{y}_i) \in \mathcal{D}_{tr}} \mathcal{L}\left(w f\left(\lambda, \mathbf{x}_i\right), \mathbf{y}_i\right),$$

where $(\mathbf{x}_i, \mathbf{y}_i)$ denotes the data samples, $\mathcal{D}_{val}$ and $\mathcal{D}_{tr}$ are the validation data and the training data, $\mathcal{L}$ corresponds to the cross-entropy loss, $f\left(\lambda; \mathbf{x}_i\right)$ represents the features extracted from the data sample. We run the experiments with ResNet-20 network He et al. (2016) on CIFAT-10 dataset Krizhevsky et al. (2009) using a batch size of 128. We treat the last two layers in ResNet-20 as the LL parameters $w$ with a dimension of 5, 130, and all remaining layers as the UL parameters $\lambda$ with a dimension of 11, 168, 832.

We compare SO-Lazy-BiO-I and SO-Lazy-BiO-II with AmIGO Arbel & Mairal (2022), SOBA Dagréou et al. (2022), MA-SOBA Chen et al. (2024), F$^2$SA Kwon et al. (2023) and F$^3$SA Kwon et al. (2023). To ensure the best performance of all the algorithms, we fine-tune the parameters using grid search with the goal of finding the lowest validation loss, which corresponds to the upper-level objective. Consequently, for SO-Lazy-BiO-I and SO-Lazy-BiO-II, we choose the step sizes to $\alpha = 0.005$, $\beta = 0.05$, and $\gamma = 0.01$, and choose a lazy update frequency of $N = 2$. The momentum coefficient is set to $\mu = 0.8$ for SO-Lazy-BiO-I and $\mu = 1.0$ for SO-Lazy-BiO-II. For AmIGO, SOBA, and MA-SOBA, we choose all the step-sizes for updating $\mathbf{x}$, $\mathbf{y}$, and $\mathbf{z}$ to 0.01. For AmIGO, we set the number of y-update iterations to be 8 and the number of z-update iterations to be 2. For MA-SOBA, we choose the momentum coefficient to be 0.9. Following the same notations as in Kwon et al. (2023), for F$^2$SA, we choose the step-sizes $\alpha = 0.1$ and $\gamma = 0.05$. We use the step-size ratio $\xi = 0.5$ and the Lagrangian multiplier $\lambda = 0.1$. We choose the number of inner-loop iterations to be 1. For F$^3$SA, we set 0.05 as $\alpha$, 0.01 as $\gamma$, 0.1 as $\xi$, 0.5 as $\lambda$, and 0.9 as momentum-weight $\eta$. We repeat the experiments 5 times with different random seeds, where the solid line represents the average validation loss, and the shaded area shows the variance containing the maximum and the minimum values. We run the deep hyper-representation experiments using NVIDIA GeForce RTX 3060 GPU.

## D SUPPORTING LEMMAS

**Lemma D.1** (Lemma 2.2 in Ghadimi & Wang (2018)). *Under Assumptions 5.1 and 5.2, we have*

$$\|\nabla\ell(\mathbf{x}_1) - \nabla\ell(\mathbf{x}_2)\| \le L_l \|\mathbf{x}_1 - \mathbf{x}_2\|, \quad \|\mathbf{y}^*(\mathbf{x}_1) - \mathbf{y}^*(\mathbf{x}_2)\| \le L_y \|\mathbf{x}_1 - \mathbf{x}_2\|,$$

*for all $\mathbf{x}, \mathbf{x}_1, \mathbf{x}_2 \in \mathbb{R}^u$, where the Lipschitz constants above are defined as:*

$$L_l = L_f^{'} + \left(L_f^{'} B_{g_{xy}}/\mu_g\right), \quad L_y = B_{g_{xy}}/\mu_g,$$

*and where $L_f^{'} = L_{f_x} + (L_{f_y} B_{g_{xy}}/\mu_g) + B_{f_y}\left[(L_{g_{xy}}/\mu_g) + (L_{g_{yy}} B_{g_{xy}}/\mu_g^2)\right]$.*

**Lemma D.2** (Lemma 3.4 in Dagréou et al. (2022)). *Under Assumptions 5.1 and 5.2, we have*

$$\|\nabla f(\mathbf{x}, \mathbf{y}, \mathbf{z}) - \nabla\ell(\mathbf{x})\| \le L_f \left(\|\mathbf{y} - \mathbf{y}^*(\mathbf{x})\| + \|\mathbf{z} - \mathbf{z}^*(\mathbf{x}, \mathbf{y}^*(\mathbf{x}))\|\right),$$

*for all $\mathbf{x} \in \mathbb{R}^u$, and $\mathbf{y}, \mathbf{z} \in \mathbb{R}^l$, where the Lipschitz constants above are defined as:*

$$L_f = \max\left\{L_{f_x} + (L_{g_{xy}} B_{f_y}/\mu_g), B_{g_{xy}}\right\}.$$

**Lemma D.3** (Lemma C.1 in Dagréou et al. (2022), Lemma 10 in Chen et al. (2024)). *Under Assumptions 5.1 and 5.2, $\forall\, \mathbf{x}, \mathbf{x}_1, \mathbf{x}_2 \in \mathbb{R}^u$ and $\mathbf{y} \in \mathbb{R}^l$, we have*

$$\|\mathbf{z}^*(\mathbf{x}_1, \mathbf{y}^*(\mathbf{x}_1)) - \mathbf{z}^*(\mathbf{x}_2, \mathbf{y}^*(\mathbf{x}_2))\| \le L_z \|\mathbf{x}_1 - \mathbf{x}_2\|, \quad \|\mathbf{z}^*(\mathbf{x}, \mathbf{y})\| \le B_{f_y}/\mu_g,$$

*where $L_z = (1 + L_y)\left((L_{g_{yy}} B_{f_y}/\mu_g^2) + L_{f_y}/\mu_g\right)$.*

**Lemma D.4** (Quadratic Problem). *For any $(\mathbf{x}, \mathbf{y}) \in \mathbb{R}^u \times \mathbb{R}^l$, the map $\mathbf{z} \mapsto q(\mathbf{x}, \mathbf{y}, \mathbf{z})$ is $\mu_g$-strongly convex and $L_q$-Lipschitz smooth with constants $\mu_g > 0$ and $L_q \ge 0$.*

# E    THEORETICAL ANALYSIS OF OPTION I IN SO-Lazy-BiO FRAMEWORK

## E.1    REFORMULATION OF **OPTION I** IN ALGORITHM 1 FOR THEORETICAL ANALYSIS

In order to analyze the theoretical performance of SO-Lazy-BiO-I, we reformulate SO-Lazy-BiO-I as follows. We note that **Option I** in Algorithm 1 is equivalent to Algorithm 2 when the number of iterations $T$ in Algorithm 2 is set to $T/N$.

---

**Algorithm 2** The SO-Lazy-BiO-I Algorithm.

---

**Input:** Initial parameters $\mathbf{x}_0^0, \mathbf{y}_0^0, \mathbf{z}_0$, stepsizes $\{\alpha_t, \beta_t, \gamma_t\}_{t=0}^{T-1}$, and momentum coefficient $\{\mu_t\}_{t=0}^{T-1}$

**for** $t = 0$ **to** $T - 1$ **do**

    Initialize $\mathbf{x}_t^0 = \mathbf{x}_{t-1}^N$ and $\mathbf{y}_t^0 = \mathbf{y}_{t-1}^N$

    Sample data batches $\mathcal{D}_t^{g_{yy}}$ $\mathcal{D}_t^{f_y}$, and $\mathcal{D}_t^{g_{xy}}$

    Compute the gradient estimate $h_t^q$ using $h_t^q = \nabla_{\mathbf{y}\mathbf{y}}^2 g(\mathbf{x}_t^0, \mathbf{y}_t^0; \mathcal{D}_t^{g_{yy}})\mathbf{z}_t + \nabla_{\mathbf{y}} f(\mathbf{x}_t^0, \mathbf{y}_t^0; \mathcal{D}_t^{f_y})$

    Update $\mathbf{z}_{t+1} = \mathbf{z}_t - \gamma_t h_t^q$

    Compute the JVP using $\mathbf{v}_t = \nabla_{\mathbf{x}\mathbf{y}}^2 g\left(\mathbf{x}_t^0, \mathbf{y}_t^0; \mathcal{D}_t^{g_{xy}}\right) \mathbf{z}_t$

    **for** $n = 0$ **to** $N - 1$ **do**

        Sample data batches $\mathcal{D}_{t,n}^g, \mathcal{D}_{t,n}^{f_x}$, and $\mathcal{D}_{t,n}^{g_{xy}}$

        Compute the gradient estimate $h_{t,n}^g$ using $h_{t,n}^g = \nabla_{\mathbf{y}} g\left(\mathbf{x}_t^n, \mathbf{y}_t^n; \mathcal{D}_{t,n}^g\right)$

        Update $\mathbf{y}_t^{n+1} = \mathbf{y}_t^n - \beta_t h_{t,n}^g$

        Compute the gradient estimate $h_{t,n}^f$ using $h_{t,n}^f = \nabla_{\mathbf{x}} f\left(\mathbf{x}_t^n, \mathbf{y}_t^n; \mathcal{D}_{t,n}^{f_x}\right) + \mathbf{v}_t$

        Compute the momentum-based $\bar{h}_{t,n+1}^f$ using $\bar{h}_{t,n+1}^f = \mu_t h_{t,n}^f + (1 - \mu_t) \bar{h}_{t,n}^f$

        Update $\mathbf{x}_t^{n+1} = \mathbf{x}_t^n - \alpha_t \bar{h}_{t,n}^f$

    **end for**

**end for**

---

## E.2    DETAILED PROOF OF THEOREM 5.5: NON-CONVEX $\ell(\mathbf{x})$

### E.2.1    PROOF OF PRELIMINARY LEMMAS

**Lemma E.1.** *Under Assumptions 5.2 and 5.3, the following inequality holds:*

$$\mathbb{E}\left[\left\|h_{t,n}^g\right\|^2\right] \leq 2L_g^2 \mathbb{E}\left[\left\|\mathbf{y}_t^n - \mathbf{y}^*(\mathbf{x}_t^n)\right\|^2\right] + 2\sigma_{g_y}^2,$$

*for all $t \in \{0, 1, \ldots, T - 1\}$ and $n \in \{0, 1, \ldots, N - 1\}$, where the expectation is taken over the stochasticity of the algorithm.*

*Proof.* We get

$$\mathbb{E}\left[\left\|h_{t,n}^g\right\|^2\right] = \mathbb{E}\left[\left\|h_{t,n}^g - \nabla_{\mathbf{y}} g\left(\mathbf{x}_t^n, \mathbf{y}_t^n\right) + \nabla_{\mathbf{y}} g\left(\mathbf{x}_t^n, \mathbf{y}_t^n\right)\right\|^2\right]$$

$$\overset{(a)}{\leq} \mathbb{E}\left[2\left\|h_{t,n}^g - \nabla_{\mathbf{y}} g\left(\mathbf{x}_t^n, \mathbf{y}_t^n\right)\right\|^2 + 2\left\|\nabla_{\mathbf{y}} g\left(\mathbf{x}_t^n, \mathbf{y}_t^n\right) - \nabla_{\mathbf{y}} g\left(\mathbf{x}_t^n, \mathbf{y}^*(\mathbf{x}_t^n)\right)\right\|^2\right]$$

$$\overset{(b)}{\leq} 2L_g^2 \mathbb{E}\left[\left\|\mathbf{y}_t^n - \mathbf{y}^*(\mathbf{x}_t^n)\right\|^2\right] + 2\sigma_{g_y}^2,$$

where (a) is because of $\nabla_{\mathbf{y}} g\left(\mathbf{x}_t^n, \mathbf{y}^*(\mathbf{x}_t^n)\right) = 0$, and (b) uses Assumptions 5.2 and 5.3. $\qquad\square$

### E.2.2    DESCENT IN THE UPPER-LEVEL OBJECTIVE FUNCTION

**Lemma E.2.** *Under Assumptions 5.1 and 5.2, the following inequality holds for successive iterations of Algorithm 2:*

$$\mathbb{E}\left[\ell\left(\mathbf{x}_t^{n+1}\right) - \ell\left(\mathbf{x}_t^n\right)\right]$$

$$\leq -\frac{\alpha_t}{2}\mathbb{E}\left[\left\|\nabla\ell\left(\mathbf{x}_t^n\right)\right\|^2\right] - \frac{\alpha_t}{2}\mathbb{E}\left[\left\|\bar{h}_{t,n}^f\right\|^2\right] + \frac{\alpha_t}{2}\mathbb{E}\left[\left\|\nabla\ell\left(\mathbf{x}_t^n\right) - \bar{h}_{t,n}^f\right\|^2\right] + \frac{\alpha_t^2 L_l}{2}\mathbb{E}\left[\left\|\bar{h}_{t,n}^f\right\|^2\right],$$

*for all $t \in \{0, 1, \ldots, T - 1\}$ and $n \in \{0, 1, \ldots, N - 1\}$, where the expectation is taken over the stochasticity of the algorithm.*

*Proof.* We have

$$\mathbb{E}\left[\ell\left(\mathbf{x}_t^{n+1}\right) - \ell\left(\mathbf{x}_t^n\right)\right]$$

$$\overset{(a)}{\leq} \mathbb{E}\left[\left\langle \nabla\ell\left(\mathbf{x}_t^n\right), \mathbf{x}_t^{n+1} - \mathbf{x}_t^n \right\rangle + \frac{L_l}{2}\left\|\mathbf{x}_t^{n+1} - \mathbf{x}_t^n\right\|^2\right]$$

$$\overset{(b)}{=} \mathbb{E}\left[-\alpha_t\left\langle \nabla\ell\left(\mathbf{x}_t^n\right), \bar{h}_{t,n}^f \right\rangle + \frac{\alpha_t^2 L_l}{2}\left\|\bar{h}_{t,n}^f\right\|^2\right]$$

$$\overset{(c)}{=} \mathbb{E}\left[-\frac{\alpha_t}{2}\left\|\nabla\ell\left(\mathbf{x}_t^n\right)\right\|^2 - \frac{\alpha_t}{2}\left\|\bar{h}_{t,n}^f\right\|^2 + \frac{\alpha_t}{2}\left\|\nabla\ell\left(\mathbf{x}_t^n\right) - \bar{h}_{t,n}^f\right\|^2 + \frac{\alpha_t^2 L_l}{2}\left\|\bar{h}_{t,n}^f\right\|^2\right],$$

where (a) uses the Lipschitz continuous gradients of $\ell$ (see Lemma D.1). (b) follows from the update rule of Algorithm 2. (c) is because of $\langle x, y\rangle = \frac{1}{2}\|x\|^2 + \frac{1}{2}\|y\|^2 - \frac{1}{2}\|x-y\|^2$. $\square$

### E.2.3 DESCENT IN THE APPROXIMATION ERROR OF $\nabla\ell\left(\mathbf{x}\right)$

**Lemma E.3.** *Under Assumptions 5.1–5.3, the approximation error of $\nabla\ell\left(\mathbf{x}\right)$ of Algorithm 2 satisfies the following inequality:*

$$\mathbb{E}\left[\left\|\nabla\ell\left(\mathbf{x}_t^{n+1}\right) - \bar{h}_{t,n+1}^f\right\|^2\right]$$

$$\leq (1-\mu_t)\,\mathbb{E}\left[\left\|\nabla\ell\left(\mathbf{x}_t^n\right) - \bar{h}_{t,n}^f\right\|^2\right] + 4\mu_t L_f^2\,\mathbb{E}\left[\left\|\mathbf{y}_t^n - \mathbf{y}^*\left(\mathbf{x}_t^n\right)\right\|^2\right] + 8\mu_t L_f^2\,\mathbb{E}\left[\left\|\mathbf{z}_t - \mathbf{z}_t^*\right\|^2\right]$$

$$+ 16 L_g^2 \mu_t^2 L_{g_{xy}}^2 B_z^2 \beta_t^2 n \sum_{i=0}^{n-1}\mathbb{E}\left[\left\|\mathbf{y}_t^i - \mathbf{y}^*\left(\mathbf{x}_t^i\right)\right\|^2\right] + \frac{2}{\mu_t}L_l^2\alpha_t^2\,\mathbb{E}\left[\left\|\bar{h}_{t,n}^f\right\|^2\right]$$

$$+ 8\mu_t^2 L_{g_{xy}}^2 B_z^2 \alpha_t^2 n \sum_{i=0}^{n-1}\mathbb{E}\left[\left\|\bar{h}_{t,i}^f\right\|^2\right] + 8\mu_t L_f^2 L_z^2 \alpha_t^2 n \sum_{i=0}^{n-1}\mathbb{E}\left[\left\|\bar{h}_{t,i}^f\right\|^2\right]$$

$$+ 4 B_z^2 \sigma_{g_{xy}}^2 \mu_t^2 + 2\sigma_{f_x}^2 \mu_t^2 + 16 L_{g_{xy}}^2 B_z^2 \beta_t^2 n^2 \sigma_{g_y}^2 \mu_t^2,$$

*for all $t \in \{0, 1, \ldots, T-1\}$ and $n \in \{0, 1, \ldots, N-1\}$, where $\mathbf{z}_t^* = \mathbf{z}^*\left(\mathbf{x}_t^0, \mathbf{y}^*(\mathbf{x}_t^0)\right)$, and the expectation is taken over the stochasticity of the algorithm.*

*Proof.* We have

$$\mathbb{E}\left[\left\|\nabla\ell\left(\mathbf{x}_t^{n+1}\right) - \bar{h}_{t,n+1}^f\right\|^2\right] = \mathbb{E}\left[\left\|\mu_t h_{t,n}^f + (1-\mu_t)\bar{h}_{t,n}^f - \nabla\ell\left(\mathbf{x}_t^{n+1}\right)\right\|^2\right]$$

$$\leq \mathbb{E}\left[(1-\mu_t)\left\|\bar{h}_{t,n}^f - \nabla\ell\left(\mathbf{x}_t^n\right)\right\|^2 + \mu_t^2\left\|h_{t,n}^f - \nabla f\left(\mathbf{x}_t^n, \mathbf{y}_t^n, \mathbf{z}_t\right)\right\|^2\right.$$

$$\left. + \mu_t\left\|\nabla f\left(\mathbf{x}_t^n, \mathbf{y}_t^n, \mathbf{z}_t\right) - \nabla\ell\left(\mathbf{x}_t^n\right) + \frac{1}{\mu_t}\left(\nabla\ell\left(\mathbf{x}_t^n\right) - \nabla\ell\left(\mathbf{x}_t^{n+1}\right)\right)\right\|^2\right]$$

$$\leq \mathbb{E}\left[(1-\mu_t)\left\|\bar{h}_{t,n}^f - \nabla\ell\left(\mathbf{x}_t^n\right)\right\|^2 + \mu_t^2\left\|h_{t,n}^f - \nabla f\left(\mathbf{x}_t^n, \mathbf{y}_t^n, \mathbf{z}_t\right)\right\|^2\right.$$

$$\left. + 2\mu_t\left\|\nabla f\left(\mathbf{x}_t^n, \mathbf{y}_t^n, \mathbf{z}_t\right) - \nabla\ell\left(\mathbf{x}_t^n\right)\right\|^2 + \frac{2}{\mu_t}\left\|\nabla\ell\left(\mathbf{x}_t^n\right) - \nabla\ell\left(\mathbf{x}_t^{n+1}\right)\right\|^2\right]$$

$$\overset{(a)}{\leq} \mathbb{E}\left[(1-\mu_t)\left\|\bar{h}_{t,n}^f - \nabla\ell\left(\mathbf{x}_t^n\right)\right\|^2 + \mu_t^2\left\|h_{t,n}^f - \nabla f\left(\mathbf{x}_t^n, \mathbf{y}_t^n, \mathbf{z}_t\right)\right\|^2\right.$$

$$\left. + 2\mu_t L_f^2\left(\left\|\mathbf{y}_t^n - \mathbf{y}^*(\mathbf{x}_t^n)\right\| + \left\|\mathbf{z}_t - \mathbf{z}^*\left(\mathbf{x}_t^n, \mathbf{y}^*(\mathbf{x}_t^n)\right)\right\|\right)^2 + \frac{2}{\mu_t}L_l^2\left\|\mathbf{x}_t^{n+1} - \mathbf{x}_t^n\right\|^2\right]$$

$$\leq \mathbb{E}\left[(1-\mu_t)\left\|\bar{h}_{t,n}^f - \nabla\ell\left(\mathbf{x}_t^n\right)\right\|^2 + \mu_t^2\left\|h_{t,n}^f - \nabla f\left(\mathbf{x}_t^n, \mathbf{y}_t^n, \mathbf{z}_t\right)\right\|^2\right.$$

$$\left. + 4\mu_t L_f^2\left\|\mathbf{y}_t^n - \mathbf{y}^*(\mathbf{x}_t^n)\right\|^2 + 4\mu_t L_f^2\left\|\mathbf{z}_t - \mathbf{z}^*\left(\mathbf{x}_t^n, \mathbf{y}^*(\mathbf{x}_t^n)\right)\right\|^2 + \frac{2}{\mu_t}L_l^2\alpha_t^2\left\|\bar{h}_{t,n}^f\right\|^2\right], \qquad (10)$$

where (a) utilizes the Lipschitzness of $\nabla \ell(\mathbf{x})$ (see Lemma D.1) and the Lipschitzness of $\nabla f(\mathbf{x}, \mathbf{y}, \mathbf{z})$ (see Lemma D.2).

Then, we bound $\left\|\mathbf{x}_t^n - \mathbf{x}_t^0\right\|^2$ and $\left\|\mathbf{y}_t^n - \mathbf{y}_t^0\right\|^2$.

$$\left\|\mathbf{x}_t^n - \mathbf{x}_t^0\right\|^2 \overset{(a)}{=} \alpha_t^2 \left\|\sum_{i=0}^{n-1} \bar{h}_{t,i}^f\right\|^2 \overset{(b)}{\leq} \alpha_t^2 n \sum_{i=0}^{n-1} \left\|\bar{h}_{t,i}^f\right\|^2 \leq \alpha_t^2 N \sum_{i=0}^{N-1} \left\|\bar{h}_{t,i}^f\right\|^2, \qquad (11)$$

where (a) is because of the update rule of Algorithm 2. (b) is due to $\|z_1 + \cdots + z_k\|^2 \leq k\|z_1\|^2 + \cdots + k\|z_k\|^2$.

Similarly,

$$\left\|\mathbf{y}_t^n - \mathbf{y}_t^0\right\|^2 \leq \beta_t^2 n \sum_{i=0}^{n-1} \left\|h_{t,i}^g\right\|^2 \leq \beta_t^2 N \sum_{i=0}^{N-1} \left\|h_{t,i}^g\right\|^2. \qquad (12)$$

We bound the term $\mathbb{E}\left[\left\|h_{t,n}^f - \nabla f(\mathbf{x}_t^n, \mathbf{y}_t^n, \mathbf{z}_t)\right\|^2\right]$ in Eq. (10).

$$\mathbb{E}\left[\left\|h_{t,n}^f - \nabla f(\mathbf{x}_t^n, \mathbf{y}_t^n, \mathbf{z}_t)\right\|^2\right]$$

$$\overset{(a)}{=} \mathbb{E}\left[\left\|\nabla_{\mathbf{x}} f\left(\mathbf{x}_t^n, \mathbf{y}_t^n, \mathcal{D}_{t,n}^{f_x}\right) + \nabla_{\mathbf{xy}}^2 g\left(\mathbf{x}_t^0, \mathbf{y}_t^0, \mathcal{D}_t^{g_{xy}}\right) \mathbf{z}_t - \nabla_{\mathbf{x}} f(\mathbf{x}_t^n, \mathbf{y}_t^n) - \nabla_{\mathbf{xy}}^2 g(\mathbf{x}_t^n, \mathbf{y}_t^n) \mathbf{z}_t\right\|^2\right]$$

$$\leq \mathbb{E}\left[2\left\|\nabla_{\mathbf{x}} f\left(\mathbf{x}_t^n, \mathbf{y}_t^n, \mathcal{D}_{t,n}^{f_x}\right) - \nabla_{\mathbf{x}} f(\mathbf{x}_t^n, \mathbf{y}_t^n)\right\|^2\right.$$

$$\left. + 2\|\mathbf{z}_t\|^2 \left\|\nabla_{\mathbf{xy}}^2 g\left(\mathbf{x}_t^0, \mathbf{y}_t^0, \mathcal{D}_t^{g_{xy}}\right) - \nabla_{\mathbf{xy}}^2 g(\mathbf{x}_t^n, \mathbf{y}_t^n)\right\|^2\right]$$

$$\overset{(b)}{\leq} \mathbb{E}\left[4\|\mathbf{z}_t\|^2 \left\|\nabla_{\mathbf{xy}}^2 g\left(\mathbf{x}_t^0, \mathbf{y}_t^0, \mathcal{D}_t^{g_{xy}}\right) - \nabla_{\mathbf{xy}}^2 g(\mathbf{x}_t^0, \mathbf{y}_t^0)\right\|^2\right.$$

$$\left. + 4\|\mathbf{z}_t\|^2 \left\|\nabla_{\mathbf{xy}}^2 g(\mathbf{x}_t^0, \mathbf{y}_t^0) - \nabla_{\mathbf{xy}}^2 g(\mathbf{x}_t^n, \mathbf{y}_t^n)\right\|^2\right] + 2\sigma_{f_x}^2$$

$$\overset{(c)}{\leq} \mathbb{E}\left[4\|\mathbf{z}_t\|^2 \sigma_{g_{xy}}^2 + 4L_{g_{xy}}^2 \|\mathbf{z}_t\|^2 \left(\left\|\mathbf{x}_t^n - \mathbf{x}_t^0\right\| + \left\|\mathbf{y}_t^n - \mathbf{y}_t^0\right\|\right)^2\right] + 2\sigma_{f_x}^2$$

$$\overset{(d)}{\leq} \mathbb{E}\left[4\|\mathbf{z}_t\|^2 \sigma_{g_{xy}}^2 + 8L_{g_{xy}}^2 \|\mathbf{z}_t\|^2 \alpha_t^2 n \sum_{i=0}^{n-1} \left\|\bar{h}_{t,i}^f\right\|^2 + 8L_{g_{xy}}^2 \|\mathbf{z}_t\|^2 \beta_t^2 n \sum_{i=0}^{n-1} \left\|h_{t,i}^g\right\|^2\right] + 2\sigma_{f_x}^2$$

$$\overset{(e)}{\leq} \mathbb{E}\left[8L_{g_{xy}}^2 B_z^2 \alpha_t^2 n \sum_{i=0}^{n-1} \left\|\bar{h}_{t,i}^f\right\|^2 + 16L_g^2 L_{g_{xy}}^2 B_z^2 \beta_t^2 n \sum_{i=0}^{n-1} \left\|\mathbf{y}_t^i - \mathbf{y}^*(\mathbf{x}_t^i)\right\|^2\right]$$

$$+ 4B_z^2 \sigma_{g_{xy}}^2 + 2\sigma_{f_x}^2 + 16L_{g_{xy}}^2 B_z^2 \beta_t^2 n^2 \sigma_{g_y}^2, \qquad (13)$$

where (a) uses the definitions of $h_{t,n}^f$ and $\nabla f(\mathbf{x}_t^n, \mathbf{y}_t^n, \mathbf{z}_t)$. (b) utilizes the bounded variance in Assumption 5.3. (c) uses Assumptions 5.2 and 5.3. (d) follows from Eq. (11) and (12), and (e) is due to $\|\mathbf{z}_t\| \leq B_z$ and Lemma E.1.

Then, we bound the term $\mathbb{E}\left[\left\|\mathbf{z}_t - \mathbf{z}^*(\mathbf{x}_t^n, \mathbf{y}^*(\mathbf{x}_t^n))\right\|^2\right]$ in Eq. (10).

$$\mathbb{E}\left[\left\|\mathbf{z}_t - \mathbf{z}^*(\mathbf{x}_t^n, \mathbf{y}^*(\mathbf{x}_t^n))\right\|^2\right]$$

$$\leq \mathbb{E}\left[2\left\|\mathbf{z}_t - \mathbf{z}^*(\mathbf{x}_t^0, \mathbf{y}^*(\mathbf{x}_t^0))\right\|^2 + 2\left\|\mathbf{z}^*(\mathbf{x}_t^0, \mathbf{y}^*(\mathbf{x}_t^0)) - \mathbf{z}^*(\mathbf{x}_t^n, \mathbf{y}^*(\mathbf{x}_t^n))\right\|^2\right]$$

$$\overset{(a)}{\leq} \mathbb{E}\left[2\left\|\mathbf{z}_t - \mathbf{z}^*(\mathbf{x}_t^0, \mathbf{y}^*(\mathbf{x}_t^0))\right\|^2 + 2L_z^2 \left\|\mathbf{x}_t^n - \mathbf{x}_t^0\right\|^2\right]$$

$$\overset{(b)}{\leq} \mathbb{E}\left[2\left\|\mathbf{z}_t - \mathbf{z}^*(\mathbf{x}_t^0, \mathbf{y}^*(\mathbf{x}_t^0))\right\|^2 + 2L_z^2 \alpha_t^2 n \sum_{i=0}^{n-1} \left\|\bar{h}_{t,i}^f\right\|^2\right], \qquad (14)$$

where (a) follows from the Lipschitzness of $\mathbf{z}^*(\mathbf{x}, \mathbf{y}^*(\mathbf{x}))$ (see Lemma D.3), and (b) uses Eq. (11).

Combining Eq. (10), (13), and (14) completes the proof of the lemma. □

### E.2.4 DESCENT IN THE APPROXIMATION ERROR OF $\mathbf{y}^*(\mathbf{x})$

**Lemma E.4.** *Under Assumptions 5.2 and 5.3, the approximation error of $\mathbf{y}^*(\mathbf{x})$ of Algorithm 2 satisfies the following inequality:*

$$\mathbb{E}\left[\left\|\mathbf{y}_t^{n+1} - \mathbf{y}^*\left(\mathbf{x}_t^{n+1}\right)\right\|^2\right]$$

$$\leq \left(1 - \frac{\beta_t \mu_g}{2}\right)\mathbb{E}\left[\left\|\mathbf{y}_t^n - \mathbf{y}^*\left(\mathbf{x}_t^n\right)\right\|^2\right] + \frac{2}{\beta_t \mu_g} L_y^2 \alpha_t^2 \mathbb{E}\left[\left\|\bar{h}_{t,n}^f\right\|^2\right] + 4\beta_t^2 \sigma_{g_y}^2,$$

*for all $t \in \{0, 1, \ldots, T-1\}$ and $n \in \{0, 1, \ldots, N-1\}$, where the expectation is taken over the stochasticity of the algorithm.*

*Proof.* We have

$$\mathbb{E}\left[\left\|\mathbf{y}_t^{n+1} - \mathbf{y}^*\left(\mathbf{x}_t^{n+1}\right)\right\|^2\right]$$

$$\overset{(a)}{\leq} \mathbb{E}\left[(1 + c_1)\left\|\mathbf{y}_t^{n+1} - \mathbf{y}^*\left(\mathbf{x}_t^n\right)\right\|^2 + \left(1 + \frac{1}{c_1}\right)\left\|\mathbf{y}^*\left(\mathbf{x}_t^n\right) - \mathbf{y}^*\left(\mathbf{x}_t^{n+1}\right)\right\|^2\right]$$

$$\overset{(b)}{\leq} \mathbb{E}\left[(1 + c_1)\left\|\mathbf{y}_t^{n+1} - \mathbf{y}^*\left(\mathbf{x}_t^n\right)\right\|^2 + \left(1 + \frac{1}{c_1}\right)L_y^2\left\|\mathbf{x}_t^{n+1} - \mathbf{x}_t^n\right\|^2\right]$$

$$\overset{(c)}{\leq} \mathbb{E}\left[(1 + c_1)\left\|\mathbf{y}_t^{n+1} - \mathbf{y}^*\left(\mathbf{x}_t^n\right)\right\|^2 + \left(1 + \frac{1}{c_1}\right)L_y^2 \alpha_t^2 \left\|\bar{h}_{t,n}^f\right\|^2\right], \tag{15}$$

where (a) results from Young's inequality. (b) is because of the Lipschitzness of $\mathbf{y}^*(\cdot)$ (see Lemma D.1). (c) follows from the update rule of Algorithm 2.

To bound the first term on the right, we have

$$\mathbb{E}\left[\left\|\mathbf{y}_t^{n+1} - \mathbf{y}^*\left(\mathbf{x}_t^n\right)\right\|^2\right]$$

$$= \mathbb{E}\left[\left\|\mathbf{y}_t^n - \mathbf{y}^*\left(\mathbf{x}_t^n\right)\right\|^2 + \beta_t^2\left\|h_{t,n}^g\right\|^2 - 2\beta_t\left\langle h_{t,n}^g, \mathbf{y}_t^n - \mathbf{y}^*\left(\mathbf{x}_t^n\right)\right\rangle\right]$$

$$\overset{(a)}{=} \mathbb{E}\left[\left\|\mathbf{y}_t^n - \mathbf{y}^*\left(\mathbf{x}_t^n\right)\right\|^2 + \beta_t^2\left\|h_{t,n}^g\right\|^2 - 2\beta_t\left\langle \nabla_{\mathbf{y}}g\left(\mathbf{x}_t^n, \mathbf{y}_t^n\right), \mathbf{y}_t^n - \mathbf{y}^*\left(\mathbf{x}_t^n\right)\right\rangle\right]$$

$$\overset{(b)}{\leq} \mathbb{E}\left[\left\|\mathbf{y}_t^n - \mathbf{y}^*\left(\mathbf{x}_t^n\right)\right\|^2 + \beta_t^2\left\|h_{t,n}^g\right\|^2 - 2\beta_t\mu_g\left\|\mathbf{y}_t^n - \mathbf{y}^*\left(\mathbf{x}_t^n\right)\right\|^2\right], \tag{16}$$

where (a) results from the unbiasedness of the stochastic gradient $h_{t,n}^g$ (see Assumption 5.3). (b) is due to $\mu_g$-strongly convexity of the lower-level function $g(\mathbf{x}, \mathbf{y})$ (see Assumption 5.2).

By substituting (16) into (15), we get

$$\mathbb{E}\left[\left\|\mathbf{y}_t^{n+1} - \mathbf{y}^*\left(\mathbf{x}_t^{n+1}\right)\right\|^2\right]$$

$$\leq \mathbb{E}\left[(1 + c_1)(1 - 2\beta_t\mu_g)\left\|\mathbf{y}_t^n - \mathbf{y}^*\left(\mathbf{x}_t^n\right)\right\|^2 + (1 + c_1)\beta_t^2\left\|h_{t,n}^g\right\|^2 + \left(1 + \frac{1}{c_1}\right)L_y^2\alpha_t^2\left\|\bar{h}_{t,n}^f\right\|^2\right]$$

$$\overset{(a)}{\leq} \mathbb{E}\left[(1 + c_1)(1 - 2\beta_t\mu_g)\left\|\mathbf{y}_t^n - \mathbf{y}^*\left(\mathbf{x}_t^n\right)\right\|^2 + \left(1 + \frac{1}{c_1}\right)L_y^2\alpha_t^2\left\|\bar{h}_{t,n}^f\right\|^2\right.$$

$$\left. +2(1 + c_1)\beta_t^2 L_g^2\left\|\mathbf{y}_t^n - \mathbf{y}^*\left(\mathbf{x}_t^n\right)\right\|^2 + 2(1 + c_1)\beta_t^2\sigma_{g_y}^2\right]$$

$$\overset{(b)}{\leq} \mathbb{E}\left[(1 + c_1)(1 - \beta_t\mu_g)\left\|\mathbf{y}_t^n - \mathbf{y}^*\left(\mathbf{x}_t^n\right)\right\|^2 + \left(1 + \frac{1}{c_1}\right)L_y^2\alpha_t^2\left\|\bar{h}_{t,n}^f\right\|^2 + 2(1 + c_1)\beta_t^2\sigma_{g_y}^2\right],$$

where (a) uses Lemma E.1, and (b) holds due to the choice $\beta_t \leq \frac{\mu_g}{2L_g^2}$.

Let $c_1 = \frac{\beta_t\mu_g}{2 - 2\beta_t\mu_g}$ and choose $\beta_t \leq \frac{2}{3\mu_g}$. This completes the proof. □

### E.2.5 DESCENT IN THE APPROXIMATION ERROR OF $\mathbf{z}^*\left(\mathbf{x}, \mathbf{y}^*(\mathbf{x})\right)$

**Lemma E.5.** *Under Assumptions 5.1–5.3, the following inequality of the approximation error of* $\mathbf{z}^*\left(\mathbf{x}, \mathbf{y}^*(\mathbf{x})\right)$ *holds for Algorithm 2:*

$$\mathbb{E}\left[\left\|\mathbf{z}_{t+1} - \mathbf{z}_{t+1}^*\right\|^2\right]$$

$$\leq \left(1 - \frac{\gamma_t \mu_g}{2}\right) \mathbb{E}\left[\left\|\mathbf{z}_t - \mathbf{z}_t^*\right\|^2\right] + \frac{2}{\gamma_t \mu_g} L_z^2 \alpha_t^2 N \sum_{n=0}^{N-1} \mathbb{E}\left[\left\|\bar{h}_{t,n}^f\right\|^2\right] + 16\sigma_{g_{yy}}^2 \frac{B_{f_y}^2}{\mu_g^2}\gamma_t^2 + 8\sigma_{f_y}^2 \gamma_t^2,$$

*for all* $t \in \{0, 1, \dots, T-1\}$ *and* $n \in \{0, 1, \dots, N-1\}$, *where* $\mathbf{z}_t^* = \mathbf{z}^*\left(\mathbf{x}_t^0, \mathbf{y}^*(\mathbf{x}_t^0)\right)$. *The expectation is taken over the stochasticity of the algorithm.*

*Proof.* We have

$$\mathbb{E}\left[\left\|\mathbf{z}_{t+1} - \mathbf{z}_{t+1}^*\right\|^2\right]$$

$$\overset{(a)}{\leq} \mathbb{E}\left[(1+c_2)\left\|\mathbf{z}_{t+1} - \mathbf{z}_t^*\right\|^2 + \left(1 + \frac{1}{c_2}\right)\left\|\mathbf{z}^*\left(\mathbf{x}_{t+1}^0, \mathbf{y}^*(\mathbf{x}_{t+1}^0)\right) - \mathbf{z}^*\left(\mathbf{x}_t^0, \mathbf{y}^*(\mathbf{x}_t^0)\right)\right\|^2\right]$$

$$\overset{(b)}{\leq} \mathbb{E}\left[(1+c_2)\left\|\mathbf{z}_{t+1} - \mathbf{z}_t^*\right\|^2 + \left(1 + \frac{1}{c_2}\right)L_z^2\left\|\mathbf{x}_{t+1}^0 - \mathbf{x}_t^0\right\|^2\right]$$

$$\overset{(c)}{\leq} \mathbb{E}\left[(1+c_2)\left\|\mathbf{z}_{t+1} - \mathbf{z}_t^*\right\|^2 + \left(1 + \frac{1}{c_2}\right)L_z^2\alpha_t^2 N \sum_{n=0}^{N-1}\left\|\bar{h}_{t,n}^f\right\|^2\right], \tag{17}$$

where (a) follows from Young's inequality. (b) is due to the Lipschitzness of $\mathbf{z}^*\left(\cdot, \cdot\right)$ (see Lemma D.3). (c) is because of Eq. (11).

Next, we bound the first term on the right:

$$\mathbb{E}\left[\left\|\mathbf{z}_{t+1} - \mathbf{z}_t^*\right\|^2\right]$$

$$= \mathbb{E}\left[\left\|\mathbf{z}_t - \mathbf{z}_t^*\right\|^2 + \gamma_t^2\left\|h_t^q\right\|^2 - 2\gamma_t\left\langle h_t^q, \mathbf{z}_t - \mathbf{z}_t^*\right\rangle\right]$$

$$\overset{(a)}{=} \mathbb{E}\left[\left\|\mathbf{z}_t - \mathbf{z}_t^*\right\|^2 + \gamma_t^2\left\|h_t^q\right\|^2 - 2\gamma_t\left\langle \nabla_{\mathbf{z}}q\left(\mathbf{x}_t^0, \mathbf{y}_t^0, \mathbf{z}_t\right), \mathbf{z}_t - \mathbf{z}_t^*\right\rangle\right]$$

$$\overset{(b)}{\leq} \mathbb{E}\left[\left\|\mathbf{z}_t - \mathbf{z}_t^*\right\|^2 + \gamma_t^2\left\|h_t^q\right\|^2 - 2\gamma_t\mu_g\left\|\mathbf{z}_t - \mathbf{z}_t^*\right\|^2\right] \tag{18}$$

where (a) results from the unbiasedness of the stochastic gradient $h_t^q$ (see Assumption 5.3), and (b) uses $\mu_g$-strongly convexity of $q\left(\mathbf{x}, \mathbf{y}, \mathbf{z}\right)$ (see Lemma D.4).

To bound $\mathbb{E}\left[\left\|h_t^q\right\|^2\right]$ in Eq. (18), we have

$$\mathbb{E}\left[\left\|h_t^q\right\|^2\right] \leq \mathbb{E}\left[2\left\|h_t^q - \nabla_{\mathbf{z}}q\left(\mathbf{x}_t^0, \mathbf{y}_t^0, \mathbf{z}_t\right)\right\|^2 + 2\left\|\nabla_{\mathbf{z}}q\left(\mathbf{x}_t^0, \mathbf{y}_t^0, \mathbf{z}_t\right)\right\|^2\right]$$

$$\overset{(a)}{=} \mathbb{E}\left[2\left\|h_t^q - \nabla_{\mathbf{z}}q\left(\mathbf{x}_t^0, \mathbf{y}_t^0, \mathbf{z}_t\right)\right\|^2 + 2\left\|\nabla_{\mathbf{z}}q\left(\mathbf{x}_t^0, \mathbf{y}_t^0, \mathbf{z}_t\right) - \nabla_{\mathbf{z}}q\left(\mathbf{x}_t^0, \mathbf{y}_t^0, \mathbf{z}_t^*\right)\right\|^2\right]$$

$$\overset{(b)}{=} \mathbb{E}\left[2\left\|h_t^q - \nabla_{\mathbf{z}}q\left(\mathbf{x}_t^0, \mathbf{y}_t^0, \mathbf{z}_t\right)\right\|^2 + 2\left\|\nabla_{\mathbf{yy}}^2 g\left(\mathbf{x}_t^0, \mathbf{y}_t^0\right)\right\|^2\left\|\mathbf{z}_t - \mathbf{z}_t^*\right\|^2\right]$$

$$\overset{(c)}{\leq} \mathbb{E}\left[2\left\|h_t^q - \nabla_{\mathbf{z}}q\left(\mathbf{x}_t^0, \mathbf{y}_t^0, \mathbf{z}_t\right)\right\|^2 + 2B_{g_{yy}}^2\left\|\mathbf{z}_t - \mathbf{z}_t^*\right\|^2\right], \tag{19}$$

where (a) is because of $\nabla_{\mathbf{z}}q\left(\mathbf{x}_t^0, \mathbf{y}_t^0, \mathbf{z}_t^*\right) = 0$. (b) follows from the definition of $\nabla_{\mathbf{z}}q\left(\mathbf{x}, \mathbf{y}, \mathbf{z}\right)$, and (c) results from Assumption 5.2.

Then, we bound the first term on the right in Eq. (19) as follows:

$$\mathbb{E}\left[\left\|\nabla_{\mathbf{z}}q\left(\mathbf{x}_t^0, \mathbf{y}_t^0, \mathbf{z}_t\right) - h_t^q\right\|^2\right]$$

$$\overset{(a)}{=} \mathbb{E}\left[\left\|\nabla_{\mathbf{yy}}^2 g\left(\mathbf{x}_t^0, \mathbf{y}_t^0\right)\mathbf{z}_t + \nabla_{\mathbf{y}}f\left(\mathbf{x}_t^0, \mathbf{y}_t^0\right) - \left(\nabla_{\mathbf{yy}}^2 g\left(\mathbf{x}_t^0, \mathbf{y}_t^0; \mathcal{D}_t^{g_{yy}}\right)\mathbf{z}_t + \nabla_{\mathbf{y}}f\left(\mathbf{x}_t^0, \mathbf{y}_t^0; \mathcal{D}_t^{f_y}\right)\right)\right\|^2\right]$$

$$\overset{(b)}{\leq} \mathbb{E}\left[2\left\|\mathbf{z}_t\right\|^2 \left\|\nabla_{\mathbf{yy}}^2 g\left(\mathbf{x}_t^0, \mathbf{y}_t^0\right) - \nabla_{\mathbf{yy}}^2 g\left(\mathbf{x}_t^0, \mathbf{y}_t^0; \mathcal{D}_t^{g_{yy}}\right)\right\|^2\right.$$

$$\left. +2\left\|\nabla_{\mathbf{y}} f\left(\mathbf{x}_t^0, \mathbf{y}_t^0\right) - \nabla_{\mathbf{y}} f\left(\mathbf{x}_t^0, \mathbf{y}_t^0; \mathcal{D}_t^{f_y}\right)\right\|^2\right]$$

$$\overset{(c)}{\leq} \mathbb{E}\left[2\sigma_{g_{yy}}^2 \left\|\mathbf{z}_t - \mathbf{z}_t^* + \mathbf{z}_t^*\right\|^2 + 2\sigma_{f_y}^2\right]$$

$$\overset{(d)}{\leq} \mathbb{E}\left[4\sigma_{g_{yy}}^2 \left\|\mathbf{z}_t - \mathbf{z}_t^*\right\|^2 + 4\sigma_{g_{yy}}^2 \left\|\mathbf{z}_t^*\right\|^2 + 2\sigma_{f_y}^2\right]$$

$$\overset{(e)}{\leq} 4\sigma_{g_{yy}}^2 \mathbb{E}\left[\left\|\mathbf{z}_t - \mathbf{z}_t^*\right\|^2\right] + 4\sigma_{g_{yy}}^2 \frac{B_{f_y}^2}{\mu_g^2} + 2\sigma_{f_y}^2, \tag{20}$$

where (a) follows from the definitions of $h_t^q$ and $\nabla_{\mathbf{z}} q\left(\mathbf{x}, \mathbf{y}, \mathbf{z}\right)$. (b) and (d) are because of $\left\|x + y\right\|^2 \leq 2\left\|x\right\|^2 + 2\left\|y\right\|^2$. (c) results from the bounded variances in Assumption 5.3. (e) utilizes the bound of $\mathbf{z}^*\left(\mathbf{x}, \mathbf{y}\right)$ in Lemma D.3.

Substituting Eq.(20) into Eq.(19), we get

$$\mathbb{E}\left[\left\|h_t^q\right\|^2\right] \leq \left(8\sigma_{g_{yy}}^2 + 2B_{g_{yy}}^2\right) \mathbb{E}\left[\left\|\mathbf{z}_t - \mathbf{z}_t^*\right\|^2\right] + 8\sigma_{g_{yy}}^2 \frac{B_{f_y}^2}{\mu_g^2} + 4\sigma_{f_y}^2. \tag{21}$$

Substituting (21) in (18) and then substituting the result in (17), we get

$$\mathbb{E}\left[\left\|\mathbf{z}_{t+1} - \mathbf{z}_{t+1}^*\right\|^2\right]$$

$$\leq \mathbb{E}\left[(1 + c_2)(1 - 2\gamma_t \mu_g)\left\|\mathbf{z}_t - \mathbf{z}_t^*\right\|^2 + \left(1 + \frac{1}{c_2}\right) L_z^2 \alpha_t^2 N \sum_{n=0}^{N-1}\left\|\bar{h}_{t,n}^f\right\|^2\right.$$

$$\left. +(1 + c_2)\gamma_t^2\left(8\sigma_{g_{yy}}^2 + 2B_{g_{yy}}^2\right)\left\|\mathbf{z}_t - \mathbf{z}_t^*\right\|^2 + 8(1 + c_2)\gamma_t^2 \sigma_{g_{yy}}^2 \frac{B_{f_y}^2}{\mu_g^2} + 4(1 + c_2)\gamma_t^2 \sigma_{f_y}^2\right]$$

$$\overset{(a)}{\leq} \mathbb{E}\left[(1 + c_2)(1 - \gamma_t \mu_g)\left\|\mathbf{z}_t - \mathbf{z}_t^*\right\|^2 + \left(1 + \frac{1}{c_2}\right) L_z^2 \alpha_t^2 N \sum_{n=0}^{N-1}\left\|\bar{h}_{t,n}^f\right\|^2\right]$$

$$+8(1 + c_2)\gamma_t^2 \sigma_{g_{yy}}^2 \frac{B_{f_y}^2}{\mu_g^2} + 4(1 + c_2)\gamma_t^2 \sigma_{f_y}^2,$$

where (a) follows from the choice $\gamma_t \leq \frac{\mu_g}{8\sigma_{g_{yy}}^2 + 2B_{g_{yy}}^2}$.

Let $c_2 = \frac{\gamma_t \mu_g}{2 - 2\gamma_t \mu_g}$ and choose $\gamma_t \leq \frac{2}{3\mu_g}$. This completes the proof. $\qquad\square$

### E.2.6 DESCENT IN THE POTENTIAL FUNCTION

We define the potential function $W_t$ as follows:

$$W_t = \ell\left(\mathbf{x}_t^0\right) + K_y\left\|\mathbf{y}_t^0 - \mathbf{y}^*\left(\mathbf{x}_t^0\right)\right\|^2 + K_z\left\|\mathbf{z}_t - \mathbf{z}^*\left(\mathbf{x}_t^0, \mathbf{y}^*(\mathbf{x}_t^0)\right)\right\|^2 + K_h\left\|\nabla\ell\left(\mathbf{x}_t^0\right) - \bar{h}_{t,0}^f\right\|^2.$$

**Lemma E.6.** *Under the same conditions as described in Theorem E.7 and using Lemmas E.2-E.5, the iterates generated by Algorithm 2 satisfies: for all $t \in \{0, 1, \ldots, T-1\}$,*

$$\mathbb{E}\left[W_{t+1} - W_t\right] \leq -\frac{\alpha_t}{2}\sum_{n=0}^{N-1}\mathbb{E}\left[\left\|\nabla\ell\left(\mathbf{x}_t^n\right)\right\|^2\right] + 4B_z^2 \sigma_{g_{xy}}^2 c_\mu^2 \alpha_t^2 N K_h + 16L_{g_{xy}}^2 B_z^2 c_\beta^2 N^3 \sigma_{g_y}^2 c_\mu^2 \alpha_t^4 K_h$$

$$+2\sigma_{f_x}^2 c_\mu^2 \alpha_t^2 N K_h + 4c_\beta^2 \alpha_t^2 \sigma_{g_y}^2 N K_y + 16\sigma_{g_{yy}}^2 \frac{B_{f_y}^2}{\mu_g^2} c_\gamma^2 \alpha_t^2 K_z + 8\sigma_{f_y}^2 c_\gamma^2 \alpha_t^2 K_z,$$

*where $K_y = \frac{\sqrt{2}L_f}{2L_y}$, $K_z = \frac{\sqrt{2}L_f}{2L_z}$, and $K_h = \frac{1}{8L_l}$.*

*Proof.* From Lemma E.2, we have

$$\sum_{n=0}^{N-1} \mathbb{E}\left[\ell\left(\mathbf{x}_t^{n+1}\right) - \ell\left(\mathbf{x}_t^n\right)\right] = \mathbb{E}\left[\ell\left(\mathbf{x}_{t+1}^0\right) - \ell\left(\mathbf{x}_t^0\right)\right]$$

$$\leq -\frac{\alpha_t}{2}\sum_{n=0}^{N-1}\mathbb{E}\left[\|\nabla\ell\left(\mathbf{x}_t^n\right)\|^2\right] - \frac{\alpha_t}{2}\sum_{n=0}^{N-1}\mathbb{E}\left[\left\|\bar{h}_{t,n}^f\right\|^2\right] + \frac{\alpha_t}{2}\sum_{n=0}^{N-1}\mathbb{E}\left[\left\|\nabla\ell\left(\mathbf{x}_t^n\right) - \bar{h}_{t,n}^f\right\|^2\right]$$

$$+ \frac{\alpha_t^2 L_l}{2}\sum_{n=0}^{N-1}\mathbb{E}\left[\left\|\bar{h}_{t,n}^f\right\|^2\right]. \tag{22}$$

Based on Lemma E.3, we have

$$\mathbb{E}\left[\left\|\nabla\ell\left(\mathbf{x}_t^{n+1}\right) - \bar{h}_{t,n+1}^f\right\|^2 - \left\|\nabla\ell\left(\mathbf{x}_t^n\right) - \bar{h}_{t,n}^f\right\|^2\right]$$

$$\leq -\mu_t\mathbb{E}\left[\left\|\nabla\ell\left(\mathbf{x}_t^n\right) - \bar{h}_{t,n}^f\right\|^2\right] + 4\mu_t L_f^2\mathbb{E}\left[\|\mathbf{y}_t^n - \mathbf{y}^*\left(\mathbf{x}_t^n\right)\|^2\right] + 8\mu_t L_f^2\mathbb{E}\left[\|\mathbf{z}_t - \mathbf{z}_t^*\|^2\right]$$

$$+ 16L_g^2\mu_t^2 L_{g_{xy}}^2 B_z^2\beta_t^2 n\sum_{i=0}^{n-1}\mathbb{E}\left[\|\mathbf{y}_t^i - \mathbf{y}^*\left(\mathbf{x}_t^i\right)\|^2\right] + \frac{2}{\mu_t}L_l^2\alpha_t^2\mathbb{E}\left[\left\|\bar{h}_{t,n}^f\right\|^2\right]$$

$$+ 8\mu_t^2 L_{g_{xy}}^2 B_z^2\alpha_t^2 n\sum_{i=0}^{n-1}\mathbb{E}\left[\left\|\bar{h}_{t,i}^f\right\|^2\right] + 8\mu_t L_f^2 L_z^2\alpha_t^2 n\sum_{i=0}^{n-1}\mathbb{E}\left[\left\|\bar{h}_{t,i}^f\right\|^2\right]$$

$$+ 4B_z^2\sigma_{g_{xy}}^2\mu_t^2 + 2\sigma_{f_x}^2\mu_t^2 + 16L_{g_{xy}}^2 B_z^2\beta_t^2 n^2\sigma_{g_y}^2\mu_t^2.$$

This implies that

$$\sum_{n=0}^{N-1}\mathbb{E}\left[\left\|\nabla\ell\left(\mathbf{x}_t^{n+1}\right) - \bar{h}_{t,n+1}^f\right\|^2 - \left\|\nabla\ell\left(\mathbf{x}_t^n\right) - \bar{h}_{t,n}^f\right\|^2\right]$$

$$= \mathbb{E}\left[\left\|\nabla\ell\left(\mathbf{x}_{t+1}^0\right) - \bar{h}_{t+1,0}^f\right\|^2 - \left\|\nabla\ell\left(\mathbf{x}_t^0\right) - \bar{h}_{t,0}^f\right\|^2\right]$$

$$\leq -\mu_t\sum_{n=0}^{N-1}\mathbb{E}\left[\left\|\nabla\ell\left(\mathbf{x}_t^n\right) - \bar{h}_{t,n}^f\right\|^2\right] + 4\mu_t L_f^2\sum_{n=0}^{N-1}\mathbb{E}\left[\|\mathbf{y}_t^n - \mathbf{y}^*\left(\mathbf{x}_t^n\right)\|^2\right]$$

$$+ 16L_g^2\mu_t^2 L_{g_{xy}}^2 B_z^2\beta_t^2 N^2\sum_{n=0}^{N-1}\mathbb{E}\left[\|\mathbf{y}_t^n - \mathbf{y}^*\left(\mathbf{x}_t^n\right)\|^2\right] + \frac{2}{\mu_t}L_l^2\alpha_t^2\sum_{n=0}^{N-1}\mathbb{E}\left[\left\|\bar{h}_{t,n}^f\right\|^2\right]$$

$$+ 8\mu_t^2 L_{g_{xy}}^2 B_z^2\alpha_t^2 N^2\sum_{n=0}^{N-1}\mathbb{E}\left[\left\|\bar{h}_{t,n}^f\right\|^2\right] + 8\mu_t L_f^2 L_z^2\alpha_t^2 N^2\sum_{n=0}^{N-1}\mathbb{E}\left[\left\|\bar{h}_{t,n}^f\right\|^2\right]$$

$$+ 8\mu_t L_f^2 N\mathbb{E}\left[\|\mathbf{z}_t - \mathbf{z}_t^*\|^2\right] + 4B_z^2\sigma_{g_{xy}}^2\mu_t^2 N + 2\sigma_{f_x}^2\mu_t^2 N + 16L_{g_{xy}}^2 B_z^2\beta_t^2 N^3\sigma_{g_y}^2\mu_t^2. \tag{23}$$

With the result from Lemma E.4, we have

$$\sum_{n=0}^{N-1}\mathbb{E}\left[\left\|\mathbf{y}_t^{n+1} - \mathbf{y}^*\left(\mathbf{x}_t^{n+1}\right)\right\|^2 - \|\mathbf{y}_t^n - \mathbf{y}^*\left(\mathbf{x}_t^n\right)\|^2\right]$$

$$= \mathbb{E}\left[\left\|\mathbf{y}_{t+1}^0 - \mathbf{y}^*\left(\mathbf{x}_{t+1}^0\right)\right\|^2 - \left\|\mathbf{y}_t^0 - \mathbf{y}^*\left(\mathbf{x}_t^0\right)\right\|^2\right]$$

$$\leq -\frac{\beta_t\mu_g}{2}\sum_{n=0}^{N-1}\mathbb{E}\left[\|\mathbf{y}_t^n - \mathbf{y}^*\left(\mathbf{x}_t^n\right)\|^2\right] + \frac{2}{\beta_t\mu_g}L_y^2\alpha_t^2\sum_{n=0}^{N-1}\mathbb{E}\left[\left\|\bar{h}_{t,n}^f\right\|^2\right] + 4\beta_t^2\sigma_{g_y}^2 N. \tag{24}$$

According to Lemma E.5, we have

$$\mathbb{E}\left[\left\|\mathbf{z}_{t+1} - \mathbf{z}_{t+1}^*\right\|^2 - \|\mathbf{z}_t - \mathbf{z}_t^*\|^2\right]$$

$$= \mathbb{E}\left[\left\|\mathbf{z}_{t+1} - \mathbf{z}^*\left(\mathbf{x}_{t+1}^0, \mathbf{y}^*(\mathbf{x}_{t+1}^0)\right)\right\|^2 - \left\|\mathbf{z}_t - \mathbf{z}^*\left(\mathbf{x}_t^0, \mathbf{y}^*(\mathbf{x}_t^0)\right)\right\|^2\right]$$

$$\leq -\frac{\gamma_t \mu_g}{2}\mathbb{E}\left[\|\mathbf{z}_t - \mathbf{z}_t^*\|^2\right] + \frac{2}{\gamma_t \mu_g}L_z^2\alpha_t^2 N \sum_{n=0}^{N-1}\mathbb{E}\left[\left\|\bar{h}_{t,n}^f\right\|^2\right] + 16\sigma_{g_{yy}}^2\frac{B_{f_y}^2}{\mu_g^2}\gamma_t^2 + 8\sigma_{f_y}^2\gamma_t^2. \quad (25)$$

Adding Eq. (22), (23), (24) and (25), we get

$$\mathbb{E}\left[W_{t+1} - W_t\right]$$

$$\leq -\frac{\alpha_t}{2}\sum_{n=0}^{N-1}\mathbb{E}\left[\|\nabla l\left(\mathbf{x}_t^n\right)\|^2\right] + C_y\sum_{n=0}^{N-1}\mathbb{E}\left[\|\mathbf{y}_t^n - \mathbf{y}^*\left(\mathbf{x}_t^n\right)\|^2\right] + C_z\mathbb{E}\left[\|\mathbf{z}_t - \mathbf{z}_t^*\|^2\right]$$

$$+ C_h\sum_{n=0}^{N-1}\mathbb{E}\left[\left\|\bar{h}_{t,n}^f\right\|^2\right] + C_l\sum_{n=0}^{N-1}\mathbb{E}\left[\left\|\nabla\ell\left(\mathbf{x}_t^n\right) - \bar{h}_{t,n}^f\right\|^2\right]$$

$$+ 4B_z^2\sigma_{g_{xy}}^2\mu_t^2 N K_h + 2\sigma_{f_x}^2\mu_t^2 N K_h + 16L_{g_{xy}}^2 B_z^2\beta_t^2 N^3\sigma_{g_y}^2\mu_t^2 K_h + 4\beta_t^2\sigma_{g_y}^2 N K_y$$

$$+ 16\sigma_{g_{yy}}^2\frac{B_{f_y}^2}{\mu_g^2}\gamma_t^2 K_z + 8\sigma_{f_y}^2\gamma_t^2 K_z,$$

where

$$C_y = 4\mu_t L_f^2 K_h + 16L_g^2\mu_t^2 L_{g_{xy}}^2 B_z^2\beta_t^2 N^2 K_h - \frac{\beta_t\mu_g}{2}K_y,$$

$$C_z = 8\mu_t L_f^2 N K_h - \frac{\gamma_t\mu_g}{2}K_z,$$

$$C_h = -\frac{\alpha_t}{2} + \frac{\alpha_t^2 L_l}{2} + \frac{2}{\mu_t}L_l^2\alpha_t^2 K_h + 8\mu_t^2 L_{g_{xy}}^2 B_z^2\alpha_t^2 N^2 K_h + 8\mu_t L_f^2 L_z^2\alpha_t^2 N^2 K_h$$

$$\quad + \frac{2}{\beta_t\mu_g}L_y^2\alpha_t^2 K_y + \frac{2}{\gamma_t\mu_g}L_z^2\alpha_t^2 N K_z,$$

$$C_l = \frac{\alpha_t}{2} - \mu_t K_h.$$

Define $\beta_t \triangleq c_\beta\alpha_t$, $\gamma_t \triangleq c_\gamma\alpha_t$, and $\mu_t \triangleq c_\mu\alpha_t$.

To ensure $C_y \leq 0$, we have

$$C_y = 4\mu_t L_f^2 K_h + 16L_g^2\mu_t^2 L_{g_{xy}}^2 B_z^2\beta_t^2 N^2 K_h - \frac{\beta_t\mu_g}{2}K_y$$

$$\overset{(a)}{\leq} \frac{c_\beta\alpha_t\mu_g}{4}K_y + \frac{c_\beta\alpha_t\mu_g}{4}K_y - \frac{c_\beta\alpha_t\mu_g}{2}K_y = 0,$$

where (a) uses $K_y = \frac{16c_\mu L_f^2 K_h}{\mu_g c_\beta}$ and $\alpha_t \leq 4\sqrt[3]{\frac{\mu_g K_y}{L_g^2 c_\mu^2 L_{g_{xy}}^2 B_z^2 c_\beta N^2 K_h}}$.

To ensure $C_z \leq 0$, we have

$$C_z = 8\mu_t L_f^2 N K_h - \frac{\gamma_t\mu_g}{2}K_z \overset{(a)}{\leq} \frac{c_\gamma\alpha_t\mu_g}{2}K_z - \frac{c_\gamma\alpha_t\mu_g}{2}K_z = 0,$$

where (a) utilizes $K_z = \frac{16c_\mu L_f^2 N K_h}{\mu_g c_\gamma}$.

To ensure $C_h \leq 0$, we have

$$C_h = -\frac{\alpha_t}{2} + \frac{\alpha_t^2 L_l}{2} + \frac{2}{\mu_t}L_l^2\alpha_t^2 K_h + 8\mu_t^2 L_{g_{xy}}^2 B_z^2\alpha_t^2 N^2 K_h + 8\mu_t L_f^2 L_z^2\alpha_t^2 N^2 K_h$$

$$\quad + \frac{2}{\beta_t\mu_g}L_y^2\alpha_t^2 K_y + \frac{2}{\gamma_t\mu_g}L_z^2\alpha_t^2 N K_z$$

$$\overset{(a)}{\leq} -\frac{\alpha_t}{2} + \frac{\alpha_t^2 L_l}{2} + \frac{2}{c_\mu}L_l^2\alpha_t K_h + 16c_\mu L_f^2 L_z^2\alpha_t^3 N^2 K_h + \frac{2}{c_\beta\mu_g}L_y^2\alpha_t K_y + \frac{2}{c_\gamma\mu_g}L_z^2\alpha_t N K_z$$

$$\overset{(b)}{\le} -\frac{\alpha_t}{2} + \frac{\alpha_t^2 L_l}{2} + \frac{4}{c_\mu} L_l^2 \alpha_t K_h + \frac{2}{c_\beta \mu_g} L_y^2 \alpha_t K_y + \frac{2}{c_\gamma \mu_g} L_z^2 \alpha_t N K_z$$

$$\overset{(c)}{\le} -\frac{\alpha_t}{2} + \frac{\alpha_t}{8} + \frac{\alpha_t}{8} + \frac{\alpha_t}{8} + \frac{\alpha_t}{8} = 0,$$

where (a) results from $\alpha_t \le \frac{L_f^2 L_z^2}{c_{\mu L} L_{g_{xy}}^2 B_z^2}$. (b) is because of $\alpha_t \le \frac{L_l}{2\sqrt{2} c_\mu L_f L_z N}$. (c) follows from

$\alpha_t \le \frac{1}{4L_l}$, $c_\mu = 32 L_l^2 K_h$, $c_\beta = \frac{16 L_y^2 K_y}{\mu_g}$, and $c_\gamma = \frac{16 L_z^2 N K_z}{\mu_g}$.

To ensure $C_l \le 0$, we have

$$C_l = \frac{\alpha_t}{2} - \mu_t K_h \overset{(a)}{\le} 0,$$

where (a) is due to $K_h = \frac{1}{2c_\mu}$.

Towards this end, the lemma is proved. $\qquad\square$

### E.2.7 PROOF OF THEOREM 5.5

**Theorem E.7** (Non-Convex $\ell(\mathbf{x})$). *Under Assumptions 5.1–5.3, choose constant step-sizes $\alpha_t = \alpha = \frac{\bar\alpha}{N\sqrt{T}}$, $\beta_t = \beta \triangleq c_\beta \alpha$, $\gamma_t = \gamma \triangleq c_\gamma \alpha$, and the momentum coefficient as $\mu_t = \mu \triangleq c_\mu \alpha$ for all $t \in \{0, 1, \ldots, T\}$ with $c_\beta = \frac{16 L_y L_f}{\sqrt{2}\mu_g}$, $c_\gamma = \frac{16 L_z L_f N}{\sqrt{2}\mu_g}$, and $c_\mu = 4L_l$. Moreover, choose $\bar\alpha$ such that*

$$\bar\alpha \le \min\left\{ \frac{\mu_g}{2L_g^2 c_\beta}, \frac{2}{3\mu_g c_\beta}, \frac{\mu_g}{\left(8\sigma_{g_{yy}}^2 + 2B_{g_{yy}}^2\right)c_\gamma}, \frac{2}{3\mu_g c_\gamma}, \frac{1}{4L_l}, \frac{L_f^2 L_z^2}{c_\mu L_{g_{xy}}^2 B_z^2}, \right.$$

$$\left. \frac{L_l}{2\sqrt{2}c_\mu L_f L_z N}, \sqrt[3]{\frac{32\mu_g^2 L_l}{L_g^2 c_\mu^2 L_{g_{xy}}^2 B_z^2 L_y^2 N^2}}, \frac{\sqrt{K_y}}{2\sqrt{K_h} L_{g_{xy}} B_z N c_\mu} \right\}.$$

*Then, the iterates generated by SO-Lazy-BiO-I in Algorithm 2 satisfy:*

$$\frac{1}{TN} \sum_{t=0}^{T-1} \sum_{n=0}^{N-1} \mathbb{E}\left[\|\nabla\ell(\mathbf{x}_t^n)\|^2\right] = \mathcal{O}\left(\frac{\Delta_0}{\sqrt{T}} + \frac{\sigma_{g_y}^2}{N\sqrt{T}} + \frac{\sigma_{g_{yy}}^2}{\sqrt{T}} + \frac{\sigma_{f_y}^2}{\sqrt{T}} + \frac{\sigma_{g_{xy}}^2}{N\sqrt{T}} + \frac{\sigma_{f_x}^2}{N\sqrt{T}}\right),$$

*where $\Delta_0 = (\ell(\mathbf{x}_0^0) - \ell^*) + \|\mathbf{y}_0^0 - \mathbf{y}^*(\mathbf{x}_0^0)\|^2 + \|\mathbf{z}_0 - \mathbf{z}^*(\mathbf{x}_0^0, \mathbf{y}^*(\mathbf{x}_0^0))\|^2$.*

*Proof.* Choose $\alpha_t$ as a constant stepsize $\alpha_t = \alpha$. Summing the result in Lemma E.6 from $t = 0$ to $T - 1$, and then dividing by $NT$ on both sides, we get

$$\frac{\mathbb{E}[W_T - W_0]}{NT} \le -\frac{\alpha}{2NT} \sum_{t=0}^{T-1} \sum_{n=0}^{N-1} \mathbb{E}\left[\|\nabla\ell(\mathbf{x}_t^n)\|^2\right] + 4B_z^2 \sigma_{g_{xy}}^2 c_\mu^2 \alpha^2 K_h + 2\sigma_{f_x}^2 c_\mu^2 \alpha^2 K_h$$

$$+ 16 L_{g_{xy}}^2 B_z^2 c_\beta^2 N^2 \sigma_{g_y}^2 c_\mu^2 \alpha^4 K_h + 4 c_\beta^2 \alpha^2 \sigma_{g_y}^2 K_y + 16\sigma_{g_{yy}}^2 \frac{B_{f_y}^2}{\mu_g^2} c_\gamma^2 \alpha^2 K_z \frac{1}{N} + 8\sigma_{f_y}^2 c_\gamma^2 \alpha^2 K_z \frac{1}{N}.$$

Rearranging the terms and multiplying by $2/\alpha$ on both sides and let $\alpha \le \frac{\sqrt{K_y}}{2\sqrt{K_h} L_{g_{xy}} B_z N c_\mu}$, we have

$$\frac{1}{TN} \sum_{t=0}^{T-1} \sum_{n=0}^{N-1} \mathbb{E}\left[\|\nabla\ell(\mathbf{x}_t^n)\|^2\right] \le \frac{2(W_0 - \ell^*)}{\alpha NT} + 8B_z^2 \sigma_{g_{xy}}^2 c_\mu^2 \alpha K_h + 4\sigma_{f_x}^2 c_\mu^2 \alpha K_h$$

$$+ 16 c_\beta^2 \alpha \sigma_{g_y}^2 K_y + 32\sigma_{g_{yy}}^2 \frac{B_{f_y}^2}{\mu_g^2} c_\gamma^2 \alpha K_z \frac{1}{N} + 16\sigma_{f_y}^2 c_\gamma^2 \alpha K_z \frac{1}{N},$$

where $W_0 = \ell(\mathbf{x}_0^0) + K_y \|\mathbf{y}_0^0 - \mathbf{y}^*(x_0^0)\|^2 + K_z \|\mathbf{z}_0 - \mathbf{z}^*(\mathbf{x}_0^0, \mathbf{y}^*(\mathbf{x}_0^0))\|^2$.

Therefore,

$$\frac{1}{TN} \sum_{t=0}^{T-1} \sum_{n=0}^{N-1} \mathbb{E}\left[\|\nabla\ell\left(\mathbf{x}_t^n\right)\|^2\right]$$

$$= \mathcal{O}\left(\frac{\ell\left(\mathbf{x}_0^0\right) - \ell^*}{NT\alpha}\right) + \mathcal{O}\left(\frac{\left\|\mathbf{y}_0^0 - \mathbf{y}^*\left(\mathbf{x}_0^0\right)\right\|^2}{NT\alpha}\right) + \mathcal{O}\left(\frac{\left\|\mathbf{z}_0 - \mathbf{z}^*\left(\mathbf{x}_0^0, \mathbf{y}^*(\mathbf{x}_0^0)\right)\right\|^2}{NT\alpha}\right)$$

$$+ \mathcal{O}\left(\sigma_{g_{xy}}^2 \alpha + \sigma_{f_x}^2 \alpha + \sigma_{g_y}^2 \alpha + \sigma_{g_{yy}}^2 N\alpha + \sigma_{f_y}^2 N\alpha\right).$$

By selecting $\alpha = \mathcal{O}\left(\frac{1}{N\sqrt{T}}\right)$, the proof of the theorem is completed. $\qquad\square$

## F    THEORETICAL ANALYSIS OF SO-Lazy-BiO-SGD

### F.1    REFORMULATION OF **OPTION I** IN ALGORITHM 1 WITH VANILLA SGD UPDATES FOR THEORETICAL ANALYSIS

In order to analyze the theoretical performance of SO-Lazy-BiO-SGD, we reformulate SO-Lazy-BiO-SGD as follows. We note that **Option I** in Algorithm 1 with vanilla SGD updates and Algorithm 3 are equivalent when choosing $T$ in Algorithm 3 to be $T/N$.

---

**Algorithm 3** The SO-Lazy-BiO-SGD Algorithm.

**Input:** Initial parameters $\mathbf{x}_0^0, \mathbf{y}_0^0, \mathbf{z}_0$, and stepsizes $\{\alpha_t, \beta_t, \gamma_t\}_{t=0}^{T-1}$
**for** $t = 0$ to $T - 1$ **do**
    Initialize $\mathbf{x}_t^0 = \mathbf{x}_{t-1}^N$ and $\mathbf{y}_t^0 = \mathbf{y}_{t-1}^N$
    Sample data batches $\mathcal{D}_t^{g_{yy}}$ $\mathcal{D}_t^{f_y}$, and $\mathcal{D}_t^{g_{xy}}$
    Compute the gradient estimate $h_t^q$ using $h_t^q = \nabla_{\mathbf{yy}}^2 g(\mathbf{x}_t^0, \mathbf{y}_t^0; \mathcal{D}_t^{g_{yy}})\mathbf{z}_t + \nabla_{\mathbf{y}} f(\mathbf{x}_t^0, \mathbf{y}_t^0; \mathcal{D}_t^{f_y})$
    Update $\mathbf{z}_{t+1} = \mathbf{z}_t - \gamma_t h_t^q$
    Compute the JVP using $\mathbf{v}_t = \nabla_{\mathbf{xy}}^2 g\left(\mathbf{x}_t^0, \mathbf{y}_t^0; \mathcal{D}_t^{g_{xy}}\right) \mathbf{z}_t$
    **for** $n = 0$ to $N - 1$ **do**
        Sample data batches $\mathcal{D}_{t,n}^g, \mathcal{D}_{t,n}^{f_x}$, and $\mathcal{D}_{t,n}^{g_{xy}}$
        Compute the gradient estimate $h_{t,n}^g$ using $h_{t,n}^g = \nabla_{\mathbf{y}} g\left(\mathbf{x}_t^n, \mathbf{y}_t^n; \mathcal{D}_{t,n}^g\right)$
        Update $\mathbf{y}_t^{n+1} = \mathbf{y}_t^n - \beta_t h_{t,n}^g$
        Compute the gradient estimate $h_{t,n}^f$ using $h_{t,n}^f = \nabla_{\mathbf{x}} f\left(\mathbf{x}_t^n, \mathbf{y}_t^n; \mathcal{D}_{t,n}^{f_x}\right) + \mathbf{v}_t$
        Update $\mathbf{x}_t^{n+1} = \mathbf{x}_t^n - \alpha_t h_{t,n}^f$
    **end for**
**end for**

---

### F.2    DETAILED PROOF OF THEOREM 5.7: NON-CONVEX $\ell\left(\mathbf{x}\right)$

#### F.2.1    DESCENT IN THE UPPER-LEVEL OBJECTIVE FUNCTION

**Lemma F.1.** *Under Assumptions 5.1–5.3, the following inequality holds for successive iterations of Algorithm 3:*

$$\mathbb{E}\left[\ell\left(\mathbf{x}_t^{n+1}\right) - \ell\left(\mathbf{x}_t^n\right)\right]$$

$$\leq -\frac{\alpha_t}{2}\mathbb{E}\left[\|\nabla\ell\left(\mathbf{x}_t^n\right)\|^2\right] - \frac{\alpha_t}{2}\mathbb{E}\left[\left\|h_{t,n}^f\right\|^2\right] + \frac{\alpha_t^2 L_l}{2}\mathbb{E}\left[\left\|h_{t,n}^f\right\|^2\right] + 2\alpha_t L_f^2 \mathbb{E}\left[\left\|\mathbf{y}_t^n - \mathbf{y}^*\left(\mathbf{x}_t^n\right)\right\|^2\right]$$

$$+ 4\alpha_t L_f^2 \mathbb{E}\left[\|\mathbf{z}_t - \mathbf{z}_t^*\|^2\right] + 4L_f^2 L_z^2 \alpha_t^3 n \sum_{i=0}^{n-1} \mathbb{E}\left[\left\|h_{t,i}^f\right\|^2\right] + 8L_{g_{xy}}^2 B_z^2 \alpha_t^3 n \sum_{i=0}^{n-1} \mathbb{E}\left[\left\|h_{t,i}^f\right\|^2\right]$$

$$+ 16L_g^2 \alpha_t L_{g_{xy}}^2 B_z^2 \beta_t^2 n \sum_{i=0}^{n-1} \mathbb{E}\left[\left\|\mathbf{y}_t^i - \mathbf{y}^*\left(\mathbf{x}_t^i\right)\right\|^2\right] + 4B_z^2 \sigma_{g_{xy}}^2 \alpha_t + 2\sigma_{f_x}^2 \alpha_t + 16L_{g_{xy}}^2 B_z^2 \beta_t^2 n^2 \sigma_{g_y}^2 \alpha_t,$$

*for all $t \in \{0, 1, \ldots, T-1\}$ and $n \in \{0, 1, \ldots, N-1\}$, where the expectation is taken over the stochasticity of the algorithm.*

*Proof.* We have

$$\mathbb{E}\left[\ell\left(\mathbf{x}_t^{n+1}\right) - \ell\left(\mathbf{x}_t^n\right)\right]$$

$$\overset{(a)}{\leq} \mathbb{E}\left[\left\langle\nabla\ell\left(\mathbf{x}_t^n\right), \mathbf{x}_t^{n+1} - \mathbf{x}_t^n\right\rangle + \frac{L_l}{2}\left\|\mathbf{x}_t^{n+1} - \mathbf{x}_t^n\right\|^2\right]$$

$$\overset{(b)}{=} \mathbb{E}\left[-\alpha_t\left\langle\nabla\ell\left(\mathbf{x}_t^n\right), h_{t,n}^f\right\rangle + \frac{\alpha_t^2 L_l}{2}\left\|h_{t,n}^f\right\|^2\right]$$

$$\overset{(c)}{=} \mathbb{E}\left[-\frac{\alpha_t}{2}\left\|\nabla\ell\left(\mathbf{x}_t^n\right)\right\|^2 - \frac{\alpha_t}{2}\left\|h_{t,n}^f\right\|^2 + \frac{\alpha_t}{2}\left\|\nabla\ell\left(\mathbf{x}_t^n\right) - h_{t,n}^f\right\|^2 + \frac{\alpha_t^2 L_l}{2}\left\|h_{t,n}^f\right\|^2\right], \qquad (26)$$

where (a) uses the Lipschitz continuous gradients of $\ell$ (see Lemma D.1). (b) follows from the update rule of Algorithm 2. (c) is because of $\langle x, y\rangle = \frac{1}{2}\|x\|^2 + \frac{1}{2}\|y\|^2 - \frac{1}{2}\|x - y\|^2$.

To bound the third term on the right-hand side of Eq. (26), we have

$$\mathbb{E}\left[\left\|\nabla\ell\left(\mathbf{x}_t^n\right) - h_{t,n}^f\right\|^2\right]$$

$$\leq \mathbb{E}\left[2\left\|\nabla\ell\left(\mathbf{x}_t^n\right) - \nabla f\left(\mathbf{x}_t^n, \mathbf{y}_t^n, \mathbf{z}_t\right)\right\|^2 + 2\left\|\nabla f\left(\mathbf{x}_t^n, \mathbf{y}_t^n, \mathbf{z}_t\right) - h_{t,n}^f\right\|^2\right]$$

$$\overset{(a)}{\leq} \mathbb{E}\left[2\left\|h_{t,n}^f - \nabla f\left(\mathbf{x}_t^n, \mathbf{y}_t^n, \mathbf{z}_t\right)\right\|^2 + 2L_f^2\left(\|\mathbf{y}_t^n - \mathbf{y}^*(\mathbf{x}_t^n)\| + \|\mathbf{z}_t - \mathbf{z}^*\left(\mathbf{x}_t^n, \mathbf{y}^*(\mathbf{x}_t^n)\right)\|\right)^2\right]$$

$$\leq \mathbb{E}\left[2\left\|h_{t,n}^f - \nabla f\left(\mathbf{x}_t^n, \mathbf{y}_t^n, \mathbf{z}_t\right)\right\|^2 + 4L_f^2\|\mathbf{y}_t^n - \mathbf{y}^*(\mathbf{x}_t^n)\|^2 + 4L_f^2\|\mathbf{z}_t - \mathbf{z}^*\left(\mathbf{x}_t^n, \mathbf{y}^*(\mathbf{x}_t^n)\right)\|^2\right], \qquad (27)$$

where (a) utilizes the Lipschitzness of $\nabla f(\mathbf{x}, \mathbf{y}, \mathbf{z})$ (see Lemma D.2).

Similar to Eq. (13), we bound the term $\mathbb{E}\left[\left\|h_{t,n}^f - \nabla f\left(\mathbf{x}_t^n, \mathbf{y}_t^n, \mathbf{z}_t\right)\right\|^2\right]$ in Eq. (27) and get

$$\mathbb{E}\left[\left\|h_{t,n}^f - \nabla f\left(\mathbf{x}_t^n, \mathbf{y}_t^n, \mathbf{z}_t\right)\right\|^2\right]$$

$$\leq \mathbb{E}\left[8L_{g_{xy}}^2 B_z^2\alpha_t^2 n\sum_{i=0}^{n-1}\left\|h_{t,i}^f\right\|^2 + 16L_g^2 L_{g_{xy}}^2 B_z^2\beta_t^2 n\sum_{i=0}^{n-1}\left\|\mathbf{y}_t^i - \mathbf{y}^*(\mathbf{x}_t^i)\right\|^2\right.$$

$$\left. + 4B_z^2\sigma_{g_{xy}}^2 + 2\sigma_{f_x}^2 + 16L_{g_{xy}}^2 B_z^2\beta_t^2 n^2\sigma_{g_y}^2\right]. \qquad (28)$$

Then, similar to Eq. (14), we bound the term $\mathbb{E}\left[\|\mathbf{z}_t - \mathbf{z}^*\left(\mathbf{x}_t^n, \mathbf{y}^*(\mathbf{x}_t^n)\right)\|^2\right]$ in Eq. (27) and get

$$\mathbb{E}\left[\|\mathbf{z}_t - \mathbf{z}^*\left(\mathbf{x}_t^n, \mathbf{y}^*(\mathbf{x}_t^n)\right)\|^2\right] \leq \mathbb{E}\left[2\left\|\mathbf{z}_t - \mathbf{z}^*\left(\mathbf{x}_t^0, \mathbf{y}^*(\mathbf{x}_t^0)\right)\right\|^2 + 2L_z^2\alpha_t^2 n\sum_{i=0}^{n-1}\left\|h_{t,i}^f\right\|^2\right]. \qquad (29)$$

Combining Eq. (26), (27), (28), and (29) completes the proof of the lemma. $\qquad\square$

### F.2.2 DESCENT IN THE APPROXIMATION ERROR OF $\mathbf{y}^*(\mathbf{x})$

Following the similar proof of Lemma E.4, we get the following lemma:

**Lemma F.2.** *Under Assumptions 5.2 and 5.3, the approximation error of $\mathbf{y}^*(\mathbf{x})$ of Algorithm 3 satisfies the following inequality:*

$$\mathbb{E}\left[\|\mathbf{y}_t^{n+1} - \mathbf{y}^*\left(\mathbf{x}_t^{n+1}\right)\|^2\right]$$

$$\leq \left(1 - \frac{\beta_t\mu_g}{2}\right)\mathbb{E}\left[\|\mathbf{y}_t^n - \mathbf{y}^*(\mathbf{x}_t^n)\|^2\right] + \frac{2}{\beta_t\mu_g}L_y^2\alpha_t^2\mathbb{E}\left[\left\|h_{t,n}^f\right\|^2\right] + 4\beta_t^2\sigma_{g_y}^2,$$

*for all $t\in\{0, 1, \ldots, T-1\}$ and $n\in\{0, 1, \ldots, N-1\}$, where the expectation is taken over the stochasticity of the algorithm.*

### F.2.3 DESCENT IN THE APPROXIMATION ERROR OF $\mathbf{z}^*(\mathbf{x}, \mathbf{y}^*(\mathbf{x}))$

Following the similar proof of Lemma E.5, we obtain the following lemma:

**Lemma F.3.** *Under Assumptions 5.1–5.3, the following inequality of the approximation error of* $\mathbf{z}^*(\mathbf{x}, \mathbf{y}^*(\mathbf{x}))$ *holds for Algorithm 3:*

$$
\mathbb{E}\left[\left\|\mathbf{z}_{t+1} - \mathbf{z}_{t+1}^*\right\|^2\right]
$$

$$
\leq \left(1 - \frac{\gamma_t \mu_g}{2}\right) \mathbb{E}\left[\left\|\mathbf{z}_t - \mathbf{z}_t^*\right\|^2\right] + \frac{2}{\gamma_t \mu_g} L_z^2 \alpha_t^2 N \sum_{n=0}^{N-1} \mathbb{E}\left[\left\|h_{t,n}^f\right\|^2\right] + 16\sigma_{g_{yy}}^2 \frac{B_{f_y}^2}{\mu_g^2} \gamma_t^2 + 8\sigma_{f_y}^2 \gamma_t^2,
$$

*for all* $t \in \{0, 1, \ldots, T-1\}$ *and* $n \in \{0, 1, \ldots, N-1\}$*, where* $\mathbf{z}_t^* = \mathbf{z}^*\left(\mathbf{x}_t^0, \mathbf{y}^*(\mathbf{x}_t^0)\right)$*. The expectation is taken over the stochasticity of the algorithm.*

### F.2.4 DESCENT IN THE POTENTIAL FUNCTION

We define the potential function $\bar{W}_t$ as follows:

$$
\bar{W}_t = \ell\left(\mathbf{x}_t^0\right) + K_y \left\|\mathbf{y}_t^0 - \mathbf{y}^*\left(\mathbf{x}_t^0\right)\right\|^2 + K_z \left\|\mathbf{z}_t - \mathbf{z}^*\left(\mathbf{x}_t^0, \mathbf{y}^*(\mathbf{x}_t^0)\right)\right\|^2.
$$

**Lemma F.4.** *Under the same conditions as described in Theorem F.5 and using Lemmas F.1-F.3, the iterates generated by Algorithm 3 satisfies: for all* $t \in \{0, 1, \ldots, T-1\}$,

$$
\mathbb{E}\left[\bar{W}_{t+1} - \bar{W}_t\right] \leq -\frac{\alpha_t}{2} \sum_{n=0}^{N-1} \mathbb{E}\left[\left\|\nabla\ell\left(\mathbf{x}_t^n\right)\right\|^2\right] + 4B_z^2 \sigma_{g_{xy}}^2 \alpha_t N + 2\sigma_{f_x}^2 \alpha_t N + 8\sigma_{f_y}^2 c_\gamma^2 \alpha_t^2 K_z
$$

$$
+ 16 L_{g_{xy}}^2 B_z^2 c_\beta^2 N^3 \sigma_{g_y}^2 \alpha_t^3 + 4 c_\beta^2 \alpha_t^2 \sigma_{g_y}^2 N K_y + 16\sigma_{g_{yy}}^2 \frac{B_{f_y}^2}{\mu_g^2} c_\gamma^2 \alpha_t^2 K_z,
$$

*where* $K_y = \frac{L_f}{2L_y}$*, and* $K_z = \frac{L_f}{2L_z}$*.*

*Proof.* From Lemma F.1, we have

$$
\sum_{n=0}^{N-1} \mathbb{E}\left[\ell\left(\mathbf{x}_t^{n+1}\right) - \ell\left(\mathbf{x}_t^n\right)\right] = \mathbb{E}\left[\ell\left(\mathbf{x}_{t+1}^0\right) - \ell\left(\mathbf{x}_t^0\right)\right]
$$

$$
\leq -\frac{\alpha_t}{2} \sum_{n=0}^{N-1} \mathbb{E}\left[\left\|\nabla\ell\left(\mathbf{x}_t^n\right)\right\|^2\right] - \frac{\alpha_t}{2} \sum_{n=0}^{N-1} \mathbb{E}\left[\left\|h_{t,n}^f\right\|^2\right] + \frac{\alpha_t^2 L_l}{2} \sum_{n=0}^{N-1} \mathbb{E}\left[\left\|h_{t,n}^f\right\|^2\right]
$$

$$
+ 4\alpha_t L_f^2 N \mathbb{E}\left[\left\|\mathbf{z}_t - \mathbf{z}_t^*\right\|^2\right] + 4L_f^2 L_z^2 \alpha_t^3 N^2 \sum_{n=0}^{N-1} \mathbb{E}\left[\left\|h_{t,n}^f\right\|^2\right] + 8L_{g_{xy}}^2 B_z^2 \alpha_t^3 N^2 \sum_{n=0}^{N-1} \mathbb{E}\left[\left\|h_{t,n}^f\right\|^2\right]
$$

$$
+ 16 L_g^2 \alpha_t L_{g_{xy}}^2 B_z^2 \beta_t^2 N^2 \sum_{n=0}^{N-1} \mathbb{E}\left[\left\|\mathbf{y}_t^n - \mathbf{y}^*\left(\mathbf{x}_t^n\right)\right\|^2\right] + 2\alpha_t L_f^2 \sum_{n=0}^{N-1} \mathbb{E}\left[\left\|\mathbf{y}_t^n - \mathbf{y}^*\left(\mathbf{x}_t^n\right)\right\|^2\right]
$$

$$
+ 4B_z^2 \sigma_{g_{xy}}^2 \alpha_t N + 2\sigma_{f_x}^2 \alpha_t N + 16 L_{g_{xy}}^2 B_z^2 \beta_t^2 N^3 \sigma_{g_y}^2 \alpha_t. \tag{30}
$$

With the result from Lemma F.2, we have

$$
\sum_{n=0}^{N-1} \mathbb{E}\left[\left\|\mathbf{y}_t^{n+1} - \mathbf{y}^*\left(\mathbf{x}_t^{n+1}\right)\right\|^2 - \left\|\mathbf{y}_t^n - \mathbf{y}^*\left(\mathbf{x}_t^n\right)\right\|^2\right]
$$

$$
= \mathbb{E}\left[\left\|\mathbf{y}_{t+1}^0 - \mathbf{y}^*\left(\mathbf{x}_{t+1}^0\right)\right\|^2 - \left\|\mathbf{y}_t^0 - \mathbf{y}^*\left(\mathbf{x}_t^0\right)\right\|^2\right]
$$

$$
\leq -\frac{\beta_t \mu_g}{2} \sum_{n=0}^{N-1} \mathbb{E}\left[\left\|\mathbf{y}_t^n - \mathbf{y}^*\left(\mathbf{x}_t^n\right)\right\|^2\right] + \frac{2}{\beta_t \mu_g} L_y^2 \alpha_t^2 \sum_{n=0}^{N-1} \mathbb{E}\left[\left\|h_{t,n}^f\right\|^2\right] + 4\beta_t^2 \sigma_{g_y}^2 N. \tag{31}
$$

According to Lemma F.3, we have

$$\mathbb{E}\left[\left\|\mathbf{z}_{t+1} - \mathbf{z}_{t+1}^*\right\|^2 - \left\|\mathbf{z}_t - \mathbf{z}_t^*\right\|^2\right]$$

$$= \mathbb{E}\left[\left\|\mathbf{z}_{t+1} - \mathbf{z}^*\left(\mathbf{x}_{t+1}^0, \mathbf{y}^*(\mathbf{x}_{t+1}^0)\right)\right\|^2 - \left\|\mathbf{z}_t - \mathbf{z}^*\left(\mathbf{x}_t^0, \mathbf{y}^*(\mathbf{x}_t^0)\right)\right\|^2\right]$$

$$\leq -\frac{\gamma_t \mu_g}{2}\mathbb{E}\left[\left\|\mathbf{z}_t - \mathbf{z}_t^*\right\|^2\right] + \frac{2}{\gamma_t \mu_g}L_z^2 \alpha_t^2 N \sum_{n=0}^{N-1}\mathbb{E}\left[\left\|h_{t,n}^f\right\|^2\right] + 16\sigma_{g_{yy}}^2 \frac{B_{f_y}^2}{\mu_g^2}\gamma_t^2 + 8\sigma_{f_y}^2 \gamma_t^2. \quad (32)$$

Adding Eq. (30), (31) and (32), we get

$$\mathbb{E}\left[\bar{W}_{t+1} - \bar{W}_t\right]$$

$$\leq -\frac{\alpha_t}{2}\sum_{n=0}^{N-1}\mathbb{E}\left[\left\|\nabla l\left(\mathbf{x}_t^n\right)\right\|^2\right] + \bar{C}_y \sum_{n=0}^{N-1}\mathbb{E}\left[\left\|\mathbf{y}_t^n - \mathbf{y}^*\left(\mathbf{x}_t^n\right)\right\|^2\right] + \bar{C}_z \mathbb{E}\left[\left\|\mathbf{z}_t - \mathbf{z}_t^*\right\|^2\right]$$

$$+ \bar{C}_h \sum_{n=0}^{N-1}\mathbb{E}\left[\left\|h_{t,n}^f\right\|^2\right] + 4B_z^2 \sigma_{g_{xy}}^2 \alpha_t N + 2\sigma_{f_x}^2 \alpha_t N + 16L_{g_{xy}}^2 B_z^2 \beta_t^2 n^2 \sigma_{g_y}^2 \alpha_t N$$

$$+ 4\beta_t^2 \sigma_{g_y}^2 N K_y + 16\sigma_{g_{yy}}^2 \frac{B_{f_y}^2}{\mu_g^2}\gamma_t^2 K_z + 8\sigma_{f_y}^2 \gamma_t^2 K_z,$$

where

$$\bar{C}_y = 2\alpha_t L_f^2 + 16L_g^2 \alpha_t L_{g_{xy}}^2 B_z^2 \beta_t^2 N^2 - \frac{\beta_t \mu_g}{2}K_y,$$

$$\bar{C}_z = 4\alpha_t L_f^2 N - \frac{\gamma_t \mu_g}{2}K_z,$$

$$\bar{C}_h = -\frac{\alpha_t}{2} + \frac{\alpha_t^2 L_l}{2} + 4L_f^2 L_z^2 \alpha_t^3 N^2 + 8L_{g_{xy}}^2 B_z^2 \alpha_t^3 N^2 + \frac{2}{\beta_t \mu_g}L_y^2 \alpha_t^2 K_y + \frac{2}{\gamma_t \mu_g}L_z^2 \alpha_t^2 N K_z.$$

Define $\beta_t \triangleq c_\beta \alpha_t$, and $\gamma_t \triangleq c_\gamma \alpha_t$.

To ensure $\bar{C}_y \leq 0$, we have

$$\bar{C}_y = 2\alpha_t L_f^2 + 16L_g^2 \alpha_t L_{g_{xy}}^2 B_z^2 \beta_t^2 N^2 - \frac{\beta_t \mu_g}{2}K_y \overset{(a)}{\leq} 4\alpha_t L_f^2 - \frac{c_\beta \alpha_t \mu_g}{2}K_y \overset{(b)}{\leq} 0,$$

where (a) uses $\alpha_t \leq \frac{\sqrt{2}L_f}{4L_{g_{xy}}B_z c_\beta N L_g}$, and (b) follows from $c_\beta = \frac{8L_f^2}{\mu_g K_y}$.

To ensure $\bar{C}_z \leq 0$, we have

$$\bar{C}_z = 4\alpha_t L_f^2 N - \frac{\gamma_t \mu_g}{2}K_z \overset{(a)}{\leq} 0,$$

where (a) utilizes $c_\gamma = \frac{8L_f^2 N}{\mu_g K_z}$.

To ensure $\bar{C}_h \leq 0$, we have

$$\bar{C}_h = -\frac{\alpha_t}{2} + \frac{\alpha_t^2 L_l}{2} + 4L_f^2 L_z^2 \alpha_t^3 N^2 + 8L_{g_{xy}}^2 B_z^2 \alpha_t^3 N^2 + \frac{2}{\beta_t \mu_g}L_y^2 \alpha_t^2 K_y + \frac{2}{\gamma_t \mu_g}L_z^2 \alpha_t^2 N K_z$$

$$\overset{(a)}{\leq} -\frac{\alpha_t}{2} + \frac{\alpha_t}{4} + \frac{\alpha_t}{16} + \frac{\alpha_t}{16} + \frac{\alpha_t}{16} + \frac{\alpha_t}{16} = 0,$$

where (a) results from $\alpha_t \leq \min\left\{\frac{1}{2L_t}, \frac{1}{8L_f L_z N}, \frac{\sqrt{2}}{16L_{g_{xy}}B_z N}\right\}$, $K_y = \frac{c_\beta \mu_g}{32L_y^2}$, and $K_z = \frac{c_\gamma \mu_g}{32L_z^2 N}$.

Towards this end, the lemma is proved. $\qquad\square$

### F.2.5   PROOF OF THEOREM 5.7

**Theorem F.5** (Non-Convex $\ell(\mathbf{x})$). *Under Assumptions 5.1–5.3, choose constant step-sizes $\alpha_t = \alpha = \bar{\alpha}$, $\beta_t = \beta \triangleq c_\beta \alpha$, and $\gamma_t = \gamma \triangleq c_\gamma \alpha$ for all $t \in \{0, 1, \ldots, T\}$ with $c_\beta = \frac{16 L_f L_y}{\mu_g}$ and $c_\gamma = \frac{16 L_f L_z N}{\mu_g}$. Moreover, choose $\bar{\alpha}$ such that*

$$\bar{\alpha} \leq \min \left\{ \frac{\mu_g}{2 L_g^2 c_\beta}, \frac{2}{3 \mu_g c_\beta}, \frac{2}{3 \mu_g c_\gamma}, \frac{\mu_g}{(8 \sigma_{g_{yy}}^2 + 2 B_{g_{yy}}^2) c_\gamma}, \frac{1}{2 L_l}, \right.$$

$$\left. \frac{1}{8 L_f L_z N}, \frac{1}{8 \sqrt{2} L_{g_{xy}} B_z N}, \frac{\sqrt{2} L_f}{4 L_{g_{xy}} B_z c_\beta L_g N} \right\}.$$

*Then, the iterates generated by* SO-Lazy-BiO-SGD *in Algorithm 3 satisfy:*

$$\frac{1}{TN} \sum_{t=0}^{T-1} \sum_{n=0}^{N-1} \mathbb{E}\left[ \|\nabla \ell(\mathbf{x}_t^n)\|^2 \right] = \mathcal{O}\left( \frac{\Delta_0}{T} + \sigma_{g_y}^2 + \sigma_{g_{yy}}^2 + \sigma_{f_y}^2 + \sigma_{g_{xy}}^2 + \sigma_{f_x}^2 \right),$$

*where $\Delta_0 = (\ell(\mathbf{x}_0^0) - \ell^*) + \|\mathbf{y}_0^0 - \mathbf{y}^*(\mathbf{x}_0^0)\|^2 + \|\mathbf{z}_0 - \mathbf{z}^*(\mathbf{x}_0^0, \mathbf{y}^*(\mathbf{x}_0^0))\|^2.$*

*Proof.* Choose $\alpha_t$ as a constant stepsize $\alpha_t = \alpha$. Summing the result in Lemma F.4 from $t = 0$ to $T-1$, and then dividing by $NT$ on both sides, we get

$$\frac{\mathbb{E}\left[ \bar{W}_T - \bar{W}_0 \right]}{NT} \leq - \frac{\alpha}{2NT} \sum_{t=0}^{T-1} \sum_{n=0}^{N-1} \mathbb{E}\left[ \|\nabla \ell(\mathbf{x}_t^n)\|^2 \right] + 4 B_z^2 \sigma_{g_{xy}}^2 \alpha + 2 \sigma_{f_x}^2 \alpha + 8 \sigma_{f_y}^2 c_\gamma^2 \alpha^2 K_z \frac{1}{N}$$

$$+ 16 L_{g_{xy}}^2 B_z^2 c_\beta^2 N^2 \sigma_{g_y}^2 \alpha^3 + 4 c_\beta^2 \alpha^2 \sigma_{g_y}^2 K_y + 16 \sigma_{g_{yy}}^2 \frac{B_{f_y}^2}{\mu_g^2} c_\gamma^2 \alpha^2 K_z \frac{1}{N}.$$

Rearranging the terms and multiplying by $2/\alpha$ on both sides, we have

$$\frac{1}{TN} \sum_{t=0}^{T-1} \sum_{n=0}^{N-1} \mathbb{E}\left[ \|\nabla \ell(\mathbf{x}_t^n)\|^2 \right] \leq \frac{2 (\bar{W}_0 - \ell^*)}{\alpha NT} + 8 B_z^2 \sigma_{g_{xy}}^2 + 4 \sigma_{f_x}^2 + 16 \sigma_{f_y}^2 c_\gamma^2 \alpha K_z \frac{1}{N}$$

$$+ 32 L_{g_{xy}}^2 B_z^2 c_\beta^2 N^2 \sigma_{g_y}^2 \alpha^2 + 8 c_\beta^2 \alpha \sigma_{g_y}^2 K_y + 32 \sigma_{g_{yy}}^2 \frac{B_{f_y}^2}{\mu_g^2} c_\gamma^2 \alpha K_z \frac{1}{N},$$

where $W_0 = \ell(\mathbf{x}_0^0) + K_y \|\mathbf{y}_0^0 - \mathbf{y}^*(x_0^0)\|^2 + K_z \|\mathbf{z}_0 - \mathbf{z}^*(\mathbf{x}_0^0, \mathbf{y}^*(\mathbf{x}_0^0))\|^2.$

Therefore,

$$\frac{1}{TN} \sum_{t=0}^{T-1} \sum_{n=0}^{N-1} \mathbb{E}\left[ \|\nabla \ell(\mathbf{x}_t^n)\|^2 \right]$$

$$= \mathcal{O}\left( \frac{\ell(\mathbf{x}_0^0) - \ell^*}{NT\alpha} \right) + \mathcal{O}\left( \frac{\|\mathbf{y}_0^0 - \mathbf{y}^*(\mathbf{x}_0^0)\|^2}{NT\alpha} \right) + \mathcal{O}\left( \frac{\|\mathbf{z}_0 - \mathbf{z}^*(\mathbf{x}_0^0, \mathbf{y}^*(\mathbf{x}_0^0))\|^2}{NT\alpha} \right)$$

$$+ \mathcal{O}\left( \sigma_{g_{xy}}^2 + \sigma_{f_x}^2 + \sigma_{g_y}^2 N\alpha + \sigma_{g_{yy}}^2 N\alpha + \sigma_{f_y}^2 N\alpha \right).$$

By selecting $\alpha = \mathcal{O}\left(\frac{1}{N}\right)$, the proof of the theorem is completed. $\qquad\square$

## G   THEORETICAL ANALYSIS OF OPTION II IN SO-Lazy-BiO FRAMEWORK

The theoretical analyses of SO-Lazy-BiO-I and SO-Lazy-BiO-II are similar, with the primary difference arising from the approximation error in the hypergradient $\nabla \ell(\mathbf{x})$ (see Lemma G.3). SO-Lazy-BiO-II can be viewed as a special case of SO-Lazy-BiO-I , in which no errors are incurred from the JVP updates.

Both SO-Lazy-BiO-I and SO-Lazy-BiO-II share the same convergence guarantees, and the main result for SO-Lazy-BiO-II is stated in Theorem G.1.

**Theorem G.1** (Convergence Rate of SO-Lazy-BiO-II). *Under Assumptions 5.1–5.3, choose constant step-sizes $\alpha_t = \alpha = \mathcal{O}((\sqrt{N}T)^{-1})$, $\beta_t = \beta = \mathcal{O}((\sqrt{N}T)^{-1})$, $\gamma_t = \gamma = \mathcal{O}(\sqrt{N}(\sqrt{T})^{-1})$, and the momentum coefficient as $\mu_t = \mu = \mathcal{O}((N\sqrt{T})^{-1})$ for all $t = 0, \dots, T - 1$. Then, the iterates generated by* SO-Lazy-BiO-II *satisfy:*

$$\frac{1}{T} \sum_{t=0}^{T-1} \mathbb{E}\left[\|\nabla \ell(\mathbf{x}_t)\|^2\right] = \mathcal{O}\left(\frac{\sqrt{N}\Delta_0}{\sqrt{T}} + \frac{\sigma_{g_y}^2}{\sqrt{NT}} + \frac{\sqrt{N}}{\sqrt{T}}\sigma_{g_{yy}}^2 + \frac{\sqrt{N}}{\sqrt{T}}\sigma_{f_y}^2 + \frac{\sigma_{g_{xy}}^2}{\sqrt{NT}} + \frac{\sigma_{f_x}^2}{\sqrt{NT}}\right),$$

*where $\Delta_0 = (\ell(\mathbf{x}_0) - \ell^*) + \|\mathbf{y}_0 - \mathbf{y}^*(\mathbf{x}_0)\|^2 + \|\mathbf{z}_0 - \mathbf{z}^*(\mathbf{x}_0, \mathbf{y}^*(\mathbf{x}_0))\|^2$.*

Moreover, the computation complexity of SO-Lazy-BiO-II immediately follows from Theorem G.1:

**Corollary G.2** (Computation Complexity of SO-Lazy-BiO-II). *Under the setting of Theorem G.1, choose the batch size as $\mathcal{O}(1)$. Then,* SO-Lazy-BiO-II *requires $\mathcal{O}(N\epsilon^{-2})$ partial gradient evaluations and JVP evaluations and $\mathcal{O}(\epsilon^{-2})$ HVP evaluations to reach an $\epsilon$-stationary solution.*

### G.1 REFORMULATION OF **OPTION II** IN ALGORITHM 1 FOR THEORETICAL ANALYSIS

In order to analyze the theoretical performance of SO-Lazy-BiO-II, we reformulate SO-Lazy-BiO-II as follows. We note that **Option II** in Algorithm 1 is equivalent to Algorithm 4 when the number of iterations $T$ in Algorithm 4 is set to $T/N$.

---

**Algorithm 4** The SO-Lazy-BiO-II Algorithm.

**Input:** Initial parameters $\mathbf{x}_0^0, \mathbf{y}_0^0, \mathbf{z}_0$, stepsizes $\{\alpha_t, \beta_t, \gamma_t\}_{t=0}^{T-1}$, and momentum coefficient $\{\mu_t\}_{t=0}^{T-1}$
**for** $t = 0$ to $T - 1$ **do**
    Initialize $\mathbf{x}_t^0 = \mathbf{x}_{t-1}^N$ and $\mathbf{y}_t^0 = \mathbf{y}_{t-1}^N$
    Sample data batches $\mathcal{D}_t^{g_{yy}}$ and $\mathcal{D}_t^{f_y}$
    Compute the gradient estimate $h_t^q$ using $h_t^q = \nabla_{\mathbf{yy}}^2 g(\mathbf{x}_t^0, \mathbf{y}_t^0; \mathcal{D}_t^{g_{yy}})\mathbf{z}_t + \nabla_{\mathbf{y}} f(\mathbf{x}_t^0, \mathbf{y}_t^0; \mathcal{D}_t^{f_y})$
    Update $\mathbf{z}_{t+1} = \mathbf{z}_t - \gamma_t h_t^q$
    **for** $n = 0$ to $N - 1$ **do**
        Sample data batches $\mathcal{D}_{t,n}^g$, $\mathcal{D}_{t,n}^{f_x}$, and $\mathcal{D}_{t,n}^{g_{xy}}$
        Compute the gradient estimate $h_{t,n}^g$ using $h_{t,n}^g = \nabla_{\mathbf{y}} g\left(\mathbf{x}_t^n, \mathbf{y}_t^n; \mathcal{D}_{t,n}^g\right)$
        Update $\mathbf{y}_t^{n+1} = \mathbf{y}_t^n - \beta_t h_{t,n}^g$
        Compute the gradient estimate $h_{t,n}^f$ using $h_{t,n}^f = \nabla_{\mathbf{x}} f\left(\mathbf{x}_t^n, \mathbf{y}_t^n; \mathcal{D}_{t,n}^{f_x}\right) + \nabla_{\mathbf{xy}}^2 g\left(\mathbf{x}_t^n, \mathbf{y}_t^n; \mathcal{D}_{t,n}^{g_{xy}}\right)\mathbf{z}_t$
        Compute the momentum-based $\bar{h}_{t,n+1}^f$ using $\bar{h}_{t,n+1}^f = \mu_t h_{t,n}^f + (1 - \mu_t)\bar{h}_{t,n}^f$
        Update $\mathbf{x}_t^{n+1} = \mathbf{x}_t^n - \alpha_t \bar{h}_{t,n}^f$
    **end for**
**end for**

---

### G.2 DETAILED PROOF OF THEOREM G.1: NON-CONVEX $\ell(\mathbf{x})$

#### G.2.1 DESCENT IN THE APPROXIMATION ERROR OF $\nabla \ell(\mathbf{x})$

**Lemma G.3.** *Under Assumptions 5.1–5.3, the approximation error of $\nabla \ell(\mathbf{x})$ of Algorithm 4 satisfies the following inequality:*

$$\mathbb{E}\left[\left\|\nabla \ell\left(\mathbf{x}_t^{n+1}\right) - \bar{h}_{t,n+1}^f\right\|^2\right]$$

$$\leq (1 - \mu_t)\mathbb{E}\left[\left\|\nabla \ell\left(\mathbf{x}_t^n\right) - \bar{h}_{t,n}^f\right\|^2\right] + 4\mu_t L_f^2 \mathbb{E}\left[\left\|\mathbf{y}_t^n - \mathbf{y}^*\left(\mathbf{x}_t^n\right)\right\|^2\right] + 8\mu_t L_f^2 \mathbb{E}\left[\left\|\mathbf{z}_t - \mathbf{z}_t^*\right\|^2\right]$$

$$+ \frac{2}{\mu_t} L_l^2 \alpha_t^2 \mathbb{E}\left[\left\|\bar{h}_{t,n}^f\right\|^2\right] + 8\mu_t L_f^2 L_z^2 \alpha_t^2 n \sum_{i=0}^{n-1} \mathbb{E}\left[\left\|\bar{h}_{t,i}^f\right\|^2\right] + 2B_z^2 \sigma_{g_{xy}}^2 \mu_t^2 + 2\sigma_{f_x}^2 \mu_t^2,$$

*for all $t \in \{0, 1, \dots, T - 1\}$ and $n \in \{0, 1, \dots, N - 1\}$, where $\mathbf{z}_t^* = \mathbf{z}^*\left(\mathbf{x}_t^0, \mathbf{y}^*(\mathbf{x}_t^0)\right)$, and the expectation is taken over the stochasticity of the algorithm.*

*Proof.* Same as Eq. (10), we have

$$
\mathbb{E}\left[\left\|\nabla \ell\left(\mathbf{x}_t^{n+1}\right) - \bar{h}_{t,n+1}^f\right\|^2\right] \leq \mathbb{E}\left[(1-\mu_t)\left\|\bar{h}_{t,n}^f - \nabla \ell\left(\mathbf{x}_t^n\right)\right\|^2 + \mu_t^2\left\|h_{t,n}^f - \nabla f\left(\mathbf{x}_t^n, \mathbf{y}_t^n, \mathbf{z}_t\right)\right\|^2\right.
$$

$$
\left. +4\mu_t L_f^2\left\|\mathbf{y}_t^n - \mathbf{y}^*\left(\mathbf{x}_t^n\right)\right\|^2 + 4\mu_t L_f^2\left\|\mathbf{z}_t - \mathbf{z}^*\left(\mathbf{x}_t^n, \mathbf{y}^*\left(\mathbf{x}_t^n\right)\right)\right\|^2 + \frac{2}{\mu_t}L_l^2\alpha_t^2\left\|\bar{h}_{t,n}^f\right\|^2\right]. \tag{33}
$$

We bound the term $\mathbb{E}\left[\left\|h_{t,n}^f - \nabla f\left(\mathbf{x}_t^n, \mathbf{y}_t^n, \mathbf{z}_t\right)\right\|^2\right]$ in Eq. (33).

$$
\mathbb{E}\left[\left\|h_{t,n}^f - \nabla f\left(\mathbf{x}_t^n, \mathbf{y}_t^n, \mathbf{z}_t\right)\right\|^2\right]
$$

$$
\stackrel{(a)}{=} \mathbb{E}\left[\left\|\nabla_{\mathbf{x}}f\left(\mathbf{x}_t^n, \mathbf{y}_t^n, \mathcal{D}_{t,n}^{f_x}\right) + \nabla_{\mathbf{xy}}^2 g\left(\mathbf{x}_t^n, \mathbf{y}_t^n, \mathcal{D}_t^{g_{xy}}\right)\mathbf{z}_t - \nabla_{\mathbf{x}}f\left(\mathbf{x}_t^n, \mathbf{y}_t^n\right) - \nabla_{\mathbf{xy}}^2 g\left(\mathbf{x}_t^n, \mathbf{y}_t^n\right)\mathbf{z}_t\right\|^2\right]
$$

$$
\leq \mathbb{E}\left[2\left\|\nabla_{\mathbf{x}}f\left(\mathbf{x}_t^n, \mathbf{y}_t^n, \mathcal{D}_{t,n}^{f_x}\right) - \nabla_{\mathbf{x}}f\left(\mathbf{x}_t^n, \mathbf{y}_t^n\right)\right\|^2\right.
$$

$$
\left. +2\left\|\mathbf{z}_t\right\|^2\left\|\nabla_{\mathbf{xy}}^2 g\left(\mathbf{x}_t^n, \mathbf{y}_t^n, \mathcal{D}_t^{g_{xy}}\right) - \nabla_{\mathbf{xy}}^2 g\left(\mathbf{x}_t^n, \mathbf{y}_t^n\right)\right\|^2\right] \stackrel{(b)}{\leq} 2B_z^2\sigma_{g_{xy}}^2 + 2\sigma_{f_x}^2, \tag{34}
$$

where (a) uses the definitions of $h_{t,n}^f$ and $\nabla f\left(\mathbf{x}_t^n, \mathbf{y}_t^n, \mathbf{z}_t\right)$. (b) utilizes the bounded variance in Assumption 5.3 and $\|\mathbf{z}_t\| \leq B_z$.

Same as Eq. (14), we have

$$
\mathbb{E}\left[\left\|\mathbf{z}_t - \mathbf{z}^*\left(\mathbf{x}_t^n, \mathbf{y}^*\left(\mathbf{x}_t^n\right)\right)\right\|^2\right] \leq \mathbb{E}\left[2\left\|\mathbf{z}_t - \mathbf{z}^*\left(\mathbf{x}_t^0, \mathbf{y}^*\left(\mathbf{x}_t^0\right)\right)\right\|^2 + 2L_z^2\alpha_t^2 n\sum_{i=0}^{n-1}\left\|\bar{h}_{t,i}^f\right\|^2\right], \tag{35}
$$

Combining Eq. (33), (34), and (35) completes the proof of the lemma. $\qquad\square$

### G.2.2 DESCENT IN THE POTENTIAL FUNCTION

We define the potential function $\hat{W}_t$ as follows:

$$
\hat{W}_t = \ell\left(\mathbf{x}_t^0\right) + K_y\left\|\mathbf{y}_t^0 - \mathbf{y}^*\left(\mathbf{x}_t^0\right)\right\|^2 + K_z\left\|\mathbf{z}_t - \mathbf{z}^*\left(\mathbf{x}_t^0, \mathbf{y}^*(\mathbf{x}_t^0)\right)\right\|^2 + K_h\left\|\nabla\ell\left(\mathbf{x}_t^0\right) - \bar{h}_{t,0}^f\right\|^2.
$$

**Lemma G.4.** *Under the same conditions as described in Theorem G.5 and using Lemmas E.2, E.4, E.5, and G.3, the iterates generated by Algorithm 4 satisfies: for all $t \in \{0, 1, \ldots, T-1\}$,*

$$
\mathbb{E}\left[\hat{W}_{t+1} - \hat{W}_t\right] \leq -\frac{\alpha_t}{2}\sum_{n=0}^{N-1}\mathbb{E}\left[\left\|\nabla\ell\left(\mathbf{x}_t^n\right)\right\|^2\right] + 2B_z^2\sigma_{g_{xy}}^2 c_\mu^2\alpha_t^2 N K_h + 2\sigma_{f_x}^2 c_\mu^2\alpha_t^2 N K_h
$$

$$
+ 4c_\beta^2\alpha_t^2\sigma_{g_y}^2 N K_y + 16\sigma_{g_{yy}}^2\frac{B_{f_y}^2}{\mu_g^2}c_\gamma^2\alpha_t^2 K_z + 8\sigma_{f_y}^2 c_\gamma^2\alpha_t^2 K_z,
$$

*where $K_y = \frac{\sqrt{2}L_f}{4L_y}$, $K_z = \frac{\sqrt{2}L_f}{2L_z}$, and $K_h = \frac{1}{8L_l}$.*

*Proof.* Based on Lemma G.3, we have

$$
\mathbb{E}\left[\left\|\nabla\ell\left(\mathbf{x}_t^{n+1}\right) - \bar{h}_{t,n+1}^f\right\|^2 - \left\|\nabla\ell\left(\mathbf{x}_t^n\right) - \bar{h}_{t,n}^f\right\|^2\right]
$$

$$
\leq -\mu_t\mathbb{E}\left[\left\|\nabla\ell\left(\mathbf{x}_t^n\right) - \bar{h}_{t,n}^f\right\|^2\right] + 4\mu_t L_f^2\mathbb{E}\left[\left\|\mathbf{y}_t^n - \mathbf{y}^*\left(\mathbf{x}_t^n\right)\right\|^2\right] + 8\mu_t L_f^2\mathbb{E}\left[\left\|\mathbf{z}_t - \mathbf{z}_t^*\right\|^2\right]
$$

$$
+ \frac{2}{\mu_t}L_l^2\alpha_t^2\mathbb{E}\left[\left\|\bar{h}_{t,n}^f\right\|^2\right] + 8\mu_t L_f^2 L_z^2\alpha_t^2 n\sum_{i=0}^{n-1}\mathbb{E}\left[\left\|\bar{h}_{t,i}^f\right\|^2\right] + 2B_z^2\sigma_{g_{xy}}^2\mu_t^2 + 2\sigma_{f_x}^2\mu_t^2.
$$

This implies that

$$\sum_{n=0}^{N-1} \mathbb{E}\left[\left\|\nabla\ell\left(\mathbf{x}_t^{n+1}\right) - \bar{h}_{t,n+1}^f\right\|^2 - \left\|\nabla\ell\left(\mathbf{x}_t^n\right) - \bar{h}_{t,n}^f\right\|^2\right]$$

$$= \mathbb{E}\left[\left\|\nabla\ell\left(\mathbf{x}_{t+1}^0\right) - \bar{h}_{t+1,0}^f\right\|^2 - \left\|\nabla\ell\left(\mathbf{x}_t^0\right) - \bar{h}_{t,0}^f\right\|^2\right]$$

$$\leq -\mu_t \sum_{n=0}^{N-1} \mathbb{E}\left[\left\|\nabla\ell\left(\mathbf{x}_t^n\right) - \bar{h}_{t,n}^f\right\|^2\right] + 4\mu_t L_f^2 \sum_{n=0}^{N-1} \mathbb{E}\left[\left\|\mathbf{y}_t^n - \mathbf{y}^*\left(\mathbf{x}_t^n\right)\right\|^2\right]$$

$$+ \frac{2}{\mu_t} L_l^2 \alpha_t^2 \sum_{n=0}^{N-1} \mathbb{E}\left[\left\|\bar{h}_{t,n}^f\right\|^2\right] + 8\mu_t L_f^2 L_z^2 \alpha_t^2 N^2 \sum_{n=0}^{N-1} \mathbb{E}\left[\left\|\bar{h}_{t,n}^f\right\|^2\right]$$

$$+ 8\mu_t L_f^2 N \mathbb{E}\left[\left\|\mathbf{z}_t - \mathbf{z}_t^*\right\|^2\right] + 2B_z^2 \sigma_{g_{xy}}^2 \mu_t^2 N + 2\sigma_{f_x}^2 \mu_t^2 N. \tag{36}$$

Adding Eq. (22), (24), (25) and (36), we get

$$\mathbb{E}\left[\hat{W}_{t+1} - \hat{W}_t\right]$$

$$\leq -\frac{\alpha_t}{2} \sum_{n=0}^{N-1} \mathbb{E}\left[\left\|\nabla l\left(\mathbf{x}_t^n\right)\right\|^2\right] + \hat{C}_y \sum_{n=0}^{N-1} \mathbb{E}\left[\left\|\mathbf{y}_t^n - \mathbf{y}^*\left(\mathbf{x}_t^n\right)\right\|^2\right] + \hat{C}_z \mathbb{E}\left[\left\|\mathbf{z}_t - \mathbf{z}_t^*\right\|^2\right]$$

$$+ \hat{C}_h \sum_{n=0}^{N-1} \mathbb{E}\left[\left\|\bar{h}_{t,n}^f\right\|^2\right] + \hat{C}_l \sum_{n=0}^{N-1} \mathbb{E}\left[\left\|\nabla\ell\left(\mathbf{x}_t^n\right) - \bar{h}_{t,n}^f\right\|^2\right]$$

$$+ 2B_z^2 \sigma_{g_{xy}}^2 \mu_t^2 N K_h + 2\sigma_{f_x}^2 \mu_t^2 N K_h + 4\beta_t^2 \sigma_{g_y}^2 N K_y + 16\sigma_{g_{yy}}^2 \frac{B_{f_y}^2}{\mu_g^2} \gamma_t^2 K_z + 8\sigma_{f_y}^2 \gamma_t^2 K_z,$$

where

$$\hat{C}_y = 4\mu_t L_f^2 K_h - \frac{\beta_t \mu_g}{2} K_y,$$

$$\hat{C}_z = 8\mu_t L_f^2 N K_h - \frac{\gamma_t \mu_g}{2} K_z,$$

$$\hat{C}_h = -\frac{\alpha_t}{2} + \frac{\alpha_t^2 L_l}{2} + \frac{2}{\mu_t} L_l^2 \alpha_t^2 K_h + 8\mu_t L_f^2 L_z^2 \alpha_t^2 N^2 K_h + \frac{2}{\beta_t \mu_g} L_y^2 \alpha_t^2 K_y + \frac{2}{\gamma_t \mu_g} L_z^2 \alpha_t^2 N K_z,$$

$$\hat{C}_l = \frac{\alpha_t}{2} - \mu_t K_h.$$

Define $\beta_t \triangleq c_\beta \alpha_t$, $\gamma_t \triangleq c_\gamma \alpha_t$, and $\mu_t \triangleq c_\mu \alpha_t$.

To ensure $\hat{C}_y \leq 0$, we have

$$\hat{C}_y = 4\mu_t L_f^2 K_h - \frac{\beta_t \mu_g}{2} K_y \overset{(a)}{\leq} 0,$$

where (a) uses $K_y = \frac{8c_\mu L_f^2 K_h}{\mu_g c_\beta}$.

To ensure $\hat{C}_z \leq 0$, we have

$$\hat{C}_z = 8\mu_t L_f^2 N K_h - \frac{\gamma_t \mu_g}{2} K_z \overset{(a)}{\leq} \frac{c_\gamma \alpha_t \mu_g}{2} K_z - \frac{c_\gamma \alpha_t \mu_g}{2} K_z = 0,$$

where (a) utilizes $K_z = \frac{16c_\mu L_f^2 N K_h}{\mu_g c_\gamma}$.

To ensure $\hat{C}_h \leq 0$, we have

$$\hat{C}_h = -\frac{\alpha_t}{2} + \frac{\alpha_t^2 L_l}{2} + \frac{2}{\mu_t} L_l^2 \alpha_t^2 K_h + 8\mu_t L_f^2 L_z^2 \alpha_t^2 N^2 K_h + \frac{2}{\beta_t \mu_g} L_y^2 \alpha_t^2 K_y + \frac{2}{\gamma_t \mu_g} L_z^2 \alpha_t^2 N K_z$$

$$\overset{(a)}{\leq} -\frac{\alpha_t}{2} + \frac{\alpha_t^2 L_l}{2} + \frac{4}{c_\mu} L_l^2 \alpha_t K_h + \frac{2}{c_\beta \mu_g} L_y^2 \alpha_t K_y + \frac{2}{c_\gamma \mu_g} L_z^2 \alpha_t N K_z$$

$$\overset{(b)}{\leq} -\frac{\alpha_t}{2} + \frac{\alpha_t}{8} + \frac{\alpha_t}{8} + \frac{\alpha_t}{8} + \frac{\alpha_t}{8} = 0,$$

where (a) is because of $\alpha_t \leq \frac{L_l}{2c_\mu L_f L_z N}$. (b) follows from $\alpha_t \leq \frac{1}{4L_l}$, $c_\mu = 32L_l^2 K_h$, $c_\beta = \frac{16 L_y^2 K_y}{\mu_g}$, and $c_\gamma = \frac{16 L_z^2 N K_z}{\mu_g}$.

To ensure $\hat{C}_l \leq 0$, we have

$$\hat{C}_l = \frac{\alpha_t}{2} - \mu_t K_h \overset{(a)}{\leq} 0,$$

where (a) is due to $K_h = \frac{1}{2c_\mu}$.

Towards this end, the lemma is proved. $\qquad\square$

### G.2.3 PROOF OF THEOREM G.1

**Theorem G.5** (Non-Convex $\ell(\mathbf{x})$). *Under Assumptions 5.1–5.3, choose constant step-sizes $\alpha_t = \alpha = \frac{\bar{\alpha}}{N\sqrt{T}}$, $\beta_t = \beta \triangleq c_\beta \alpha$, $\gamma_t = \gamma \triangleq c_\gamma \alpha$, and the momentum coefficient as $\mu_t = \mu \triangleq c_\mu \alpha$ for all $t \in \{0, 1, \ldots, T\}$ with $c_\beta = \frac{16 L_y L_f}{\sqrt{2}\mu_g}$, $c_\gamma = \frac{16 L_z L_f N}{\sqrt{2}\mu_g}$, and $c_\mu = 4L_l$. Moreover, choose $\bar{\alpha}$ such that*

$$\bar{\alpha} \leq \min\left\{ \frac{\mu_g}{2L_g^2 c_\beta}, \frac{2}{3\mu_g c_\beta}, \frac{\mu_g}{\left(8\sigma_{g_{yy}}^2 + 2B_{g_{yy}}^2\right) c_\gamma}, \frac{2}{3\mu_g c_\gamma}, \frac{1}{4L_l}, \frac{L_l}{2c_\mu L_f L_z N} \right\}.$$

*Then, the iterates generated by SO-Lazy-BiO-II in Algorithm 4 satisfy:*

$$\frac{1}{TN} \sum_{t=0}^{T-1} \sum_{n=0}^{N-1} \mathbb{E}\left[\|\nabla \ell\left(\mathbf{x}_t^n\right)\|^2\right] = \mathcal{O}\left( \frac{\Delta_0}{\sqrt{T}} + \frac{\sigma_{g_y}^2}{N\sqrt{T}} + \frac{\sigma_{g_{yy}}^2}{\sqrt{T}} + \frac{\sigma_{f_y}^2}{\sqrt{T}} + \frac{\sigma_{g_{xy}}^2}{N\sqrt{T}} + \frac{\sigma_{f_x}^2}{N\sqrt{T}} \right),$$

*where $\Delta_0 = (\ell(\mathbf{x}_0^0) - \ell^*) + \|\mathbf{y}_0^0 - \mathbf{y}^*(\mathbf{x}_0^0)\|^2 + \|\mathbf{z}_0 - \mathbf{z}^*(\mathbf{x}_0^0, \mathbf{y}^*(\mathbf{x}_0^0))\|^2$.*

*Proof.* Choose $\alpha_t$ as a constant stepsize $\alpha_t = \alpha$. Summing the result in Lemma G.4 from $t = 0$ to $T - 1$, and then dividing by $NT$ on both sides, we get

$$\frac{\mathbb{E}\left[W_T - W_0\right]}{NT} \leq -\frac{\alpha}{2NT} \sum_{t=0}^{T-1} \sum_{n=0}^{N-1} \mathbb{E}\left[\|\nabla \ell\left(\mathbf{x}_t^n\right)\|^2\right] + 2B_z^2 \sigma_{g_{xy}}^2 c_\mu^2 \alpha^2 K_h + 2\sigma_{f_x}^2 c_\mu^2 \alpha^2 K_h$$

$$+ 4c_\beta^2 \alpha^2 \sigma_{g_y}^2 K_y + 16\sigma_{g_{yy}}^2 \frac{B_{f_y}^2}{\mu_g^2} c_\gamma^2 \alpha^2 K_z \frac{1}{N} + 8\sigma_{f_y}^2 c_\gamma^2 \alpha^2 K_z \frac{1}{N}.$$

Rearranging the terms and multiplying by $2/\alpha$ on both sides, we have

$$\frac{1}{TN} \sum_{t=0}^{T-1} \sum_{n=0}^{N-1} \mathbb{E}\left[\|\nabla \ell\left(\mathbf{x}_t^n\right)\|^2\right] \leq \frac{2\left(W_0 - \ell^*\right)}{\alpha NT} + 4B_z^2 \sigma_{g_{xy}}^2 c_\mu^2 \alpha K_h + 4\sigma_{f_x}^2 c_\mu^2 \alpha K_h$$

$$+ 8c_\beta^2 \alpha \sigma_{g_y}^2 K_y + 32\sigma_{g_{yy}}^2 \frac{B_{f_y}^2}{\mu_g^2} c_\gamma^2 \alpha K_z \frac{1}{N} + 16\sigma_{f_y}^2 c_\gamma^2 \alpha K_z \frac{1}{N},$$

where $W_0 = \ell\left(\mathbf{x}_0^0\right) + K_y \left\|\mathbf{y}_0^0 - \mathbf{y}^*\left(x_0^0\right)\right\|^2 + K_z \left\|\mathbf{z}_0 - \mathbf{z}^*\left(\mathbf{x}_0^0, \mathbf{y}^*(\mathbf{x}_0^0)\right)\right\|^2$.

Therefore,

$$\frac{1}{TN} \sum_{t=0}^{T-1} \sum_{n=0}^{N-1} \mathbb{E}\left[\|\nabla \ell\left(\mathbf{x}_t^n\right)\|^2\right]$$

$$= \mathcal{O}\left(\frac{\ell\left(\mathbf{x}_0^0\right) - \ell^*}{NT\alpha}\right) + \mathcal{O}\left(\frac{\left\|\mathbf{y}_0^0 - \mathbf{y}^*\left(\mathbf{x}_0^0\right)\right\|^2}{NT\alpha}\right) + \mathcal{O}\left(\frac{\left\|\mathbf{z}_0 - \mathbf{z}^*\left(\mathbf{x}_0^0, \mathbf{y}^*(\mathbf{x}_0^0)\right)\right\|^2}{NT\alpha}\right)$$

$$+ \mathcal{O}\left(\sigma_{g_{xy}}^2\alpha + \sigma_{f_x}^2\alpha + \sigma_{g_y}^2\alpha + \sigma_{g_{yy}}^2 N\alpha + \sigma_{f_y}^2 N\alpha\right).$$

By selecting $\alpha = \mathcal{O}\left(\frac{1}{N\sqrt{T}}\right)$, the proof of the theorem is completed. $\qquad\square$

