# OpenReview forum: "SO-Lazy-BiO: Accelerating Bilevel Optimization with Reduced Second-Order Information Computation"
_ICLR.cc/2026/Conference — Submitted to ICLR 2026_

### Official Review · Reviewer_Hero · 2025-10-23

**Soundness:** 2
**Presentation:** 3
**Contribution:** 2
**Rating:** 6
**Confidence:** 5

**Summary:**

This paper considers stochastic bilevel optimization. The authors introduce SO-Lazy-BiO (Second-Order Lazy Bilevel Optimization) — a new framework that accelerates bilevel optimization (BiO) by evaluating second-order information (Hessian/Jacobian-vector products) only infrequently, rather than at every iteration. Despite this “lazy” update, the method preserves convergence guarantees comparable to state-of-the-art (SOTA) bilevel algorithms while substantially reducing computational cost and wall-clock time.

Theoretically, the authors prove the convergence rates as well as the sampel complexity of their proposed algorithms, and conduct experiments to compare different bilevel optimization algorithms to support their claims on reducing the number of Hessian-vector products needed in bilevel optimization.

**Strengths:**

Theoreticall speaking, the authors provide a novel stochastic bilevel optimization algorithm that only updates the Hessian-related parameters every N steps where N is a hyperparameter to tune. They further provide a solid analysis of their algorithm to showcase the convergence rates and sample complexity. The results can be seen as direct generalization of existing bilevel optimization algorithms.

Empirically speaking, the authors conduct experiments ranging from computer vision to LLM alignment. The results align well with the theoretical findings. Source code is provided in the supplementary materials.

**Weaknesses:**

Major:

1. One limitation of the theory part is, the authors seem not to discuss the optimal choice of $N$. For example, the big-O bound in theorem 5.5 is about $\sqrt{N/T} + 1/\sqrt{NT}$, which obtains its minimum at $N=1$, and thus optimal $N$ should be $\mathcal{O}(1)$ if we take the ignored constants into account -- this suggests that the potential improvement on reducing the number of Hessian-vector products is of order constant, not $\epsilon$. They authors may want to discuss this in detail.

2. The tricks of lazy-Hessian already exist in literature. Introducing them to bilevel optimization does not have too much novelty despite that the work is solid.

Some other comments:

1. The momentum update for $h$ in Eq. (9) is not novel and the authors may want to add a few discussions about MA-SOBA work or some other related papers that use this momentum in their algorithms

2. The authors do not provide the motivation for using bilevel optimization for solving problems like RLHF and data reweighting.

**Questions:**

1. Could the authors provide some discussions on how to choose the optimal $N$ from a theoretical perspective? Furthermore, are there empirical findings or guidance on setting $N$ in different experiments/settings.

2. For experiments like RLHF and data reweighting, from eixsting literature we understand they could be reformulated as bilevel optimization problems. Could the authors share their insights/comments/discussions on why practitioners would choose bilevel optimization algorithms, which are typically hard to implement in large scale because of the Hessian-vector products, rather than classical optimization methods? For example RLHF was originally executed in two stages -- one reward model training and one policy model training, but bilevel optimization for RLHF seems to unify these two steps into a one-stage problem. Are there benefits of doing this, from an empirical perspective?

---

### Official Review · Reviewer_d1on · 2025-11-01

**Soundness:** 2
**Presentation:** 3
**Contribution:** 2
**Rating:** 4
**Confidence:** 4

**Summary:**

The paper studies stochastic bilevel optimization where lower-level problem is strongly-convex and proposes a lazy Hessian strategy to reduce the computational overhead of second-order bilevel methods. The method keeps the same convergence rate, reduces Hessian/Jacobian-vector calculations by orders. This paper also proposed a momentum variant without the large-batch requirement. Numerical experiments on LLM fine-tuning validates the proposed algorithm.

**Strengths:**

1. This paper has strong theoretical guarantee which has the order-wise improvement for the Hessian and Jacobian-vector computations.
2. The experiments on LLM fine-tuning task are innovative and solid.

**Weaknesses:**

1. Corollary 5.6 did not specify the requirements for $N$ to achieve such computation complexity. Also as N is a key hyperparameter in the proposed algorithm, what is the guideline for choosing N in practice?
2. The novelty appears limited: lazy updates are well studied in optimization, and the adaptations required for bilevel optimization appear to be standard combinations of bilevel tools with lazy-update strategies.
3. This result is not entirely surprising: order-wise savings in Hessian- and Jacobian–vector products via lazy updates are expected, given that fully single-loop methods already perform just one such product per round [1].

[1] Mathieu Dagr´eou, Pierre Ablin, Samuel Vaiter, and Thomas Moreau. A framework for bilevel optimization that enables stochastic and global variance reduction algorithms. NeurIPS, 2022.

**Questions:**

See weakness.

---

### Official Review · Reviewer_khk4 · 2025-11-01

**Soundness:** 2
**Presentation:** 3
**Contribution:** 2
**Rating:** 2
**Confidence:** 4

**Summary:**

The paper proposes SO-Lazy-BiO, a single-loop stochastic bilevel framework that reuses stale second-order information: the Hessian-inverse-vector product (HIVP) (via a one-step SGD solve) and the Jacobian-vector product (JVP) are updated only every $N$ steps, all other steps reuse the last values. The main theorem shows that under the nonconvex-strongly-convex bilevel setting, SO-Lazy-BiO requires $O(N\epsilon^{-2})$ partial gradient evaluations and $O(\epsilon^{-2})$ second-order information
evaluations to reach an $\epsilon$-stationary point. A momentum-free variant is also analyzed but achieves a weaker bound and requires larger batches. Experiments on (i) RLHF reward-model data-weighting, (ii) LLM-alignment data-weighting (Llama-3.2-1B), and (iii) hyper-representation with ResNet-20 report lower wall-clock and far fewer HVP/JVP calls than non-lazy baselines.

**Strengths:**

- The single-loop algorithm is described clearly with two laziness modes, and they are easy to implement.

- The proposed method preserves $O(\epsilon^{-2})$ second-order complexity while cutting HVP calls.

-  Experimental results verify that the infrequent evaluations of second-order information lead to computational savings.

**Weaknesses:**

- Corollary 5.8 requires large batch sizes of $\Theta(\epsilon\^{-1})$ and $\Theta(N\epsilon\^{-1})$, which is inconsistent with practical settings that typically use small batch sizes.

- This work does not demonstrate clear advantages over the MA-SOBA method proposed in [1], which is an optimal single-loop algorithm achieving $O(\epsilon^{-2})$ complexity with the same order of HVP/JVP calls. Moreover, [1] does not rely on large batch-size assumptions.


[1] Chen, X., Xiao, T., & Balasubramanian, K. (2024). Optimal algorithms for stochastic bilevel optimization under relaxed smoothness conditions. Journal of Machine Learning Research, 25(151), 1-51.

**Questions:**

- The Corollary 5.8 for SO-Lazy-BiO-SGD specifies batch sizes of $\Theta(\epsilon\^{-1})$ and $\Theta(N\epsilon\^{-1})$. Were these batch-size settings actually used in the experiments?

- Could you provide a detailed theoretical and technical comparison to clarify the differences and any advantages of SO-Lazy-BiO over MA-SOBA?

---

### Official Review · Reviewer_pMDx · 2025-11-05

**Soundness:** 3
**Presentation:** 3
**Contribution:** 3
**Rating:** 4
**Confidence:** 3

**Summary:**

The paper proposes a bilevel optimization method that refreshes second-order information infrequently and reuses it for several iterations while keeping first-order gradients up to date. The analysis argues that the method attains a convergence rate comparable to standard second-order approaches.

**Strengths:**

Originality comes from combining periodic reuse of Hessian and Jacobian vector products with a lightweight approximation of the Hessian inverse vector product, together with proofs that the lazy updates retain the same order of convergence.

**Weaknesses:**

The paper repeatedly claims a good balance between accuracy and cost, but does not quantify it with a rigorous oracle breakdown at a fixed target stationarity. Add a table reporting counts of upper- and lower-level gradients, counts of Hessian and Jacobian vector products, and the mini-batch sizes required to reach a common tolerance across all methods.

**Questions:**

The authors claim that the proposed algorithm strikes an effective balance in its use of second-order information. Could you quantify this more precisely relative to both fully first-order and standard second-order methods? For instance, the authors said that very large batch gradients are necessary in the fully first-order method. Is it possible to quantify the difference?

On the other hand, I understand the intuition, but the practical advantage remains unclear. The paper would be stronger if you could articulate an optimal or near-optimal policy for deploying the Hessian, or at least provide principled guidance on when and how often to use it.

---

### Meta-Review · Area_Chair_EW3T · 2026-01-05

**Summary:**

The paper proposes SO-Lazy-BiO, a stochastic bilevel optimization framework that employs a "lazy" update strategy for second-order information to reduce computational overhead while maintaining convergence rates.

**I concur with the negative consensus (initial scores 2, 4, and 6) and recommend rejection**.  While Reviewer Hero initially leaned positive, significant concerns were raised regarding the theoretical assumptions, baseline comparisons, and novelty. Furthermore, the authors did not submit a rebuttal, leaving these critical criticisms unaddressed. The decision is based on technical flaws identified by the reviewers: specifically, the missing comparison against the state-of-the-art MA-SOBA [Chen et al., 2024], which achieves optimal complexity without restrictive assumptions; the reliance on impractical large batch sizes (e.g., Corollary 5.8) that differ from standard stochastic optimization practice; and the limited novelty, as the application of lazy updates is viewed as a standard combination of existing tools rather than a significant innovation.

**Reviewer Concerns:**

**Addressed**:

None. (No rebuttal submitted).

**Outstanding**:

Comparison with MA-SOBA (Reviewer (Score 2)): The validity of the claimed contribution is undermined by the lack of comparison with this optimal single-loop baseline.

Batch Size Requirements (Reviewer (Score 2), Reviewer d1on): The discrepancy between the theoretical requirements for batch sizes ($N_H, N_J$) and practical settings remains unresolved.

Hyperparameter Selection (Reviewer d1on, Reviewer Hero): There is no provided guideline for selecting the update frequency $N$. Reviewer Hero specifically questioned whether the theoretical gain is merely constant rather than order-wise.

Motivation (Reviewer Hero): The specific advantage of using bilevel optimization for tasks like RLHF over standard two-stage approaches was not justified.

**Reviewer Scores:**

All scores remain unchanged, as the authors did not submit a rebuttal.

---

### Decision · Program_Chairs · 2026-01-26

Reject